# Neural Tangent Kernels Motivate Graph Neural Networks with Cross-Covariance Graphs

## Abstract

Neural tangent kernels (NTKs) provide a theoretical regime to analyze the learning and generalization behavior of over-parametrized neural networks. For a supervised learning task, the association between the eigenvectors of the NTK kernel and given data (a concept referred to as alignment in this paper) can govern the rate of convergence of gradient descent, as well as generalization to unseen data. Building upon this concept, we investigate NTKs and alignment in the context of graph neural networks (GNNs), where our analysis reveals that optimizing alignment translates to optimizing the graph representation or the graph shift operator in a GNN. Our results further establish the theoretical guarantees on the optimality of the alignment for a two-layer GNN and these guarantees are characterized by the graph shift operator being a function of the cross-covariance between the input and the output data. The theoretical insights drawn from the analysis of NTKs are validated by our experiments focused on a multi-variate time series prediction task for a publicly available dataset. Specifically, they demonstrate that GNNs with cross-covariance as the graph shift operator indeed outperform those that operate on the covariance matrix from only the input data.

## 1 Introduction

The remarkable success of deep learning frameworks for numerous inference tasks is well established LeCun et al. (2015). Motivated by the practical implications of the gaps between the empirical observations and theoretical foundations of deep learning, many recent works have explored various approaches to rigorously understand the theory of deep learning models. Multi-layer neural networks have been analyzed extensively in the mean-field regime Mei et al. (2018; 2019); Sirignano & Spiliopoulos (2020). The random features model has also been studied to capture the effects of the regime of parameterization and study phenomenon such as generalization, and "double descent" (see e.g., Mei & Montanari (2022);Lin & Dobriban (2021);Adlam & Pennington (2020)). Among such approaches, the NTKs, first introduced in Jacot et al. (2018), have commonly been used to study the behavior of over-parameterized neural networks Cao et al. (2021b); Bietti & Mairal (2019); and are informally defined next.

**Neural Tangent Kernel.** For any given predictor $f(\mathbf{x}; \mathbf{w}) : \mathbb{R}^{n \times 1} \times \mathbb{R}^p \to \mathbb{R}$ the NTK is the kernel matrix $\Theta$ defined by the gradient of the predictor output, $f(\mathbf{x}; \mathbf{w})$, with respect to its learnable parameters, $\mathbf{w}$, as

$$\Theta_{(\mathbf{x}_i, \mathbf{x}_j)}(\mathbf{w}) := \langle \nabla_{\mathbf{w}} f(\mathbf{x}_i; \mathbf{w}), \nabla_{\mathbf{w}} f(\mathbf{x}_j; \mathbf{w}) \rangle , \tag{1}$$

where $f(\mathbf{x}; \mathbf{w})$ represents the predictor output for input data point $\mathbf{x} \in \mathbb{R}^{n \times 1}$ with the learnable parameters represented by $\mathbf{w} \in \mathbb{R}^p$. The typical setting to study NTKs is that of neural networks in the asymptote of infinite width, where the NTK is constant with respect to the learnable parameters during training, in contrast to the finite-width scenario Jacot et al. (2018). This constancy of the NTK is a result of certain neural networks transitioning to linearity as their width goes to infinity Liu et al. (2020). NTKs have been leveraged to gain theoretical insights on the behavior of neural networks such as over-parameterized neural networks achieving zero training loss over a non-convex loss landscape Du et al. (2018), the spectral bias of neural networks Cao et al. (2021b) and the inductive biases of different neural network architectures Bietti & Mairal (2019).

In particular, the eigenspectrum of the NTK kernel has been linked with the convergence rate of gradient descent for an over-parameterized deep learning model Liu et al. (2022); Arora et al. (2019);

Wang et al. (2022a). For instance, gradient descent can achieve faster convergence for a supervised learning problem if the vector of output labels, $\mathbf{y}$, aligns well with the dominant eigenvectors of the NTK matrix $\Theta$ Arora et al. (2019); Wang et al. (2022a). For the regression problem pertaining to predicting $\mathbf{y}$ from $\mathbf{x}$, our analysis in Section 2 demonstrates that

$$\text{Convergence of gradient descent} \propto \mathbf{y}^\mathsf{T}\Theta\mathbf{y} \tag{2}$$

By leveraging the observation above as a motivation, we define $\mathbf{y}^\mathsf{T}\Theta\mathbf{y}$ as Alignment $\mathcal{A}$.

**GNNs and NTKs.** Given that the NTK $\Theta$ depends on input data $\mathbf{x}$ (see equation 1), the alignment $\mathcal{A}$ inherently captures some version of covariance between output $\mathbf{y}$ and input $\mathbf{x}$. Thus, if the NTK $\Theta$ is 'structured' for a given predictor $f$, the alignment $\mathcal{A}$ could be leveraged to provide further insights into the design of the predictor $f$. GNNs are an example of a class of predictors for which the NTK is a function of the graph representation or the graph shift operator (GSO) and input data $\mathbf{x}$ Krishnagopal & Ruiz (2023). GNNs operating on covariance matrices derived from the input data have been studied previously in Sihag et al. (2022), albeit without any consideration of the insights that could be drawn from the NTKs regarding the choice of graph derived from the data for a supervised learning problem. Many of the existing works that analyze NTKs for GNNs focus on explaining empirically observed trends for GNNs (see Appendix B for expanded literature review).

In this paper, we leverage the structure of NTKs for GNNs to theoretically motivate the choice of a particular GSO. Specifically, if the NTK $\Theta$ for a GNN is considered to be a function of the form $\Theta(S, \mathbf{x})$ for a GSO $S$, the alignment can be represented as $\mathcal{A}(S, \mathbf{x}, \mathbf{y})$, i.e., as a function of the input data $\mathbf{x}$, output data $\mathbf{y}$ and $S$. It is then apparent that optimizing the alignment $\mathcal{A}$ for a GNN can inform the choice of the GSO $S$ for a given dataset. A key observation made in this paper is that the alignment $\mathcal{A}$ is characterized by the cross-covariance between the input and the output and as a result, the optimal GSO for statistical inference is closely related to the cross-covariance.

**Contributions.** In this paper, we consider the setting where the predictor $f$ is a GNN with graph filter as the convolution operator Gama et al. (2020). Our theoretical contributions in this context are summarized next.

- Our analysis of alignment $\mathcal{A}$ with graph filter as the predictor motivates cross-covariance between the input and output data as the graph. More precisely, we pose an optimization problem with alignment $\mathcal{A}$ as the objective function and demonstrate that using the cross-covariance as the GSO maximizes a lower bound on this objective.
- We further extend the results from the graph filter to the scenario of a two-layer GNN as the predictor. Our results show that under certain assumptions, the cross-covariance between the input and the ouput optimizes a lower bound on the alignment for the GNN that has $\tanh$ activation function. Thus, our analysis motivates using cross-covariance based graphs for a GNN as well.

We validated the insights drawn fron our theoretical results via experiments on the publicly available resting state functional magnetic resonance imaging (rfMRI) data from the Human Connectome Project-Young Adult (HCP-YA) dataset Van Essen et al. (2012). In particular, we considered the task of time series prediction and observed that the GNNs that operated on the cross-covariance between the input and output data achieved better convergence and generalization than those that used the covariance matrix only from the input data.

## 2 ALIGNMENT AND CONVERGENCE OF GRADIENT DESCENT

In this section, we formalize the concept of alignment $\mathcal{A}$ and demonstrate its relationship with the convergence of gradient descent for a regression problem. Consider a dataset $\{(\mathbf{x}_i, \mathbf{y}_i)\}_{i=1}^{M}$, where $\mathbf{x}_i \in \mathbb{R}^{n \times 1}$, $\mathbf{y}_i \in \mathbb{R}^{n \times 1}$. We aim to leverage the inputs $\mathbf{x}_i$ to estimate the outputs $\mathbf{y}_i$ using a predictor denoted by $\boldsymbol{f} : \mathbb{R}^{n \times 1} \times \mathbb{R}^p \to \mathbb{R}^n$. We use the notation $\boldsymbol{h} \in \mathbb{R}^p$ to denote the vector of all learnable parameters of the predictor. To emphasize the dependence of the predictor $\boldsymbol{f}$ on the parameters $\boldsymbol{h}$, we use the notation $\boldsymbol{f}_{\mathbf{x}_i}(\boldsymbol{h})$ for $\boldsymbol{f}(\mathbf{x}_i, \boldsymbol{h})$ subsequently. Also the parameters are initialized randomly from a Gaussian distribution $\boldsymbol{h}^{(0)} \sim \mathcal{N}(0, \kappa^2 I)$, where the constant $\kappa$ controls the magnitude of the initialized parameters. The objective is to minimize the mean squared error (MSE) loss function, defined as

$$\Phi(\boldsymbol{h}) \triangleq \min_{\boldsymbol{h} \in \mathbb{R}^p} \frac{1}{2} \sum_{i=1}^{M} ||\mathbf{y}_i - \boldsymbol{f}_{\mathbf{x}_i}(\boldsymbol{h})||_2^2 . \tag{3}$$

For this purpose, we consider a gradient descent based optimization framework with a learning rate $\eta > 0$. The evolution of the predictor output for a single input $\mathbf{x}_i$ is given by

$$\boldsymbol{f}_{\mathbf{x}_i}(\boldsymbol{h}^{(t+1)}) = \boldsymbol{f}_{\mathbf{x}_i}(\boldsymbol{h}^{(t)} - \eta \cdot \nabla\Phi(\boldsymbol{h}^{(t)})) \tag{4}$$

where $t$ denotes the $t$-th step or epoch of gradient descent. To characterize the evolution of the predictor output over the entire dataset, we provide the following definitions

$$\tilde{\boldsymbol{f}}_{\mathbf{x}}(\boldsymbol{h}) \triangleq \left[ [\boldsymbol{f}_{\mathbf{x}_1}(\boldsymbol{h})]^{\mathsf{T}}, \, [\boldsymbol{f}_{\mathbf{x}_2}(\boldsymbol{h})]^{\mathsf{T}}, \cdots, [\boldsymbol{f}_{\mathbf{x}_M}(\boldsymbol{h})]^{\mathsf{T}} \right]^{\mathsf{T}}, \tag{5}$$

$$\tilde{\mathbf{x}} \triangleq \left[ \mathbf{x}_1^{\mathsf{T}}, \, \mathbf{x}_2^{\mathsf{T}}, \, \cdots, \, \mathbf{x}_M^{\mathsf{T}} \right]^{\mathsf{T}}, \; \tilde{\mathbf{y}} \triangleq \left[ \mathbf{y}_1^{\mathsf{T}}, \, \mathbf{y}_2^{\mathsf{T}}, \, \cdots, \, \mathbf{y}_M^{\mathsf{T}} \right]^{\mathsf{T}} \tag{6}$$

where $\tilde{\boldsymbol{f}}_{\mathbf{x}}(\boldsymbol{h}), \tilde{\mathbf{x}}$, and $\tilde{\mathbf{y}}$ are vectors of length $nM$. We also define the NTK matrix $\tilde{\boldsymbol{\Theta}}(\boldsymbol{h}) \in \mathbb{R}^{nM \times nM}$, which consists of $M^2$ number of $n \times n$ blocks, such that, the $(i, j)$-th block is the matrix $\Theta(\mathbf{x}_i, \mathbf{x}_j) \in \mathbb{R}^{n \times n}$ and is given by

$$\Theta(\mathbf{x}_i, \mathbf{x}_j) \triangleq \mathrm{J}_{\boldsymbol{f}_{\mathbf{x}_i}}(\boldsymbol{h}^{(t)}) \big( \mathrm{J}_{\boldsymbol{f}_{\mathbf{x}_j}}(\boldsymbol{h}^{(t)}) \big)^{\mathsf{T}}. \tag{7}$$

In equation 7, $\mathrm{J}_{\boldsymbol{f}_{\mathbf{x}_i}}(\boldsymbol{h})$ denotes the Jacobian matrix with its $(a, b)$-th entry being $(\mathrm{J}_{\boldsymbol{f}_{\mathbf{x}_i}}(\boldsymbol{h}))_{ab} = \frac{\partial (\boldsymbol{f}_{\mathbf{x}_i}(\boldsymbol{h}))_a}{\partial h_b}$. If the step size $\eta$ from equation 4 is sufficiently small, the function $\boldsymbol{f}_{\mathbf{x}_i}(\boldsymbol{h}^{(t)})$ can be linearized at each step. In this scenario, the linearized version of the evolution of the predictor output in equation 4 is

$$\tilde{\boldsymbol{f}}_{\mathbf{x}}(\boldsymbol{h}^{(t+1)}) = \tilde{\boldsymbol{f}}_{\mathbf{x}}(\boldsymbol{h}^{(t)}) - \eta \cdot \tilde{\boldsymbol{\Theta}}(\boldsymbol{h}^{(t)}) \cdot (\tilde{\boldsymbol{f}}_{\mathbf{x}}(\boldsymbol{h}^{(t)}) - \tilde{\mathbf{y}}). \tag{8}$$

A typical setting of interest in the existing literature is that of the NTK $\tilde{\boldsymbol{\Theta}}(\boldsymbol{h}^{(t)})$ being a constant with respect to $\boldsymbol{h}^{(t)}$. This is because the NTK converges to a constant for many neural networks in the infinite width limit Liu et al. (2020). Theorem 1 characterizes the convergence of gradient descent for the considered multivariate regression problem in this setting (also see Arora et al. (2019), Wang et al. (2022a)). The NTK that is constant with respect to $\boldsymbol{h}^{(t)}$ is denoted by $\tilde{\boldsymbol{\Theta}}$.

**Theorem 1.** *In the multivariate regression setting, as described in the beginning of section 2, if the NTK $\tilde{\boldsymbol{\Theta}}(\boldsymbol{h}^{(t)})$ is constant during training and $\kappa = \mathcal{O}(\varepsilon\sqrt{\frac{\delta}{nM}})$, then with probability at least $1 - \delta$, the training error after $t$ steps of gradient descent is bounded as*

$$\tilde{\mathbf{y}}^{\mathsf{T}} \left( I - 2t\eta \cdot \tilde{\boldsymbol{\Theta}} \right) \tilde{\mathbf{y}} \pm \mathcal{O}(\varepsilon) \leq ||\tilde{\boldsymbol{f}}_{\mathbf{x}}(\boldsymbol{h}^{(t)}) - \tilde{\mathbf{y}}||_2^2 \leq \tilde{\mathbf{y}}^{\mathsf{T}} \left( I - \eta \cdot \tilde{\boldsymbol{\Theta}} \right) \tilde{\mathbf{y}} \pm \mathcal{O}(\varepsilon)$$

**Remark 1.** *In this paper, we primarily consider two classes of predictors. The first class is that of a linear predictor, for which the NTK is a constant given the definition in equation 7. The second class of predictors is that of infinitely wide neural networks (GNNs in particular). We refer the reader to Appendix F and Liu et al. (2020) for a detailed discussion of when and why the NTK can be a constant for neural networks.*

Since the term $\tilde{\mathbf{y}}^{\mathsf{T}}\tilde{\boldsymbol{\Theta}}\tilde{\mathbf{y}}$ characterizes the upper and lower bounds, the loss $||\tilde{\boldsymbol{f}}_{\mathbf{x}}(\boldsymbol{h}^{(t)}) - \tilde{\mathbf{y}}||_2^2$ is proportional to this term. Based on Theorem 1, we formalize the alignment in Definition 1. A similar definition can be found in Wang et al. (2022a) in the context of active learning.

**Definition 1** (Alignment). *The alignment between the output $\tilde{\mathbf{y}}$ and NTK $\tilde{\boldsymbol{\Theta}}$ is defined as*

$$\mathcal{A} \triangleq \tilde{\mathbf{y}}^{\mathsf{T}}\tilde{\boldsymbol{\Theta}}\tilde{\mathbf{y}}$$

The alignment $\mathcal{A}$ can be perceived as a metric of correlation between output data and the NTK, and is a characteristic of learning with gradient descent. Using Definition 1, Theorem 1 can be restated as

$$\tilde{\mathbf{y}}^{\mathsf{T}}\tilde{\mathbf{y}} - 2t\eta \cdot \mathcal{A} \pm \mathcal{O}(\varepsilon) \leq ||\tilde{\boldsymbol{f}}_{\mathbf{x}}(\boldsymbol{h}^{(t)}) - \tilde{\mathbf{y}}||_2^2 \leq \tilde{\mathbf{y}}^{\mathsf{T}}\tilde{\mathbf{y}} - \eta \cdot \mathcal{A} \pm \mathcal{O}(\varepsilon) \tag{9}$$

Equation (9) shows that the convergence of gradient descent is positively correlated with $\mathcal{A}$.

Recall that the NTK $\tilde{\boldsymbol{\Theta}}$ is a function of the input data $\tilde{\mathbf{x}}$ and the learning model $\boldsymbol{f}$, even when constant with respect to $\boldsymbol{h}^{(t)}$. Therefore, maximizing $\mathcal{A}$ is contingent on maximizing some kind of cross-covariance between the output data $\tilde{\mathbf{y}}$ and a function of the input data $\tilde{\mathbf{x}}$, where the function depends by the learning model $\boldsymbol{f}$. This observation motivates us to study the setting where the predictor $\boldsymbol{f}$ is a GNN, as a GNN architecture can provide appropriate structure to analyze the connection between alignment, cross-covariance and the structure of the network.

## 3 ALIGNMENT MOTIVATES CROSS-COVARIANCE IN GNNS

In this paper, we consider the GNNs for which the convolution operation is a graph filter. A graph filter is characterized by a linear-shift-and-sum operation on the input data and is representative of a large family of convolution operations in GNNs (see the section 'implementation of GCNNs' from Gama et al. (2020)). We begin with the setting where $\boldsymbol{f}_\mathbf{x}(\boldsymbol{h})$ is a graph filter.

### 3.1 NTK AND ALIGNMENT FOR GRAPH FILTER MODEL

The formal definition for a graph filter is provided in Definition 1 Gama et al. (2020).

**Definition 2** (Graph Filter). *Consider a symmetric GSO $S \in \mathbb{R}^{n \times n}$. A graph filter processes an input $\mathbf{x} \in \mathbb{R}^n$ via a linear-shift-and-sum operation characterized by $S$, such that, its output is*

$$\boldsymbol{f}_\mathbf{x}(\boldsymbol{h}) = \sum_{k=0}^{K-1} h_k S^k \mathbf{x} = H(S)\mathbf{x} , \quad where \quad H(S) \triangleq \sum_{k=0}^{K-1} h_k S^k , \tag{10}$$

*and $\boldsymbol{h} = \{h_0, h_1, \cdots, h_{K-1}\}$ is the set of scalars, also referred to as the filter taps or coefficients.*

Recall from equation 7 that $\tilde{\boldsymbol{\Theta}}(\boldsymbol{h}^{(t)})$ is a function of the Jacobian matrix $\mathbf{J}_{\boldsymbol{f}_{\mathbf{x}_i}}(\boldsymbol{h}^{(t)})$, given by

$$\mathbf{J}_{\boldsymbol{f}_{\mathbf{x}_i}}(\boldsymbol{h}^{(t)}) = \left[\mathbf{x}_i | S\mathbf{x}_i | S^2\mathbf{x}_i | \cdots | S^{K-1}\mathbf{x}_i\right] . \tag{11}$$

Using equation 11, for any pair of input vectors $(\mathbf{x}_i, \mathbf{x}_j)$, the $(i, j)$-the block of the NTK $\tilde{\boldsymbol{\Theta}}(\boldsymbol{h}^{(t)})$ for a graph filter is given by $\Theta_{(\mathbf{x}_i, \mathbf{x}_j)}(\boldsymbol{h}^{(t)}) = \sum_{k=0}^{K-1} S^k \mathbf{x}_i (S^k \mathbf{x}_j)^\mathsf{T}$. Since the graph filter is a linear model, $\Theta_{(\mathbf{x}_i, \mathbf{x}_j)}(\boldsymbol{h}^{(t)})$ is independent of $\boldsymbol{h}^{(t)}$. Consequently, the NTK $\Theta_{(\mathbf{x}_i, \mathbf{x}_j)}(\boldsymbol{h}^{(t)})$ for a graph filter is a constant with respect to $\boldsymbol{h}^{(t)}$. Next, we provide the NTK for a graph filter.

**Proposition 1** (NTK for a graph filter). *The NTK for a graph filter is given by*

$$\tilde{\boldsymbol{\Theta}}_{\mathsf{filt}}(\boldsymbol{h}^{(t)}) = \sum_{k=0}^{K-1} \tilde{S}^k \tilde{\mathbf{x}}\tilde{\mathbf{x}}^\mathsf{T} \tilde{S}^k . \tag{12}$$

*where $\tilde{S} \in \mathbb{R}^{nM \times nM}$ is a block diagonal matrix consisting of $M$ blocks of matrix $S$ on the diagonal and zeros everywhere else.*

Given equation 12, we further investigate the impact of shift operator $S$ on the alignment $\mathcal{A}$. Also we define the data matrices $X, Y$ where $X$ is the input data matrix where the $i$-th column is equal to $\boldsymbol{x}_i$ and similarly for $Y$. From equation 12, note that the NTK is independent of the filter coefficients $\boldsymbol{h}$. As a consequence, $\mathcal{A}$ for a graph filter (denoted by $\mathcal{A}_{\mathsf{filt}}$) depends on the shift operator $S$ and dataset $(X, Y)$ as follows

$$\mathcal{A}_{\mathsf{filt}}(S, X, Y) = \tilde{\mathbf{y}}^\mathsf{T} \left(\sum_{k=0}^{K-1} \tilde{S}^k \tilde{\mathbf{x}}\tilde{\mathbf{x}}^\mathsf{T} \tilde{S}^k\right) \tilde{\mathbf{y}} = \sum_{k=0}^{K-1} \left(\tilde{\mathbf{y}}^\mathsf{T} \tilde{S}^k \tilde{\mathbf{x}}\right)^2 = \sum_{k=0}^{K-1} \left(\mathbf{tr}(Y^\mathsf{T} S^k X)\right)^2 \tag{13}$$

The equivalence between different terms in equation 13 follows from the symmetry of $\tilde{S}$ and the fact that $\tilde{\mathbf{y}}^\mathsf{T} \tilde{S}^k \tilde{\mathbf{x}}$ is a scalar. Since a larger $\mathcal{A}_{\mathsf{filt}}$ is correlated with faster convergence of gradient descent (see equation 9), we further investigate whether the alignment $\mathcal{A}_{\mathsf{filt}}$ can be optimized by appropriate selection of shift operator matrix $S$. The objective to optimize $\mathcal{A}_{\mathsf{filt}}$ can be stated as follows.

$$S^* = \arg\max_S \sum_{k=0}^{K-1} \left(\tilde{\mathbf{y}}^\mathsf{T} \tilde{S}^k \tilde{\mathbf{x}}\right)^2 \quad \textbf{s.t.} \ \eta \cdot ||\tilde{\boldsymbol{\Theta}}_{\mathsf{filt}}||_{\mathsf{op}} < \alpha . \tag{14}$$

The constraint $||\eta \cdot \tilde{\boldsymbol{\Theta}}_{\mathsf{filt}}||_{\mathsf{op}} < \alpha$, for some $\alpha > 0$, in equation 14 is necessary to ensure the convergence of gradient descent. This constraint also eliminates trivial solutions (such as multiplying a given $S$ with an arbitrarily large positive constant to inflate $\mathcal{A}$ in isolation). The optimization problem in equation 14, while meaningful, can be analytically intractable due to complications arising from the polynomial functions of $S$ and the objective function and the constraint being non-convex. In order to provide an analytically tractable solution to $S$, we consider the lower bound on $\mathcal{A}$ next.

**Lemma 1.** *[Lower bound on $\mathcal{A}_{\text{filt}}$.] The alignment $\mathcal{A}_{\text{filt}}$ satisfies $\mathcal{A}_{\text{filt}}(S, X, Y) \geq \mathcal{A}_{\text{L}}(S, X, Y)$, where*

$$\mathcal{A}_{\text{L}}(S, X, Y) \triangleq \left( \frac{1}{\sqrt{K}} \textit{tr}\left( \Big( \sum_{k=0}^{K-1} S^k \Big) C_{XY} \right) \right)^2 , \quad and \quad C_{XY} \triangleq \frac{1}{2}(XY^{\mathsf{T}} + YX^{\mathsf{T}}) . \tag{15}$$

Henceforth, we focus on characterizing $S$ that maximizes $\mathcal{A}_{\text{L}}(S, X, Y)$. Our experiments in Section 4 also demonstrate that the insights drawn from optimizing $\mathcal{A}_{\text{L}}$ are practically meaningful. Next, we provide a constraint that depends on the choice of GSO and not on the input data.

**Lemma 2.** *If the degree $K$ polynomial in the shift operator $S$ has a bounded Frobenius norm, the operator norm of the NTK matrix is also bounded as follows:*

$$\left\| \sum_{k=0}^{K-1} S^k \right\|_F \leq \sqrt{\alpha/(\eta M)} \;\; \Rightarrow \;\; \eta \cdot \left\| \sum_{k=0}^{K-1} \tilde{S}^k \tilde{\mathbf{x}} \tilde{\mathbf{x}}^{\mathsf{T}} \tilde{S}^k \right\|_{\text{op}} \leq \alpha \tag{16}$$

The constraint on the left in equation 16 is more straightforward to work with in the analysis since it only depends on $S$, while also ensuring that the constraint in equation 14 is satisfied. Putting together $\mathcal{A}_{\text{L}}(S, X, Y)$ and the revised constraint, we get the following underline{optimization problem}.

$$S^* = \arg\max_S \mathcal{A}_{\text{L}}(S, X, Y) \;\; \textbf{s.t.} \;\; \left\| \sum_{k=0}^{K-1} S^k \right\|_F \leq \sqrt{\alpha/(\eta M)} . \tag{17}$$

**Theorem 2** (GSO in graph filter.)**.** *A GSO $S^*$ that satisfies*

$$\sum_{k=0}^{K-1} (S^*)^k = \mu \cdot C_{XY} , \quad where \quad \mu = \frac{\sqrt{\alpha/(\eta M)}}{||C_{XY}||_F} . \tag{18}$$

*is the solution to the optimization problem in equation 17.*

Theorem 2 clearly demonstrates the association between the optimal GSO that optimizes $\mathcal{A}_{\text{L}}(S, X, Y)$ and $C_{XY}$, which is a measure of cross-covariance. For instance, if $K = 2$, then it can be concluded from equation 18 that

$$I + S^* = \mu \cdot C_{XY} \Rightarrow S^* = \mu \cdot C_{XY} - I \tag{19}$$

The observation in equation 19 motivates the potential choice of a normalized cross-covariance matrix as a GSO when graph filter is deployed as the predictor $f_{\mathbf{x}}(h)$. Next, we discuss how this observation extends to the setting where $f_{\mathbf{x}}(h)$ is a GNN.

## 3.2 NTK and Alignment for GNN

To start with, we formalize the GNN architecture that is the focus of our analysis. The ability to learn non-linear mappings by GNNs is fundamentally based on concatenating an element-wise non-linearity with a graph filter to form a underline{graph perceptron}, which is realized via a point-wise non-linearity $\sigma(\cdot)$ as $\sigma(H(S)\mathbf{x})$. In the remainder of this paper, we will focus on a two-layer GNN that admits a single input feature $\mathbf{x} \in \mathbb{R}^n$ and the GNN output is a vector of length $n$, as dictated by the problem definition in Section 2. The general definition for a GNN, along with additional experimental results for GNNs with depth larger than two layers, have been provided in Appendix H.

**Two-Layer GNN Architecture** (see Fig. 3). In the first layer, the input vector $\mathbf{x} \in \mathbb{R}^n$, is processed by $F$ graph perceptrons to output $F$ $n$-dimensional outputs given by $\mathbf{q}_{(1)}^f, \forall f \in \{1, \cdots, F\}$, as follows

$$\mathbf{u}_{(1)}^f = H_{(1)}^f(S)\mathbf{x} = \sum_{k=0}^{K-1} h_{(1),k}^f S^k \mathbf{x}, \forall f \in \{1, \ldots, F\}; \quad \text{and} \quad \mathbf{q}_{(1)}^f = \sigma\left( \mathbf{u}_{(1)}^f \right) . \tag{20}$$

In the second layer, each of the outputs of the previous layer, $\mathbf{q}_{(1)}^f$ are processed by a graph filter as

$$\mathbf{u}_{(2)}^f = H_{(2)}^f(S)\mathbf{q}_{(1)}^f = \sum_{k=0}^{K-1} h_{(2),k}^f S^k \mathbf{q}_{(1)}^f, \forall f = \in \{1, \ldots, F\} . \tag{21}$$

Finally, the terms $\mathbf{u}^f_{(2)}$ are aggregated to get the output at the second layer (also the GNN output) as

$$\boldsymbol{f}_{\mathbf{x}}(\boldsymbol{h}) = \frac{1}{\sqrt{F}} \sum_{f=1}^{F} \mathbf{u}^f_{(2)} \tag{22}$$

The absence of a non-linearity in the final layer (equation 22) is necessary for having a constant NTK in the infinite width limit (see Liu et al. (2020)).

**Proposition 2** (NTK for a two-layer GNN). *The NTK for the two-layer GNN is given by*

$$\tilde{\boldsymbol{\Theta}}_{GNN}(\boldsymbol{h}) = \frac{1}{F} \sum_{f=1}^{F} \sum_{k=0}^{K-1} \left(\mathbf{c}^{(1)}_{f,k}\right) \left(\mathbf{c}^{(1)}_{f,k}\right)^{\mathsf{T}} + \frac{1}{F} \sum_{f=1}^{F} \sum_{k=0}^{K-1} \left(\mathbf{c}^{(2)}_{f,k}\right) \left(\mathbf{c}^{(2)}_{f,k}\right)^{\mathsf{T}} \tag{23}$$

*where* $\quad \mathbf{c}^{(1)}_{f,k} \triangleq H_f(\tilde{S}) \cdot \boldsymbol{diag}(\sigma'(G_f(\tilde{S})\tilde{\mathbf{x}})\tilde{S}^k\tilde{\mathbf{x}}), \quad and \quad \mathbf{c}^{(2)}_{f,k} \triangleq \tilde{S}^k \sigma\left(G_f(\tilde{S})\tilde{\mathbf{x}}\right) . \tag{24}$

*In equation 24, $\mathbf{c}^{(\ell)}_{f,k} \in \mathbb{R}^{nM \times 1}$ is the vector determined by picking out the column that pertains to the derivative of the network output with regards to the parameter indexed by $(f, k, \ell)$, namely, the $k$-th coefficient of the $f$-th filter in layer $\ell$, from every Jacobian matrices $J_{\boldsymbol{f}_{\mathbf{x}_i}}, \forall i \in \{1, \cdots M\}$ and stacking all these vectors together.*

The NTK in equation 23 is an aggregation of two terms, where the first term is associated with the first layer and the second term with the second layer. It follows from Definition 1 and equation 23 that the alignment for a two-layer GNN is also composed of two terms that represent the two layers. Henceforth, we focus on the results pertaining to the second term in equation 23. This implies that our results correspond to a two-layer GNN where only the parameters of the second layer are trained and the parameters of the first layer are fixed. The analysis (and results) if we also consider the first layer in this analysis is similar and has been relegated to Appendix G.

In the subsequent discussions, the notation $\tilde{\boldsymbol{\Theta}}_{GNN}$ denotes the second term in equation 23 when the width of hidden layer approaches infinity i.e., $F \to \infty$. Therefore, $\tilde{\boldsymbol{\Theta}}_{GNN}$ is given by

$$\tilde{\boldsymbol{\Theta}}_{GNN}(\boldsymbol{h}) = \lim_{F \to \infty} \frac{1}{F} \sum_{f=1}^{F} \sum_{k=0}^{K-1} \left(\tilde{S}^k \sigma\left(G_f(\tilde{S})\tilde{\mathbf{x}}\right)\right) \left(\tilde{S}^k \sigma\left(G_f(\tilde{S})\tilde{\mathbf{x}}\right)\right)^{\mathsf{T}}$$

$$= \sum_{k=0}^{K-1} \tilde{S}^k \underset{\boldsymbol{g} \sim \mathcal{N}(0,I)}{\mathbb{E}} \left[\sigma\left(G(\tilde{S})\tilde{\mathbf{x}}\right)\left(\sigma\left(G(\tilde{S})\tilde{\mathbf{x}}\right)\right)^{\mathsf{T}}\right] \tilde{S}^k = \sum_{k=0}^{K-1} \tilde{S}^k E \tilde{S}^k \tag{25}$$

The expectation matrix $E \triangleq \underset{\boldsymbol{g} \sim \mathcal{N}(0,I)}{\mathbb{E}} \left[\sigma\left(G(\tilde{S})\tilde{\mathbf{x}}\right)\left(\sigma\left(G(\tilde{S})\tilde{\mathbf{x}}\right)\right)^{\mathsf{T}}\right]$ is instrumental for the analysis of the alignment. Before proceeding, we provide the following remark pertinent to the analysis.

**Remark 2.** *As a byproduct of the output layer being linear, the NTK $\tilde{\boldsymbol{\Theta}}_{GNN}$ in equation 25 does not depend on the parameters of the second layer, i.e., $\boldsymbol{h}_f, \forall f \in \{1, \cdots, F\}$. Hence, the NTK in equation 25 could be considered a constant if only the second layer of GNN is trained. For completeness, our discussion in Appendix F demonstrates further that as $F \to \infty$, the NTK in equation 23 also approaches a constant behavior.*

From equation 25, the alignment can be written in terms of $E$ as follows

$$\mathcal{A} = \tilde{\mathbf{y}}^{\mathsf{T}} \tilde{\boldsymbol{\Theta}}_{GNN} \tilde{\mathbf{y}} = \tilde{\mathbf{y}}^{\mathsf{T}} \left(\sum_{k=0}^{K-1} \tilde{S}^k E \tilde{S}^k\right) \tilde{\mathbf{y}} = \boldsymbol{tr}(QE) , \tag{26}$$

where we have defined the matrix $Q$ as $Q \triangleq \sum_{k=0}^{K-1} \tilde{S}^k \tilde{\mathbf{y}} \tilde{\mathbf{y}}^{\mathsf{T}} \tilde{S}^k$. Above, we used the cyclic property of the trace and the fact that $\tilde{\mathbf{y}}^{\mathsf{T}} \tilde{\boldsymbol{\Theta}}_{GNN} \tilde{\mathbf{y}}$ is a scalar. In order to evaluate $E$, we define the vectors $\boldsymbol{z}^{(\ell)} \in \mathbb{R}^{K \times 1}$ as $\boldsymbol{z}^{(\ell)} \triangleq \left[\tilde{\mathbf{x}}_{\ell}, (\tilde{S}\tilde{\mathbf{x}})_{\ell}, \cdots, (\tilde{S}^{K-1}\tilde{\mathbf{x}})_{\ell}\right]^{\mathsf{T}}$ where $\ell \in \{1, \cdots, nM\}$ and $(\tilde{S}^k\tilde{\mathbf{x}})_{\ell}$ denotes the $\ell$-th entry of the vector $\tilde{S}^k\tilde{\mathbf{x}}$ . Thus, the $(a, b)$-th entry of $E$, i.e., $E_{ab}$, from equation 25 is

$$E_{ab} = \underset{\boldsymbol{g} \sim \mathcal{N}(0,I)}{\mathbb{E}} \left[\sigma\left(\langle \boldsymbol{g}, \boldsymbol{z}^{(a)} \rangle\right) \cdot \sigma\left(\langle \boldsymbol{g}, \boldsymbol{z}^{(b)} \rangle\right)\right] . \tag{27}$$

**Linear GNNs.** We next discuss the scenario when the function $\sigma(\cdot)$ is an identity function, i.e., $\sigma(z) = z$. The results drawn from this setting will be leveraged later in the setting when $\sigma(\cdot)$ is not an identity function. When $\sigma(z) = z$, equation 27 reduces to

$$E_{ab} = \mathop{\mathbb{E}}_{\boldsymbol{g} \sim \mathcal{N}(0, I)} \left[ \langle \boldsymbol{g}, \boldsymbol{z}^{(a)} \rangle \cdot \langle \boldsymbol{g}, \boldsymbol{z}^{(b)} \rangle \right] = \langle \boldsymbol{z}^{(a)}, \boldsymbol{z}^{(b)} \rangle \tag{28}$$

We denote the matrix $E$ in this linear setting by $B_{\mathsf{lin}}$, which is given by

$$(B_{\mathsf{lin}})_{ab} \triangleq \langle \boldsymbol{z}^{(a)}, \boldsymbol{z}^{(b)} \rangle \Rightarrow B_{\mathsf{lin}} = \sum_{k=0}^{K-1} \tilde{S}^k \tilde{\mathbf{x}} \tilde{\mathbf{x}}^{\mathsf{T}} \tilde{S}^k \tag{29}$$

Thus, the alignment in the linear setting is given by

$$\mathcal{A}_{\mathsf{lin}} \triangleq \mathbf{tr}(Q B_{\mathsf{lin}}) = \sum_{k=0}^{K-1} \tilde{\mathbf{y}}^T \tilde{S}^k B \tilde{S}^k \tilde{\mathbf{y}} = \sum_{k=0}^{K-1} \sum_{k'=0}^{K-1} \tilde{\mathbf{y}}^T \tilde{S}^{k+k'} \tilde{\mathbf{x}} \tilde{\mathbf{x}}^T \tilde{S}^{k+k'} \tilde{\mathbf{y}} \tag{30}$$

The analysis of alignment $\mathcal{A}_{\mathsf{lin}}$ in equation 30 using similar arguments as that for a graph filter in Section 3.1 yields a similar condition on the GSO $S$ as in Theorem 2. The corollaries provided next formalize this observation. First, the following corollary provides a lower bound on $\mathcal{A}_{\mathsf{lin}}$.

**Corollary 1.** *[Lower bound on $\mathcal{A}_{\mathsf{lin}}$] The term $\mathcal{A}_{\mathsf{lin}} = \boldsymbol{tr}(Q B_{\mathsf{lin}})$ satisfies $\mathcal{A}_{\mathsf{lin}} \geq \mathcal{A}_{\mathsf{L}'}(S, X, Y)$,*

$$where \quad \mathcal{A}_{\mathsf{L}'}(S, X, Y) \triangleq \left( \frac{1}{\sqrt{K}} \boldsymbol{tr}\left( \left( \sum_{k=0}^{K-1} \sum_{k'=0}^{K-1} (S^*)^{k+k'} \right) C_{XY} \right) \right)^2, \tag{31}$$

Next, we present an optimization problem similar to the one for the graph filter in equation 17 next.

$$S^* = \arg \max_{S} \mathcal{A}_{\mathsf{L}'}(S, X, Y) \ \textbf{s.t.} \ \left\| \sum_{k=0}^{K-1} \sum_{k'=0}^{K-1} S^{k+k'} \right\|_F \leq \sqrt{\alpha/(\eta M)}. \tag{32}$$

The solution to the optimization problem in equation 32 is presented next.

**Corollary 2.** *[Extension of Theorem 1 to linear GNN] The GSO $S^*$ that solves the optimization problem in equation 32 must satisfy*

$$\sum_{k=0}^{K-1} \sum_{k=0}^{K-1} (S^*)^{k+k'} = \mu \cdot C_{XY}, \quad where \quad \mu = \frac{\sqrt{\alpha/(\eta M)}}{\|C_{XY}\|_F}. \tag{33}$$

Corollary 2 establishes that the cross-covariance $C_{XY}$ is instrumental to optimizing $\mathcal{A}_{\mathsf{L}'}$ for the considered two-layer GNN architecture when $\sigma(\cdot)$ is an identity function. In general, this observation holds for linear GNNs of any arbitrary depth.

**GNNs with non-linear activation function.** Next, we investigate the conditions under which the observation in Corollary 2 extends to a more general setting, in which $\sigma(\cdot)$ is not an identity function. We will focus our theoretical analysis on the case where $\sigma(z) = \tanh(z)$ and from here on $\mathcal{A}$ will denote the alignment for this case. The experimental results (see Appendix H) show that similar results hold for some other activation functions like ReLU in practice. First, we evaluate the expectation in equation 25. By leveraging the theory of Hermite polynomials [1], the Hermite expansions of $\sigma\left(\langle \boldsymbol{g}, \boldsymbol{z}^{(a)} \rangle\right)$ and $\sigma\left(\langle \boldsymbol{g}, \boldsymbol{z}^{(b)} \rangle\right)$ enables the expansion of $E$ and subsequently $\mathcal{A}$. These expansions are formalized next.

**Lemma 3** (Expansion of $E$ and $\mathcal{A}$)**.** *The Hermite expansion of $E$ is given by $E = B + \Delta B$, where $B \in \mathbb{R}^{nM \times nM}$ represents the first non-zero term in the expansion and $\Delta B \in \mathbb{R}^{nM \times nM}$ includes all the subsequent terms. For the $(a, b)$-th element of $B$ and $\Delta B$, we have*

$$B_{ab} = \alpha_1 \beta_1 \cdot \frac{\langle \boldsymbol{z}^{(a)}, \boldsymbol{z}^{(b)} \rangle}{\|\boldsymbol{z}^{(a)}\|_2 \cdot \|\boldsymbol{z}^{(b)}\|_2}, \ and \ (\Delta B)_{ab} = \sum_{i=3,5,\cdots}^{\infty} \alpha_i \beta_i \cdot \left( \frac{\langle \boldsymbol{z}^{(a)}, \boldsymbol{z}^{(b)} \rangle}{\|\boldsymbol{z}^{(a)}\|_2 \cdot \|\boldsymbol{z}^{(b)}\|_2} \right)^i. \tag{34}$$

*Hence, the alignment $\mathcal{A}$ in equation 26 admits the expansion*

$$\mathcal{A} = \boldsymbol{tr}(QE) = \boldsymbol{tr}(QB) + \boldsymbol{tr}(Q\Delta B). \tag{35}$$

---

[1] See the proof of Lemma 3 for an overview of the Hermite polynomials and how we utilized the Hermite expansion.

The scalar coefficients $\alpha_i, \beta_i$ in equation 34 depend on $||\boldsymbol{z}^{(a)}||_2$ and $||\boldsymbol{z}^{(b)}||_2$, respectively and the choice of $\sigma(\cdot)$. The disintegration of the alignment into two terms in equation 35 is useful because the second term can be shown to be relatively small and, using the observation that in the linear case $E_{ab} = \langle \boldsymbol{z}^{(a)}, \boldsymbol{z}^{(b)} \rangle$, $\mathbf{tr}(QB)$ can be related to $\mathcal{A}_{\text{lin}}$. We next provide two lemmas relevant to this.

**Lemma 4.** *Given a family of matrices $S \in \mathbb{S}^{n \times n}$ that have a bounded norm, $||S||_{\text{op}} \leq \nu$, we have*

$$\boldsymbol{tr}(QB) \geq \rho \mathcal{A}_{\text{lin}} \tag{36}$$

*where $\rho$ is a constant that depends on the choice of non-linearity function $\sigma(\cdot)$.*

The next lemma shows that the elements of the matrix $\Delta B$ are smaller compared to corresponding elements in $B$, which implies that the second term in equation 35 can't decrease $\mathcal{A}$ too much.

**Lemma 5.** *Each element of $\Delta B$ has the same sign as the corresponding element in $B$. Also, the following element-wise inequality holds between the two matrices:*

$$|\Delta B| \leq \beta \cdot |B| \tag{37}$$

*where $\beta$ is a constant that depends on our choice of non-linearity and is determined from the proof.*

Putting the above two lemmas together, we reach the following conclusion about the alignment of the two-layer GNN with $\sigma(\cdot)$ as $\tanh$.

**Theorem 3.** *Given a family of matrices $S \in \mathbb{S}^{n \times n}$ that have a bounded norm, $||S||_{\text{op}} \leq \nu$ and that satisfy $\mathcal{A}_{\text{lin}} = \boldsymbol{tr}(QB_{\text{lin}}) \geq \xi \cdot ||Q||_F ||B_{\text{lin}}||_F$ for some constant $0 < \xi \leq 1$, $\mathcal{A}_{\text{lin}}$ lower bounds the alignment for the two-layer GNN with $\tanh$ non-linearity, $\mathcal{A}$, up to a constant as follows*

$$\mathcal{A} \geq \left( c - \frac{d}{\xi} \right) \mathcal{A}_{\text{lin}} , \tag{38}$$

*for some positive constants $c$ and $d$.*

**Remark 3** (GSO in GNNs). *Our key takeaway from Theorem 3 is that when maximizing $\mathcal{A}_{\text{lin}}$ over the family of shift operators that satisfy the assumption with a sufficiently large $\xi$, such that $c - d/\xi$ is a positive constant, we are essentially maximizing a lower bound on the alignment $\mathcal{A}$ of the two-layer GNN. This observation, together with Corollary 3, motivates using the cross-covariance matrix $C_{XY}$ as a GSO for the two-layer GNN.*

**Alignment, The NTK and Generalization** Thus far, we have provided the theoretical results motivated by the fact that larger alignment can imply faster convergence of gradient descent during training. However, the alignment and NTK are also closely related to generalization. Specifically, the analyses pertaining to generalization from Arora et al. (2019) and Wang et al. (2022a) can be extended to the case of graph filters, which leads to the conclusion that larger alignment could also lead to smaller generalization error. Hence, the results on improved training and generalization together motivate models with larger alignment in practice. The analysis regarding generalization has been provided in Appendix E.

## 4 EXPERIMENTS

In this section, we provide the experiments that validate the theoretical insights pertaining to the cross-covariance matrix being an optimal GSO for GNN training and generalization with respect to GSO derived only from the input data for a regression task. The dataset and inference task for this purpose are described below.

**Data.** The HCP-YA dataset is a publicly available brain imaging dataset collected over a population of 1003 healthy adults in the age range of 22–35 years Van Essen et al. (2012; 2013). In our experiments, we leveraged the rfMRI data for each subject made available by HCP. This data consisted of a multi-variate time series of 100 features, with each time series consisting of 4500 time points.

**Inference task.** Noting that the 100 features could be considered as 100 nodes of a graph, our objective was to use the data at all nodes at the current time step for an individual to predict the data at all nodes at a future time step. Specifically, given the signal value at time step $t$ as $\boldsymbol{z}^{(t)} \in \mathbb{R}^{100}$ for an individual, we aimed to predict the signal value after $\Delta t$ time steps, i.e., $\boldsymbol{z}^{(t+\Delta t)} \in \mathbb{R}^{100}$.

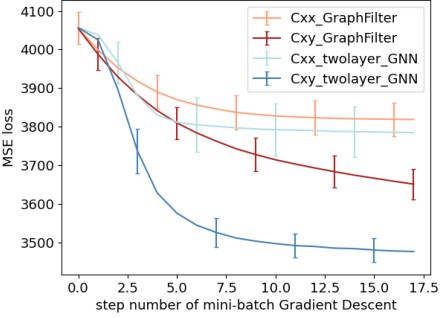

(a) Training loss for one individual in HCP-YA dataset.

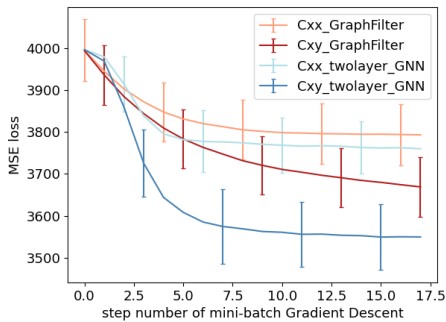

(b) Test loss for the same individual as in (a).

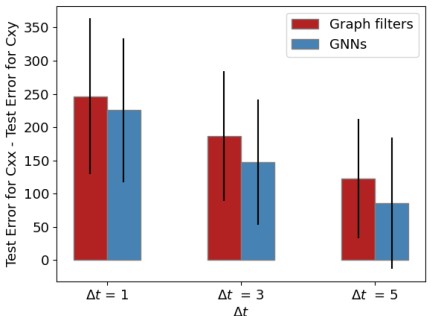

(c) **Generalization.** The gap between the final test error for $C_{XY}$ and $C_{XX}$ for different predictors averaged over the complete dataset of individuals.

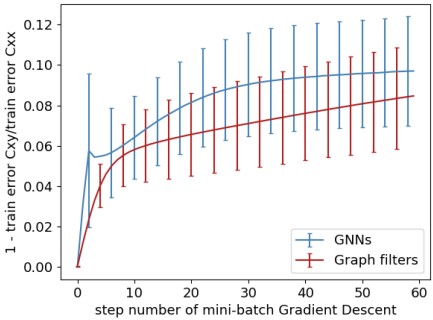

(d) Improvement in training error at different epochs of gradient descent averaged over the complete dataset of individuals.

Figure 1: Experimental Results

For every $\Delta t \in \{1, 2, 3, 4, 5\}$, a separate training/test set of size $N_{train} = 1000$, $N_{test} = 100$ was created, such that, for the signal at a time point $t$, i.e., $\boldsymbol{z}^{(t)}$ as the input, the signal after $\Delta t$ time steps $\boldsymbol{z}^{(t+\Delta t)}$ was the output to be predicted. For additional implementation details, see Appendix H.

**Performance evaluation.** We trained two sets of GNNs and two sets of graph filters using the time series data of each individual for a given $\Delta t$, where one set comprised of predictors with $C_{XY}$ as the GSO and the other with $C_{XX}$ as the GSO. The GNNs with $C_{XX}$ as the GSO have been studied before as VNNs in Sihag et al. (2022) and provide an appropriate baseline for comparison as it is representative of GNNs with GSOs extracted only from the input data. Figures 8a and 8b illustrate faster convergence of both training loss and test loss during gradient descent for predictors with $C_{XY}$ as compared to those with $C_{XX}$ for one representative individual when $\Delta t = 1$. This observation was consistent for graph filters and GNNs. For each individual, the training process for every architecture was repeated 10 times. The average of these runs is shown in Figures 8a and 8b.

Further, we checked whether these observations were consistent across the dataset and for different $\Delta t$. Figure 1c illustrates the gap between the test error for predictors with $C_{XY}$ and $C_{XX}$ and different values of $\Delta t$, averaged across all individuals. Even as the accuracy of prediction diminished with increasing $\Delta t$, we observed a consistent gain in test performance when using $C_{XY}$ as compared to $C_{XX}$. Similarly, Fig. 1d shows that predictors with $C_{XY}$ achieved smaller training error relative to those with $C_{XX}$ at each epoch of gradient descent, averaged across the dataset. Thus, the observations in Fig. 1 validated the theoretical insights drawn from the analysis that argued for $C_{XY}$ as an appropriate GSO for GNNs that can achieve smaller training error and better generalization. We refer the reader to Appendix C for the conclusions, limitations, and potential future directions of our work.

## 5 REPRODUCIBILITY STATEMENT

**Theoretical Proofs.** The proofs for all theorems and lemmas in the main body of the paper can be found in Appendix D.

**Experiments.** Additional experimental details and results have been provided in Appendix H. In addition, the code and data for training the relevant models and producing results similar to what is shown in Figures 8a and 8b for one individual from the HCP-YA dataset can be found in https://github.com/shervinkh2000/Cross_Covariance_NTK. The complete HCP-YA dataset is publicly available and accessible from https://db.humanconnectome.org/ as per the data use terms.

**Additional Details.** Comments made regarding the relation between the NTK and generalization made in the main body of the paper have been thoroughly explained in Appendix E. Additional necessary theoretical considerations that did not directly contribute to the main message of the paper, namely, that the NTK of the GNN architecture that we analyzed is constant in the infinite-width limit, and that training the first layer of the GNN leads to similar results to those we saw for the second layer in section 3.2, have been discussed further in Appendices F and G respectively.

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

## 6   APPENDICES

## A   ACKNOWLEDGEMENTS

Data used in the experiments were provided [in part] by the Human Connectome Project, WU-Minn Consortium (Principal Investigators: David Van Essen and Kamil Ugurbil; 1U54MH091657) funded by the 16 NIH Institutes and Centers that support the NIH Blueprint for Neuroscience Research; and by the McDonnell Center for Systems Neuroscience at Washington University.

## B   ADDITIONAL LITERATURE REVIEW

**Analysis of GNNs with NTKs.** Du et al. (2019) prove that the NTK for GNNs can learn a broad class of functions on graphs. Sabanayagam et al. (2021) use the GNTK as a hyperparameter-free surrogate for GNNs to empirically analyze the effect of increasing depth and using skip connections on the performance of GNNs. They also show that the GNTK captures the performance trend of the corresponding finite-width GNNs. Krishnagopal & Ruiz (2023) analyze how the graph size affects the GNTK and shows that as the graph gets larger the GNTK converges to the graphon NTK. Sabanayagam et al. (2022) use the GNTK to theoretically (and empirically) explain the effects of different architecture choices such as symmetric vs row normalization and increasing depth on GNN performance.

**Empirical implications of theoretical insights derived using NTKs.** Existing studies have used the theoretical insights drawn from NTKs to understand practical observations and inform practical applications. The studies in Chen et al. (2021) and Zhu et al. (2022) leverage NTKs to study theory-inspired neural architecture search protocols. Insights into convergence and generalization properties of neural networks derived using NTKs are well known Jacot et al. (2018). The training dynamics of physics inspired neural networks were studied using NTKs in Wang et al. (2022b) and an NTK-inspired gradient descent algorithm was proposed. Huang et al. (2022) utilize insights gained from NTK-based analysis of the optimization of GNNs to explain the degrading performance of GNNs as the number of layers increases. They further suggest a potential solution to this problem through randomly dropping a certain percentage of the edges of the graph during training. Our work is similar in spirit to such studies, where we draw upon the theoretical insights derived from NTKs in the context of GNNs to motivate the choice of GSO for an inference task.

## C   DISCUSSION

In this paper, we have demonstrated that the analysis of NTKs in the context of GNNs motivates cross-covariance graph as the GSO. Specifically, we have shown that for a two-layer GNN, choosing the cross-covariance matrix between the input and output data as the GSO maximizes lower bounds on the alignment (a function of correlation between the NTK and the available dataset) and this alignment, in turn, governs the convergence rate and generalization properties of the predictor. We have validated that the GNNs with cross-covariance graphs indeed outperform GNNs with covariance graphs (that are representative of GNNs with graphs obtained only using the input data) in a time-series prediction task on the HCP-YA dataset. The cross-covariance based GNNs exhibit faster convergence and smaller training and test errors and these empirical observations even extend to GNNs deeper than the two-layer one that we have theoretically analyzed.

A main limitation of this work is the restricted focus on a dataset with input and output vectors of the same dimensionalities. Such a setting lends itself well to using GNN architectures and makes their theoretical analysis tractable. However, in general, the inputs and outputs for multi-variate regression problems can often have different dimensionalities. Deriving similar results for such cases is a potential avenue for future work. Another limitation of our theoretical contributions is the lack of thorough analysis of the tightness of the lower bounds analyzed.

Cross-covariance graphs in GNNs for regression models could be tied in principle with the traditional statistical approaches for multi-variate regression approaches, such as partial least squares (PLS) regression Höskuldsson (1988). Specifically, PLS regression relies on finding a hyperplane that maximizes the cross-covariance between the latent spaces of input and output data for a regres-

sion problem. While our work does not rely on any artificial dimensionality reduction, equivalence between information processing using GNNs over covariance matrices and underlying principal components have been demonstrated previously Sihag et al. (2022). In this context, establishing the foundational analyses of GNNs that operate on cross-covariance graphs is of immediate interest.

# D  ADDITIONAL PROOFS

## D.1  PROOF OF THEOREM 1

*Proof.* Gradient descent is deployed to minimize the following cost

$$\Phi(\boldsymbol{h}) \triangleq \min_{\boldsymbol{h} \in \mathbb{R}^p} \frac{1}{2} \sum_{i=1}^{M} ||\mathbf{y}_i - \boldsymbol{f}_{\mathbf{x}_i}(\boldsymbol{h})||_2^2 \ . \tag{39}$$

The change in predictor output at the $t$-th step of gradient descent can be written as

$$\boldsymbol{f}_{\mathbf{x}_i}(\boldsymbol{h}^{(t+1)}) = \boldsymbol{f}_{\mathbf{x}_i}(\boldsymbol{h}^{(t)} - \eta \cdot \nabla \Phi(\boldsymbol{h}^{(t)})) \tag{40}$$

Assuming $\eta$ to be sufficiently small we can linearize $\boldsymbol{f}_{\mathbf{x}_i}$ near the point $\boldsymbol{h}^{(t)}$:

$$\boldsymbol{f}_{\mathbf{x}_i}(\boldsymbol{h}^{(t)} - \eta \cdot \nabla \Phi(\boldsymbol{h}^{(t)})) = \boldsymbol{f}_{\mathbf{x}_i}(\boldsymbol{h}^{(t)}) - \eta \cdot \mathsf{J}_{\boldsymbol{f}_{\mathbf{x}_i}}(\boldsymbol{h}^{(t)}) \cdot \nabla \Phi(\boldsymbol{h}^{(t)}) \ , \tag{41}$$

where $\mathsf{J}_{\boldsymbol{f}_{\mathbf{x}_i}}(\boldsymbol{h}^{(t)})$ is the Jacobian matrix of the vector-valued function $\boldsymbol{f}_{\mathbf{x}_i}$, evaluated at point $\boldsymbol{h}^{(t)}$. We can also write the gradient of the loss function $\Phi(\boldsymbol{h})$ in terms of the Jacobians $\mathsf{J}_{\boldsymbol{f}_{\mathbf{x}_j}}$ as follows:

$$\nabla \Phi(\boldsymbol{h}^{(t)}) = \sum_{j=1}^{M} \left(\mathsf{J}_{\boldsymbol{f}_{\mathbf{x}_j}}(\boldsymbol{h}^{(t)})\right)^{\mathsf{T}} \cdot (\boldsymbol{f}_{\mathbf{x}_j}(\boldsymbol{h}^{(t)}) - \mathbf{y}_j) \tag{42}$$

Putting equation 40 and equation 41 and equation 42 together we get:

$$\begin{aligned} \boldsymbol{f}_{\mathbf{x}_i}(\boldsymbol{h}^{(t+1)}) &= \boldsymbol{f}_{\mathbf{x}_i}(\boldsymbol{h}^{(t)}) - \eta \cdot \sum_{j=1}^{M} \mathsf{J}_{\boldsymbol{f}_{\mathbf{x}_i}}(\boldsymbol{h}^{(t)}) \left(\mathsf{J}_{\boldsymbol{f}_{\mathbf{x}_j}}(\boldsymbol{h}^{(t)})\right)^{\mathsf{T}} \cdot (\boldsymbol{f}_{\mathbf{x}_j}(\boldsymbol{h}^{(t)}) - \mathbf{y}_j) \\ &= \boldsymbol{f}_{\mathbf{x}_i}(\boldsymbol{h}^{(t)}) - \eta \cdot \sum_{j=1}^{M} \Theta(\mathbf{x}_i, \mathbf{x}_j) \cdot (\boldsymbol{f}_{\mathbf{x}_j}(\boldsymbol{h}^{(t)}) - \mathbf{y}_j) \end{aligned} \tag{43}$$

where the $n \times n$ matrix $\Theta(\mathbf{x}_i, \mathbf{x}_j)$ is defined as the product of two Jacobian matrices (which corresponds to the inner product of two gradient vectors that appears in the scalar output case; (see equation 1):

$$\Theta(\mathbf{x}_i, \mathbf{x}_j) = \mathsf{J}_{\boldsymbol{f}_{\mathbf{x}_i}}(\boldsymbol{h}^{(t)}) \left(\mathsf{J}_{\boldsymbol{f}_{\mathbf{x}_j}}(\boldsymbol{h}^{(t)})\right)^{\mathsf{T}} \tag{44}$$

Further, equation 43 can be re-written in the vectorized form as follows

$$\tilde{\boldsymbol{f}}_X(\boldsymbol{h}^{(t+1)}) = \tilde{\boldsymbol{f}}_X(\boldsymbol{h}^{(t)}) - \eta \cdot \tilde{\boldsymbol{\Theta}}(\boldsymbol{h}^{(t)}) \cdot (\tilde{\boldsymbol{f}}_X(\boldsymbol{h}^{(t)}) - \tilde{\mathbf{y}}) \tag{45}$$

If the NTK matrix $\tilde{\boldsymbol{\Theta}}(\boldsymbol{h}^{(t)})$ is constant or non-evolving with respect to epoch $t$, we can use equation 45 to analyze the evolution of gradient descent as follows:

$$\begin{aligned} \tilde{\boldsymbol{f}}_X(\boldsymbol{h}^{(t+1)}) - \tilde{\mathbf{y}} &= \tilde{\boldsymbol{f}}_X(\boldsymbol{h}^{(t)}) - \tilde{\mathbf{y}} - \eta \cdot \tilde{\boldsymbol{\Theta}} \cdot (\tilde{\boldsymbol{f}}_X(\boldsymbol{h}^{(t)}) - \tilde{\mathbf{y}}) \\ &= (I - \eta \cdot \tilde{\boldsymbol{\Theta}})(\tilde{\boldsymbol{f}}_X(\boldsymbol{h}^{(t)}) - \tilde{\mathbf{y}}) \\ &= (I - \eta \cdot \tilde{\boldsymbol{\Theta}})^{t+1}(\tilde{\boldsymbol{f}}_X(\boldsymbol{h}^{(0)}) - \tilde{\mathbf{y}}) \\ &= -(I - \eta \cdot \tilde{\boldsymbol{\Theta}})^{t+1}\tilde{\mathbf{y}} + (I - \eta \cdot \tilde{\boldsymbol{\Theta}})^{t+1}\tilde{\boldsymbol{f}}_X(\boldsymbol{h}^{(0)}) \end{aligned} \tag{46}$$

where we have used the notation $\tilde{\boldsymbol{\Theta}}$ to denote an NTK matrix with constant behavior. By choosing the initialization to be small i.e., choosing a small $\kappa$, we can ensure that $\tilde{\boldsymbol{f}}_X(\boldsymbol{h}^{(0)})$ is sufficiently close to 0. Recall that the parameters are initialized randomly as

$$\boldsymbol{h}^{(0)} \sim \mathcal{N}(\mathbf{0}, \kappa^2 I). \tag{47}$$

We can thus say that the output at initialization is 0 in expectation and that the variance of each entry of the output vector is also proportional to $\kappa^2$:

$$\mathbb{E}\left[\tilde{\boldsymbol{f}}_X(\boldsymbol{h}^{(0)})\right] = \mathbf{0}, \ \mathbb{E}\left[\left(\tilde{\boldsymbol{f}}_X(\boldsymbol{h}^{(0)})\right)_\ell^2\right] = \mathcal{O}(\kappa^2) \tag{48}$$

where $\left(\tilde{\boldsymbol{f}}_X(\boldsymbol{h}^{(0)})\right)_\ell$ denotes the $\ell$-th entry of the vector $\tilde{\boldsymbol{f}}_X(\boldsymbol{h}^{(0)})$. Since $\tilde{\boldsymbol{f}}_X(\boldsymbol{h}^{(0)})$ is a vector of $nM$ independently initialized entries we have $\mathbb{E}\left[||\tilde{\boldsymbol{f}}_X(\boldsymbol{h}^{(0)})||_2^2\right] = \mathcal{O}(nM\kappa^2)$. Therefore, using Markov's inequality, we obtain

$$\mathbb{P}\left(||\tilde{\boldsymbol{f}}_X(\boldsymbol{h}^{(0)})||_2^2 \geq \frac{nM\kappa^2}{\delta}\right) \leq \delta \tag{49}$$

If we choose $\kappa = \mathcal{O}(\varepsilon\sqrt{\frac{\delta}{nM}})$ we have with probability at least $1 - \delta$ that $||\tilde{\boldsymbol{f}}_X(\boldsymbol{h}^{(0)})||_2 < \varepsilon$ which leads to the following:

$$||(I - \eta \cdot \tilde{\boldsymbol{\Theta}})^{t+1}\tilde{\boldsymbol{f}}_X(\boldsymbol{h}^{(0)})||_2 \leq ||(I - \eta \cdot \tilde{\boldsymbol{\Theta}})^{t+1}||_{\mathsf{op}}||\tilde{\boldsymbol{f}}_X(\boldsymbol{h}^{(0)})||_2 \leq \underbrace{(1 - \eta\lambda_{min})^{t+1}}_{\mathcal{O}(1)}\underbrace{||\tilde{\boldsymbol{f}}_X(\boldsymbol{h}^{(0)})||_2}_{\mathcal{O}(\varepsilon)}$$
$$\tag{50}$$

Therefore $||(I - \eta \cdot \tilde{\boldsymbol{\Theta}})^{t+1}\tilde{\boldsymbol{f}}_X(\boldsymbol{h}^{(0)})||_2$ has $\mathcal{O}(\varepsilon)$ behavior. Then from equation 46 we can write:

$$||\tilde{\boldsymbol{f}}_X(\boldsymbol{h}^{(t+1)}) - \tilde{\mathbf{y}}||_2 = ||(I - \eta \cdot \tilde{\boldsymbol{\Theta}})^{t+1}\tilde{\mathbf{y}}||_2 \pm \mathcal{O}(\varepsilon) \tag{51}$$

Since $\varepsilon$ can be chosen to be arbitrarily small, we subsequently focus only on the term $(I - \eta \cdot \tilde{\boldsymbol{\Theta}})^{t+1}\tilde{\mathbf{y}}$. From equation 46 we have

$$\tilde{\boldsymbol{f}}_X(\boldsymbol{h}^{(t)}) - \tilde{\mathbf{y}} = -(I - \eta \cdot \tilde{\boldsymbol{\Theta}})^t \cdot \tilde{\mathbf{y}} \pm \mathcal{O}\left(\varepsilon\sqrt{\frac{\delta}{nM}}\right) \tag{52}$$

Because $\tilde{\boldsymbol{\Theta}}$ is symmetric with real valued entries, its eigen-decomposition, when $\mathsf{rank}(\tilde{\boldsymbol{\Theta}}) = r$, is given by

$$\tilde{\boldsymbol{\Theta}} = \sum_{\ell=1}^{r}\lambda_\ell \boldsymbol{v}_\ell \boldsymbol{v}_\ell^\mathsf{T} \tag{53}$$

equation 53 implies the following eigen-decomposition for $(I - \eta \cdot \tilde{\boldsymbol{\Theta}})$:

$$I - \eta \cdot \tilde{\boldsymbol{\Theta}} = \sum_{\ell=1}^{r}(1 - \eta\lambda_\ell)\boldsymbol{v}_\ell \boldsymbol{v}_\ell^\mathsf{T} + \sum_{\ell=r+1}^{nM} 1 \cdot \boldsymbol{v}_\ell \boldsymbol{v}_\ell^\mathsf{T} \tag{54}$$

Using equation 54 we rewrite equation 52 as

$$\begin{aligned}\tilde{\boldsymbol{f}}_X(\boldsymbol{h}^{(t)}) - \tilde{\mathbf{y}} &= -\left(\sum_{\ell=1}^{r}(1 - \eta\lambda_\ell)\boldsymbol{v}_\ell \boldsymbol{v}_\ell^\mathsf{T} + \sum_{\ell=r+1}^{nM} 1 \cdot \boldsymbol{v}_\ell \boldsymbol{v}_\ell^\mathsf{T}\right)^t \tilde{\mathbf{y}} \pm \mathcal{O}\left(\varepsilon\sqrt{\frac{\delta}{nM}}\right) \\ &= -\sum_{\ell=1}^{r}((1 - \eta\lambda_\ell)^\mathsf{T}\boldsymbol{v}_\ell^\mathsf{T}\tilde{\mathbf{y}})\,\boldsymbol{v}_\ell - \sum_{\ell=r+1}^{nM}(\boldsymbol{v}_\ell^\mathsf{T}\tilde{\mathbf{y}})\,\boldsymbol{v}_\ell \pm \mathcal{O}\left(\varepsilon\sqrt{\frac{\delta}{nM}}\right)\end{aligned} \tag{55}$$

Using the fact that the eigenvectors form an orthonormal basis we can write the training loss after $t$ steps of Gradient Descent as

$$||\tilde{\boldsymbol{f}}_X(\boldsymbol{h}^{(t)}) - \tilde{\mathbf{y}}||_2^2 = \sum_{\ell=1}^{r}(1 - \eta\lambda_\ell)^{2t}(\boldsymbol{v}_\ell^\mathsf{T}\tilde{\mathbf{y}})^2 + \sum_{\ell=r+1}^{nM}(\boldsymbol{v}_\ell^\mathsf{T}\tilde{\mathbf{y}})^2 \pm \mathcal{O}(\varepsilon) \tag{56}$$

In equation 56 we can see how the non-zero eigenvalues of the NTK and the corresponding eigenvectors characterize the training process:

1. Of the two sums in equation 56, $\sum_{\ell=r+1}^{nM}(v_\ell^\mathsf{T}\tilde{y})^2$ remains constant during training. Therefore it is the hard limit for the minimum achievable training error and cannot be optimized.

2. Each summand in $\sum_{\ell=1}^{r}(1-\eta\lambda_\ell)^{2t}(v_\ell^\mathsf{T}\tilde{y})^2$, converges linearly to zero. The rate of convergence is determined by $(1-\eta\lambda_\ell)^2$ i.e., the larger $\lambda_\ell$ is, the faster the convergence.

3. From the previous two points we can surmise that if $\tilde{y}$ is well-aligned with the eigenvectors of the NTK that correspond to its larger eigenvalues i.e., $(v_\ell^\mathsf{T}\tilde{y})^2$ is large whenever $\lambda_\ell$ is large, we'll have faster convergence.

4. We know that since the eigenvectors form a basis, $\sum_{\ell=1}^{nM}(v_\ell^\mathsf{T}\tilde{y})^2 = ||\tilde{y}||_2^2$ which is constant. Therefore the more of $\tilde{y}$ that is aligned with the eigenvectors of the NTK with non-zero eigenvalues, the smaller the final training error ( after a large enough number of steps) will be.

Next, we combine the above insights to derive the result in Theorem 1. First, for simplicity, we write equation 56 as

$$||\tilde{f}_X(h^{(t)}) - \tilde{y}||_2^2 = \sum_{\ell=1}^{nM}(1-\eta\lambda_\ell)^{2t}(v_\ell^\mathsf{T}\tilde{y})^2 \pm \mathcal{O}(\varepsilon) \tag{57}$$

keeping in mind that that only the first $r$ eigenvalues are non-zero. Then we have:

$$\begin{aligned}
\sum_{\ell=1}^{nM}(1-\eta\lambda_\ell)^{2t}(v_\ell^\mathsf{T}\tilde{y})^2 &\leq \sum_{\ell=1}^{nM}(1-\eta\lambda_\ell)(v_\ell^\mathsf{T}\tilde{y})^2 \\
&= \tilde{y}^\mathsf{T}\left(\sum_{\ell=1}^{nM}(1-\eta\lambda_\ell)v_\ell v_\ell^\mathsf{T}\right)\tilde{y} \\
&= \tilde{y}^\mathsf{T}\left(I - \eta\cdot\tilde{\Theta}\right)\tilde{y}
\end{aligned} \tag{58}$$

Equation 58 gives the desired upper bound. Now we move on to the lower bound for equation 57. Using Bernoulli's inequality which states $(1+x)^m \geq 1+mx$ for every integer $m \geq 1$ and real number $x > -1$ we can write:

$$\begin{aligned}
\sum_{\ell=1}^{nM}(1-\eta\lambda_\ell)^{2t}(v_\ell^\mathsf{T}\tilde{y})^2 &\geq \sum_{\ell=1}^{nM}(1-2t\eta\lambda_\ell)(v_\ell^\mathsf{T}\tilde{y})^2 \\
&= \tilde{y}^\mathsf{T}\left(\sum_{\ell=1}^{nM}(1-2t\eta\lambda_\ell)v_\ell v_\ell^\mathsf{T}\right)\tilde{y} \\
&= \tilde{y}^\mathsf{T}\left(I - 2t\eta\cdot\tilde{\Theta}\right)\tilde{y}
\end{aligned} \tag{59}$$

Note that in order for gradient descent to converge, $\eta$ must be small enough so that for all $\ell$, $\eta\lambda_\ell \leq 1$. This leads to the condition for Bernoulli's inequality to also be satisfied i.e., $-\eta\lambda_\ell \geq -1$. Putting together the upper bound from equation 58 and the lower bound from equation 59 we can bound the quantity in equation 57 from both sides:

$$\tilde{y}^\mathsf{T}\left(I - 2t\eta\cdot\tilde{\Theta}\right)\tilde{y} \pm \mathcal{O}(\varepsilon) \leq \sum_{\ell=1}^{nM}(1-\eta\lambda_\ell)^{2t}(v_\ell^\mathsf{T}\tilde{y})^2 \leq \tilde{y}^\mathsf{T}\left(I - \eta\cdot\tilde{\Theta}\right)\tilde{y} \pm \mathcal{O}(\varepsilon) \tag{60}$$

which concludes the proof. □

## D.2 Proof of Lemma 1

*Proof.*

$$
\begin{aligned}
\mathcal{A}_{\text{filt}}(S, X, Y) &= \sum_{k=0}^{K-1} \left( \tilde{\mathbf{y}}^{\mathsf{T}} \tilde{S}^k \tilde{\mathbf{x}} \right)^2 \\
&= \sum_{k=0}^{K-1} \left( \sum_{i=1}^{M} \mathbf{y}_i^{\mathsf{T}} S^k \mathbf{x}_i \right)^2 \\
&= \sum_{k=0}^{K-1} \left( \mathbf{tr}(Y^{\mathsf{T}} S^k X) \right)^2
\end{aligned} \tag{61}
$$

Using the cyclic property of the trace (and symmetry of $S^k$) we can write:

$$
\mathbf{tr}(Y^{\mathsf{T}} S^k X) = \mathbf{tr}(S^k XY^{\mathsf{T}}) = \mathbf{tr}(XY^{\mathsf{T}} S^k) = \mathbf{tr}(S^k Y X^{\mathsf{T}}) \tag{62}
$$

Using the above and equation 61, we have:

$$
\mathcal{A}_{\text{filt}}(S, X, Y) = \sum_{k=0}^{K-1} \left( \frac{1}{2} \left( \mathbf{tr}(S^k XY^{\mathsf{T}}) + \mathbf{tr}(S^k Y X^{\mathsf{T}}) \right) \right)^2 \tag{63}
$$

$$
= \sum_{k=0}^{K-1} \left( \mathbf{tr}(S^k \cdot \frac{1}{2}(XY^{\mathsf{T}} + YX^{\mathsf{T}})) \right)^2 \tag{64}
$$

$$
= \sum_{k=0}^{K-1} \left( \mathbf{tr}(S^k C_{XY}) \right)^2 \geq \left( \frac{1}{\sqrt{K}} \sum_{k=0}^{K-1} |\mathbf{tr}(S^k C_{XY})| \right)^2 \tag{65}
$$

$$
\geq \left( \frac{1}{\sqrt{K}} \left| \sum_{k=0}^{K-1} \mathbf{tr}(S^k C_{XY}) \right| \right)^2 = \underbrace{\left( \frac{1}{\sqrt{K}} \mathbf{tr} \left( \left( \sum_{k=0}^{K-1} S^k \right) C_{XY} \right) \right)^2}_{\mathcal{A}_{\text{L}}} \tag{66}
$$

Above in equation 65 we used the triangle inequality and in equation 65 we used the fact that for any vector $\mathbf{z} \in \mathbb{R}^d$: $||\mathbf{z}||_2 \geq \frac{1}{\sqrt{d}} ||\mathbf{z}||_1$.[2]  $\qquad \square$

## D.3 Proof of Lemma 2

*Proof.* Starting with the original constraint: $|| \sum_{k=0}^{K-1} \tilde{S}^k \tilde{\mathbf{x}} \tilde{\mathbf{x}}^{\mathsf{T}} \tilde{S}^k ||_{\text{op}} < \alpha.$ , we will use upper bounds to get rid of the dependence on the data and to have a constraint that only depends on $S$:

$$
|| \sum_{k=0}^{K-1} \tilde{S}^k \tilde{\mathbf{x}} \tilde{\mathbf{x}}^{\mathsf{T}} \tilde{S}^k ||_{\text{op}} \leq \sum_{k=0}^{K-1} || \tilde{S}^k \tilde{\mathbf{x}} \tilde{\mathbf{x}}^{\mathsf{T}} \tilde{S}^k ||_{\text{op}} = \sum_{k=0}^{K-1} || \tilde{S}^k \tilde{\mathbf{x}} ||_2^2 \tag{67}
$$

Above, we used the fact that $\tilde{S}^k \tilde{\mathbf{x}} \tilde{\mathbf{x}}^{\mathsf{T}} \tilde{S}^k$ is a rank one matrix with its only non-zero eigenvalue being $|| \tilde{S}^k \tilde{\mathbf{x}} ||_2^2$.

$$
\sum_{k=0}^{K-1} || \tilde{S}^k \tilde{\mathbf{x}} ||_2^2 = \sum_{k=0}^{K-1} \sum_{i=1}^{M} || S^k \mathbf{x}_i ||_2^2 = \sum_{k=0}^{K-1} \sum_{i=1}^{M} \mathbf{x}_i^{\mathsf{T}} S^{2k} \mathbf{x}_i \tag{68}
$$

The eigen-decomposition of $S$ is given by

$$
S = \sum_{\ell=1}^{n} \gamma_\ell \boldsymbol{v}_\ell \boldsymbol{v}_\ell^{\mathsf{T}} . \tag{69}
$$

---

[2]A relevant question: When are these inequalities tight? whenever the terms $\mathbf{tr}(S^k C_{XY})$ are close to each other for different values of $k$, the inequalities are tighter. If $\mathbf{tr}(S^k C_{XY})$ is the same for every value of $k$, equality holds for both inequalities

Inserting equation 69 into equation 68 leads to

$$\sum_{k=0}^{K-1}\sum_{i=1}^{M}\mathbf{x}_i^\mathsf{T} S^{2k}\mathbf{x}_i = \sum_{k=0}^{K-1}\sum_{i=1}^{M}\sum_{\ell=1}^{n}\gamma_\ell^{2k}(\boldsymbol{v}_\ell^\mathsf{T}\mathbf{x}_i)^2 \tag{70}$$

Using the notation $\hat{x}_i$ to denote $\boldsymbol{v}_\ell^\mathsf{T}$, we have

$$\sum_{k=0}^{K-1}\sum_{i=1}^{M}\sum_{\ell=1}^{n}\gamma_\ell^{2k}(\boldsymbol{v}_\ell^\mathsf{T}\mathbf{x}_i)^2 = \sum_{k=0}^{K-1}\sum_{i=1}^{M}\langle\boldsymbol{\gamma}^{\odot 2k}, \hat{\mathbf{x}}_i^{\odot 2}\rangle \tag{71}$$

By the linearity of the inner product and Holder's inequality (keeping in mind that every element of both vectors is non-negative),

$$\sum_{k=0}^{K-1}\sum_{i=1}^{M}\langle\boldsymbol{\gamma}^{\odot 2k}, \hat{\mathbf{x}}_i^{\odot 2}\rangle = \langle\sum_{k=0}^{K-1}\boldsymbol{\gamma}^{\odot 2k}, \sum_{i=1}^{M}\hat{\mathbf{x}}_i^{\odot 2}\rangle \leq ||\sum_{k=0}^{K-1}\boldsymbol{\gamma}^{\odot 2k}||_1 \cdot ||\sum_{i=1}^{M}\hat{\mathbf{x}}_i^{\odot 2}||_\infty \tag{72}$$

Since $\hat{\mathbf{x}}_i$ consists of the coefficients of $\mathbf{x}_i$ projected onto the eigenspace of $S$, and recall that the input dataset is normalized, i.e., $||\hat{\mathbf{x}}_i||_2 = ||\mathbf{x}_i||_2 = 1$, the term $||\sum_{i=1}^{M}\hat{\mathbf{x}}_i^{\odot 2}||_\infty$ in equation 72 can be upper bounded as

$$||\sum_{i=1}^{M}\hat{\mathbf{x}}_i^{\odot 2}||_\infty \leq \sum_{i=1}^{M}||\hat{\mathbf{x}}_i^{\odot 2}||_\infty \leq \sum_{i=1}^{M}||\hat{\mathbf{x}}_i^{\odot 2}||_1 = \sum_{i=1}^{M}||\hat{\mathbf{x}}_i||_2^2 \leq M \tag{73}$$

Using the upper bound from equation 73 we can further upper bound the quantity from equation 72:

$$\begin{aligned}
\sum_{k=0}^{K-1}\sum_{i=1}^{M}\langle\boldsymbol{\gamma}^{\odot 2k}, \hat{\mathbf{x}}_i^{\odot 2}\rangle &\leq M \cdot ||\sum_{k=0}^{K-1}\boldsymbol{\gamma}^{\odot 2k}||_1 \\
&= M \cdot ||\sum_{k=0}^{K-1}\boldsymbol{\gamma}^{\odot k}||_2^2 \\
&= M \cdot ||\sum_{k=0}^{K-1}S^k||_F^2
\end{aligned} \tag{74}$$

Putting together equations equation 67 - equation 74 we get:

$$||\sum_{k=0}^{K-1}S^k||_F \leq \sqrt{\alpha/(\eta M)} \Rightarrow \eta \cdot ||\sum_{k=0}^{K-1}\tilde{S}^k\tilde{\mathbf{x}}\tilde{\mathbf{x}}^\mathsf{T}\tilde{S}^k||_{\mathsf{op}} \leq \alpha \tag{75}$$

which concludes the proof. $\qquad\square$

### D.4 PROOF OF THEOREM 2

*Proof.* We restate the optimization problem to be considered for the result in this theorem.

$$S^* = \underset{S}{\arg\max}\left(\frac{1}{\sqrt{K}}\mathbf{tr}\left(\left(\sum_{k=0}^{K-1}S^k\right)C_{XY}\right)\right)^2 \quad \textbf{s.t. } ||\sum_{k=0}^{K-1}S^k||_F \leq \sqrt{\alpha/(\eta M)} \tag{76}$$

The vectorized form of equation 76 is given by

$$\mathbf{vec}(S^*) = \underset{\mathbf{vec}(S)}{\arg\max}\left(\frac{1}{\sqrt{K}}\langle\mathbf{vec}(\sum_{k=0}^{K-1}S^k), \mathbf{vec}(C_{XY})\rangle\right)^2 \quad \textbf{s.t. } ||\mathbf{vec}(\sum_{k=0}^{K-1}S^k)||_2 \leq \sqrt{\alpha/(\eta M)} \tag{77}$$

Thus, equation 77 is equivalent to maximizing the projection of the vector $\mathbf{vec}(\sum_{k=0}^{K-1}S^k)$ along the direction of $\mathbf{vec}(C_{XY})$ on an $\ell_2$-norm ball of radius $\sqrt{\alpha/(\eta M)}$, which has the following solution

$$\sum_{k=0}^{K-1}(S^*)^k = \mu \cdot C_{XY} \tag{78}$$

where $\mu = \frac{\sqrt{\alpha/(\eta M)}}{||C_{XY}||_F}$ is a normalizing constant to ensure the Frobenius norm of $\sum_{k=0}^{K-1}(S^*)^k$ satisfies our constraint. Note that while a solution $S^*$ satisfying equation 78 might not exist for some values of $K$, When designing the architecture we could choose $K$ in a way that equation 78 has a solution. For example with $K = 2$ there is always a solution as given by equation 19. $\qquad\square$

### D.5 PROOF OF LEMMA 3

*Proof.* From equation 27, we can write:

$$
\begin{aligned}
E_{ab} &= \mathop{\mathbb{E}}_{\boldsymbol{g}\sim\mathcal{N}(0,I)} \left[ \sigma\left(\langle \boldsymbol{g}, \boldsymbol{z}^{(a)}\rangle\right) \cdot \sigma\left(\langle \boldsymbol{g}, \boldsymbol{z}^{(b)}\rangle\right) \right] \\
&= \mathop{\mathbb{E}}_{\boldsymbol{g}\sim\mathcal{N}(0,I)} \left[ \sigma\left(||\boldsymbol{z}^{(a)}||_2 \cdot \langle \boldsymbol{g}, \frac{\boldsymbol{z}^{(a)}}{||\boldsymbol{z}^{(a)}||_2}\rangle\right) \cdot \sigma\left(||\boldsymbol{z}^{(b)}||_2 \cdot \langle \boldsymbol{g}, \frac{\boldsymbol{z}^{(b)}}{||\boldsymbol{z}^{(b)}||_2}\rangle\right) \right]
\end{aligned}
\tag{79}
$$

In order to analyze equation 79, we define two random variables $u \triangleq \langle \boldsymbol{g}, \frac{\boldsymbol{z}^{(a)}}{||\boldsymbol{z}^{(a)}||_2}\rangle$ an $u' \triangleq \langle \boldsymbol{g}, \frac{\boldsymbol{z}^{(b)}}{||\boldsymbol{z}^{(b)}||_2}\rangle$. It is evident that the random variables $u, u'$ are correlated, mean-zero Gaussian random variables with their joint distribution given by

$$
u, u' \sim \mathcal{N}\left( \begin{bmatrix} 0 \\ 0 \end{bmatrix}, \Lambda = \begin{bmatrix} 1 & \frac{\langle \boldsymbol{z}^{(a)}, \boldsymbol{z}^{(b)}\rangle}{||\boldsymbol{z}^{(a)}||_2 \cdot ||\boldsymbol{z}^{(b)}||_2} \\ \frac{\langle \boldsymbol{z}^{(a)}, \boldsymbol{z}^{(b)}\rangle}{||\boldsymbol{z}^{(a)}||_2 \cdot ||\boldsymbol{z}^{(b)}||_2} & 1 \end{bmatrix} \right)
\tag{80}
$$

Therefore from equation 79 we can write

$$
E_{ab} = \mathop{\mathbb{E}}_{u, u'\sim\mathcal{N}(0,\Lambda)} \left[ \sigma\left(||\boldsymbol{z}^{(a)}||_2 \cdot u\right) \cdot \sigma\left(||\boldsymbol{z}^{(b)}||_2 \cdot u'\right) \right]
\tag{81}
$$

In order to analyze $E_{ab}$, we leverage Hermite polynomials, which are discussed next.

**Hermite Polynomials.** Hermite polynomials are a collection of functions $(p_j)_{j\in\mathbb{N}}$ which form an orthonormal basis for the space of square-integrable functions. The first few of these polynomials can be seen below:

$$
p_0(z) = 1, \quad p_1(z) = z, \quad p_2(z) = \frac{z^2 - 1}{\sqrt{2}}, \quad p_3(z) = \frac{z^3 - 3z}{\sqrt{6}}, \quad \ldots
\tag{82}
$$

We can define the inner product between two square-integrable functions $f$ and $g$ as

$$
\langle f, g\rangle = \int_{-\infty}^{\infty} f(z) \cdot g(z) \cdot e^{-z^2/2} dz
\tag{83}
$$

Keeping in mind that the Hermite polynomials are orthonormal with respect to the inner product defined in equation 83, we can expand any function $f(z)$ for which $\langle f, f\rangle$ is bounded, in terms of these polynomials:

$$
f(z) = \sum_{\ell=0}^{\infty} \alpha_\ell \cdot p_\ell(z)
\tag{84}
$$

where we have

$$
\alpha_\ell = \langle f, p_l\rangle = \int_{-\infty}^{\infty} f(z) \cdot p_\ell(z) \cdot e^{-z^2/2} dz
\tag{85}
$$

In our analysis, we also leverage another property of these polynomials (O'Donnell, 2014, section 11.2), which is given by

$$
\mathop{\mathbb{E}}_{\substack{(z,z') \\ \rho\text{-correlated}}} [p_j(z)p_k(z')] = \begin{cases} \rho^j & \text{if } j = k, \\ 0 & \text{if } j \neq k \end{cases}
\tag{86}
$$

Essentially, equation 86 establishes orthogonality of the polynomials $p_j(z)$ and $p_k(z')$ when $(z, z')$ are a pair of $\rho$-correlated standard Gaussian random variables. It is straightforward to see that setting

$\rho = 1$ in equation 86 recovers the orthonormality of the Hermite polynomials. Based on equation 85 can now write the Hermite expansion of the two functions in equation 81 as:

$$\sigma\left(||\boldsymbol{z}^{(a)}||_2 \cdot u\right) = \sum_{\ell=0}^{\infty} \alpha_\ell \cdot p_\ell(u), \ \ \sigma\left(||\boldsymbol{z}^{(b)}||_2 \cdot u'\right) = \sum_{\ell'=0}^{\infty} \beta_{\ell'} \cdot p_{\ell'}(u') \tag{87}$$

Inserting these into equation 81 we get

$$E_{ab} = \mathop{\mathbb{E}}_{u,u' \sim \mathcal{N}(0,\Lambda)} \left[ \sum_{\ell=0}^{\infty} \alpha_\ell \cdot p_\ell(u) \sum_{\ell'=0}^{\infty} \beta_{\ell'} \cdot p_{\ell'}(u') \right] \tag{88}$$

$$= \sum_{\ell=0}^{\infty} \sum_{\ell'=0}^{\infty} \alpha_\ell \beta_{\ell'} \mathop{\mathbb{E}}_{u,u' \sim \mathcal{N}(0,\Lambda)} \left[ p_\ell(u) p_{\ell'}(u') \right] \tag{89}$$

$$= \sum_{\ell=0}^{\infty} \alpha_\ell \beta_\ell \cdot \left( \frac{\langle \boldsymbol{z}^{(a)}, \boldsymbol{z}^{(b)} \rangle}{||\boldsymbol{z}^{(a)}||_2 \cdot ||\boldsymbol{z}^{(b)}||_2} \right)^\ell \tag{90}$$

where to get from equation 89 to equation 90 we used the orthogonality property of Hermite polynomials from equation 86. Here, we remark that the coefficients $\alpha_\ell, \beta_\ell$ depend on the choice of non-linear activation function and the magnitudes $||\boldsymbol{z}^{(a)}||_2$ and $||\boldsymbol{z}^{(b)}||_2$ respectively.

For the subsequent analysis, we consider the setting where the non-linear activation function is the hyperbolic tangent function, i.e., $\sigma(y) = \tanh(y)$. Note that $\tanh$ is an odd function. Since the Hermite polynomial $p_\ell(z)$ is odd for odd $\ell$ and even for even $\ell$, for even $\ell$ we have

$$\alpha_\ell = \int_{-\infty}^{\infty} \sigma(||\boldsymbol{z}_a||_2 u) \cdot p_\ell(u) \cdot e^{-u^2/2} du = 0 \tag{91}$$

and similarly we have $\beta_\ell = 0$. This leaves us with only the odd terms in the sum from equation 90, therefore we can write $E_{ab}$ as

$$E_{ab} = \underbrace{\alpha_1 \beta_1 \cdot \frac{\langle \boldsymbol{z}^{(a)}, \boldsymbol{z}^{(b)} \rangle}{||\boldsymbol{z}^{(a)}||_2 \cdot ||\boldsymbol{z}^{(b)}||_2}}_{B_{ab}} + \underbrace{\sum_{\ell=3,5,\cdots}^{\infty} \alpha_\ell \beta_\ell \cdot \left( \frac{\langle \boldsymbol{z}^{(a)}, \boldsymbol{z}^{(b)} \rangle}{||\boldsymbol{z}^{(a)}||_2 \cdot ||\boldsymbol{z}^{(b)}||_2} \right)^\ell}_{\Delta B_{ab}} \tag{92}$$

For conciseness moving forward, we define the matrices $B \in \mathbb{R}^{nM \times nM}$ which represents the first non-zero term in the expansion and $\Delta B \in \mathbb{R}^{nM \times nM}$ which includes all the subsequent terms. Now, recalling equation 26 and replacing $E$ with the expansion given in equation 92, the alignment $\mathcal{A}$ is given by

$$\mathcal{A} = \mathbf{tr}(QE) = \mathbf{tr}(QB) + \mathbf{tr}(Q\Delta B) . \tag{93}$$

Hence, we conclude the proof. □

### D.6 PROOF OF LEMMA 4

*Proof.* We begin with the analysis of the $(a, b)$-th entry of matrix $B$, i.e.,

$$B_{ab} = \alpha_1 \beta_1 \cdot \frac{\langle \boldsymbol{z}^{(a)}, \boldsymbol{z}^{(b)} \rangle}{||\boldsymbol{z}^{(a)}||_2 \cdot ||\boldsymbol{z}^{(b)}||_2} \tag{94}$$

For this purpose, we define the nonlinear function $\hat{\sigma}(\cdot)$ as

$$\hat{\sigma}(||\boldsymbol{z}_a||_2^2) \triangleq \frac{\alpha_1}{||\boldsymbol{z}_a||_2} \tag{95}$$

$$= \frac{\int_{-\infty}^{\infty} \sigma(||\boldsymbol{z}_a||_2 u) \cdot p_1(u) \cdot e^{-u^2/2} du}{||\boldsymbol{z}_a||_2} \tag{96}$$

Keeping in mind that $(B_{\mathsf{lin}})_{aa} = \langle \boldsymbol{z}_a, \boldsymbol{z}_a \rangle = ||\boldsymbol{z}_a||_2^2$, we can write:

$$B_{ab} = \hat{\sigma}((B_{\mathsf{lin}})_{aa}) \cdot \hat{\sigma}((B_{\mathsf{lin}})_{bb}) \cdot (B_{\mathsf{lin}})_{ab} \tag{97}$$

The element-wise equation in equation 97 implies the following:

$$B = \hat{\sigma}(\mathbf{diag}(B_{\mathsf{lin}}))B_{\mathsf{lin}}\hat{\sigma}(\mathbf{diag}(B_{\mathsf{lin}})) \tag{98}$$

where $\mathbf{diag}(B_{\mathsf{lin}})$ is a square $nM \times nM$ matrix equal to $B_{\mathsf{lin}}$ on the diagonal and zero everywhere else. Next, we focus on the first term in the expansion of the alignment given in equation 35.

$$\mathbf{tr}(QB) = \mathbf{tr}\left(\left(\sum_{k=0}^{K-1} \tilde{S}^k \tilde{\mathbf{y}} \tilde{\mathbf{y}}^\mathsf{T} \tilde{S}^k\right) B\right) = \sum_{k=0}^{K-1} \mathbf{tr}(Q_k B) \tag{99}$$

where $Q_k \triangleq \tilde{S}^k \tilde{\mathbf{y}} \tilde{\mathbf{y}}^\mathsf{T} \tilde{S}^k$. Using equation 98 we can write:

$$\mathbf{tr}\left(Q_k B\right) = \mathbf{tr}\left(Q_k \cdot \hat{\sigma}(\mathbf{diag}(B_{\mathsf{lin}}))B_{\mathsf{lin}}\hat{\sigma}(\mathbf{diag}(B_{\mathsf{lin}}))\right) \tag{100}$$

We note that $Q_k$ is a rank one matrix. Therefore $Q_k B$ also has rank at most one. Keeping in mind that $\hat{\sigma}(\mathbf{diag}(B_{\mathsf{lin}}))$ is a diagonal matrix (with non-negative entries) and therefore its eigenvalues are the elements on its diagonal, calling the largest of these $\lambda_{max}(\hat{\sigma}(\mathbf{diag}(B_{\mathsf{lin}})))$ or $\lambda_{max}$ for short and the smallest one $\lambda_{min}$, we can write:

$$
\begin{aligned}
\lambda_{min}\mathbf{tr}\left(Q_k B_{\mathsf{lin}}\right) &\leq \mathbf{tr}\left(Q_k B_{\mathsf{lin}} \cdot \hat{\sigma}(\mathbf{diag}(B))\right) \leq \lambda_{max}\mathbf{tr}\left(Q_k B_{\mathsf{lin}}\right) \\
\Rightarrow \lambda_{min}\mathbf{tr}\left(Q_k B_{\mathsf{lin}}\right) &\leq \mathbf{tr}\left(B_{\mathsf{lin}} \cdot \hat{\sigma}(\mathbf{diag}(B_{\mathsf{lin}}))Q_k\right) \\
\Rightarrow \lambda_{min}^2\mathbf{tr}\left(Q_k B_{\mathsf{lin}}\right) &\leq \mathbf{tr}\left(\hat{\sigma}(\mathbf{diag}(B_{\mathsf{lin}}))B_{\mathsf{lin}} \cdot \hat{\sigma}(\mathbf{diag}(B_{\mathsf{lin}}))Q_k\right) \\
&= \mathbf{tr}\left(Q_k B\right)
\end{aligned}
\tag{101}
$$

We recall from equation 30 that

$$\mathcal{A}_{\mathsf{lin}} = \mathbf{tr}(QB_{\mathsf{lin}}) = \sum_{k=0}^{K-1} \mathbf{tr}(Q_k B_{\mathsf{lin}}) \tag{102}$$

Using equation 101 and equation 102 we get

$$\lambda_{min}^2\mathcal{A}_{\mathsf{lin}} \leq \sum_{k=0}^{K-1} \mathbf{tr}\left(Q_k B\right) = \mathbf{tr}(QB) \tag{103}$$

Note that

$$\lambda_{min} = \min_a \hat{\sigma}(||\mathbf{z}^{(a)}||_2^2) = \min_a \hat{\sigma}(\sum_{k=0}^{K-1}(\tilde{S}^k \tilde{\mathbf{x}})_a^2) = \hat{\sigma}(\max_a \sum_{k=0}^{K-1}(\tilde{S}^k \tilde{\mathbf{x}})_a^2) , \tag{104}$$

where we have used the fact that $\hat{\sigma}(.)$ is a non-increasing function (see Figure 2) to establish equation 104.

Further, because $\hat{\sigma}(.)$ is a non-increasing function, an upper bound on $\sum_{k=0}^{K-1}(\tilde{S}^k \tilde{\mathbf{x}})_a^2$ will give us a corresponding lower bound on $\lambda_{min}$. To find an upper bound on $\sum_{k=0}^{K-1}(\tilde{S}^k \tilde{\mathbf{x}})_a^2$, we provide the following analyses

$$\max_a \sum_{k=0}^{K-1}(\tilde{S}^k \tilde{\mathbf{x}})_a^2 \leq \max_{a',i} \sum_{k=0}^{K-1}(S^k \mathbf{x}_i)_{a'}^2 \tag{105}$$

$$\leq \max_i \sum_{k=0}^{K-1} ||S^k \mathbf{x}_i||_2^2 \leq \max_i \sum_{k=0}^{K-1} ||S^k||_{\mathsf{op}}^2 ||\mathbf{x}_i||_2^2 \tag{106}$$

$$\leq \sum_{k=0}^{K-1} ||S^k||_{\mathsf{op}}^2 = \sum_{k=0}^{K-1} ||S||_{\mathsf{op}}^{2k} \tag{107}$$

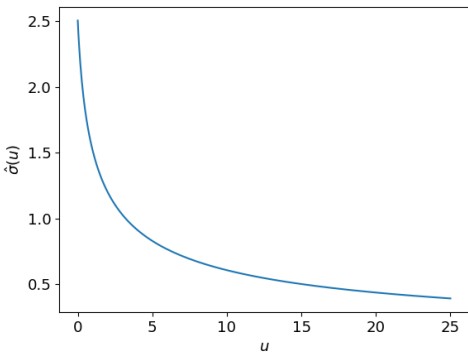

Figure 2: The function $\hat{\sigma}(\cdot)$ in the case where $\sigma(x) = \tanh(x)$

where in the last inequality we used the fact that for every sample $\mathbf{x}_i$ in the dataset we have $||\mathbf{x}_i||_2 \leq 1$. Therefore, we can write

$$\lambda_{min}^2 = \left( \hat{\sigma}(\max_a \sum_{k=0}^{K-1} (\tilde{S}^k \tilde{\mathbf{x}})_a^2) \right)^2 \geq \left( \hat{\sigma} \left( \sum_{k=0}^{K-1} ||S||_{\mathsf{op}}^{2k} \right) \right)^2 \tag{108}$$

Considering that, in practice, bounding the norm of the GSO $S$ through normalization, (see Appendix H) ensures that the constraint $||\eta \cdot \tilde{\Theta}||_{\mathsf{op}} < \alpha$ of our optimization problem is satisfied, we next consider the mild assumption that $||S||$ is upper bounded. Formally, this assumption is given by

$$||S||_{\mathsf{op}} \leq \nu \tag{109}$$

Using equation 109 and equation 108, we reach

$$\lambda_{min}^2 \geq \rho \tag{110}$$

$$\text{where } \rho \triangleq \left( \hat{\sigma} \left( \sum_{k=0}^{K-1} \nu^{2k} \right) \right)^2 \tag{111}$$

Thus, we can rewrite equation 103 in terms of $\rho$ as

$$\mathbf{tr}(QB) \geq \rho \mathcal{A}_{\mathsf{lin}} \tag{112}$$

which concludes the proof.

$\square$

### D.7 PROOF OF LEMMA 5

*Proof.* We start by defining the Hermite transform of the $\tanh$ functioon as $g_\ell$ for clarity in subsequent analysis and to emphasize that the $\ell$-th coefficient $\alpha_\ell$ is a function of $||\mathbf{z}^{(a)}||_2$.

$$g_\ell(||\mathbf{z}^{(a)}||_2) \triangleq \int_{-\infty}^{\infty} \tanh(||\mathbf{z}^{(a)}||_2 u) \cdot p_\ell(u) \cdot e^{-u^2/2} du = \alpha_i \tag{113}$$

From equation 34, we recall that

$$\begin{aligned} \Delta B_{ab} &= \sum_{i=1}^{\infty} \alpha_{2i+1} \beta_{2i+1} \cdot \left( \frac{\langle \mathbf{z}^{(a)}, \mathbf{z}^{(b)} \rangle}{||\mathbf{z}^{(a)}||_2 \cdot ||\mathbf{z}^{(b)}||_2} \right)^{2i+1} \\ &= \sum_{i=1}^{\infty} g_{2i+1}(||\mathbf{z}^{(a)}||_2) \cdot g_{2i+1}(||\mathbf{z}^{(b)}||_2) \cdot \left( \frac{\langle \mathbf{z}^{(a)}, \mathbf{z}^{(b)} \rangle}{||\mathbf{z}^{(a)}||_2 \cdot ||\mathbf{z}^{(b)}||_2} \right)^{2i+1} \end{aligned} \tag{114}$$

It can readily be verified numerically that for a given index $\ell$, we either have $g_\ell(y) \geq 0$, $\forall y \geq 0$ or $g_\ell(y) \leq 0$, $\forall y \geq 0$.[3] Consequently, we have $g_{2i+1}(||\mathbf{z}^{(a)}||_2) \cdot g_{2i+1}(||\mathbf{z}^{(b)}||_2) \geq 0$. We can further

---

[3] A simple Python script can be found in https://github.com/shervinkh2000/Cross_Covariance_NTK that plots $g_\ell(y)$ against $y$ for any given index $\ell$. See the file "numerical_verification_1.py"

conclude that $\Delta B_{ab}$ and $B_{ab}$ have the same sign, which is given by $\mathrm{sign}(\langle \boldsymbol{z}^{(a)}, \boldsymbol{z}^{(b)} \rangle)$ since recall that:

$$B_{ab} = g_1(||\boldsymbol{z}^{(a)}||_2)g_1(||\boldsymbol{z}^{(b)}||_2) \cdot \frac{\langle \boldsymbol{z}^{(a)}, \boldsymbol{z}^{(b)} \rangle}{||\boldsymbol{z}^{(a)}||_2 \cdot ||\boldsymbol{z}^{(b)}||_2} \tag{115}$$

Next, in the scenario $\langle \boldsymbol{z}^{(a)}, \boldsymbol{z}^{(b)} \rangle \geq 0$, we have

$$\sum_{i=1}^{\infty} g_{2i+1}(||\boldsymbol{z}^{(a)}||_2) \cdot g_{2i+1}(||\boldsymbol{z}^{(b)}||_2) \cdot \left( \frac{\langle \boldsymbol{z}^{(a)}, \boldsymbol{z}^{(b)} \rangle}{||\boldsymbol{z}^{(a)}||_2 \cdot ||\boldsymbol{z}^{(b)}||_2} \right)^{2i+1}$$

$$\leq \sum_{i=1}^{\infty} g_{2i+1}(||\boldsymbol{z}^{(a)}||_2) \cdot g_{2i+1}(||\boldsymbol{z}^{(b)}||_2) \cdot \left( \frac{\langle \boldsymbol{z}^{(a)}, \boldsymbol{z}^{(b)} \rangle}{||\boldsymbol{z}^{(a)}||_2 \cdot ||\boldsymbol{z}^{(b)}||_2} \right) \tag{116}$$

$$= \frac{\left( \sum_{i=1}^{\infty} g_{2i+1}(||\boldsymbol{z}^{(a)}||_2) \cdot g_{2i+1}(||\boldsymbol{z}^{(b)}||_2) \right)}{g_1(||\boldsymbol{z}^{(a)}||_2) \cdot g_1(||\boldsymbol{z}^{(b)}||_2)} \cdot B_{ab} \tag{117}$$

We established previously that $g_{2i+1}(||\boldsymbol{z}^{(a)}||_2)$ and $g_{2i+1}(||\boldsymbol{z}^{(b)}||_2)$ have the same sign for any $i \geq 0$. Further, $\left| \frac{g_{2i+1}(||\boldsymbol{z}^{(a)}||_2)}{g_1(||\boldsymbol{z}^{(a)}||_2)} \right|$ is a non-decreasing function in $||\boldsymbol{z}^{(a)}||_2$ for any $i \geq 0$ [4]. Hence, we can continue from equation 117, such that, we have

$$\frac{\left( \sum_{i=1}^{\infty} g_{2i+1}(||\boldsymbol{z}^{(a)}||_2) \cdot g_{2i+1}(||\boldsymbol{z}^{(b)}||_2) \right)}{g_1(||\boldsymbol{z}^{(a)}||_2) \cdot g_1(||\boldsymbol{z}^{(b)}||_2)} \cdot B_{ab} \quad \leq \quad \lim_{y \to \infty} \frac{\left( \sum_{i=1}^{\infty} g_{2i+1}(y) \cdot g_{2i+1}(y) \right)}{g_1(y) \cdot g_1(y)} \cdot B_{ab}$$

$$= \quad \left( \lim_{y \to \infty} \frac{\sum_{i=1}^{\infty} (g_{2i+1}(y))^2}{(g_1(y))^2} \right) B_{ab} \tag{118}$$

which implies that

$$\Delta B_{ab} \leq \beta \cdot B_{ab} \tag{119}$$

where

$$\beta \quad \triangleq \quad \lim_{y \to \infty} \frac{\sum_{i=1}^{\infty} (g_{2i+1}(y))^2}{(g_1(y))^2}$$

$$= \quad \frac{\sum_{i=1}^{\infty} \left( \int_{-\infty}^{\infty} (\lim_{y \to \infty} \tanh(y \cdot u)) \cdot p_\ell(u) \cdot e^{-u^2/2} du \right)^2}{\left( \int_{-\infty}^{\infty} (\lim_{y \to \infty} \tanh(y \cdot u)) \cdot p_1(u) \cdot e^{-u^2/2} du \right)^2} = \frac{\pi - 2}{2} \simeq 0.57 \tag{120}$$

Noting that all of the inequalities from equation 116 and equation 118 hold in the opposite direction when $B_{ab} < 0$, the proof is concluded. $\qquad\square$

### D.8   PROOF OF THEOREM 3

*Proof.* From Lemma 5, we know that the first term in the expansion of the alignment i.e., $\mathbf{tr}(QB)$ is lower bounded by $\mathcal{A}_{\mathsf{lin}}$ multiplied by a constant. Hence, it remains to analyze how the second term, $\mathbf{tr}(Q\Delta B)$, can affect alignment. To begin with, we make the following assumption that the alignment in the linear case is sufficiently large, i.e.,

$$\mathcal{A}_{\mathsf{lin}} = \mathbf{tr}(QB_{\mathsf{lin}}) \geq \xi \cdot ||Q||_F ||B_{\mathsf{lin}}||_F \tag{121}$$

where $0 \leq \xi \leq 1$, is some constant that quantifies how large the alignment is in the linear case. We define the matrix $W \in \mathbb{R}^{nM \times nM}$, such that, for any $(a, b)$-th entry of the matrices $B, \Delta B$, and $W$, the following holds

$$(\Delta B)_{ab} = B_{ab} W_{ab} \tag{122}$$

---

[4] A simple Python script can be found in https://github.com/shervinkh2000/Cross_Covariance_NTK that plots $\left| \frac{g_{2i+1}(y)}{g_1(y)} \right|$ against $y$ for any given index $i$. See the file "numerical_verification_2.py"

Recall from Lemma equation 5 that for the $(a, b)$-th entries of $B$ and $\Delta B$, we have

$$|(\Delta B)_{ab}| \leq \beta B_{ab} \quad \text{and} \quad (\Delta B)_{ab} B_{ab} \geq 0 \tag{123}$$

Therefore, the entries of $W$ must satisfy $0 \leq W_{ab} \leq \beta$.

**Analysis of term $\mathbf{tr}\,(Q \Delta B)$.** Based on the discussion above, we now analyze the term $\mathbf{tr}\,(Q \Delta B)$ in the alignment.

$$
\begin{aligned}
\mathbf{tr}\,(Q \Delta B) &= \sum_{a,b} Q_{ab} (\Delta B)_{ab} \\
&= \sum_{a,b} Q_{ab} B_{ab} W_{ab} \\
&= \sum_{Q_{ab} B_{ab} \geq 0} Q_{ab} B_{ab} W_{ab} + \sum_{Q_{ab} B_{ab} < 0} Q_{ab} B_{ab} W_{ab} \\
&\geq \beta \cdot \sum_{Q_{ab} B_{ab} < 0} Q_{ab} B_{ab}
\end{aligned}
\tag{124}
$$

To achieve the last inequality, we consider the worst case scenario, i.e., the most negative value possible for $\mathbf{tr}\,(Q \cdot (\Delta B))$. The worst case scenario corresponds to the setting when $W_{ab} = 0$ for all non-negative terms in the sum $\sum_{a,b} Q_{ab} B_{ab} W_{ab}$, and $W_{ab} = \beta$ for all the negative terms. For conciseness, we introduce the following notation:

$$\mathbf{tr}\,(QB)_{+} = \sum_{Q_{ab} B_{ab} \geq 0} Q_{ab} B_{ab}, \quad \text{and} \quad \mathbf{tr}\,(QB)_{-} = \sum_{Q_{ab} B_{ab} < 0} Q_{ab} B_{ab} \tag{125}$$

Thus, we can write

$$\mathbf{tr}\,(Q \Delta B) \geq -\beta \left| \mathbf{tr}\,(QB)_{-} \right| \tag{126}$$

Next, we aim to provide an upper bound on $\left| \mathbf{tr}\,(QB)_{-} \right|$. Recall from equation 98 that

$$B = \hat{\sigma}(\mathbf{diag}(B_{\mathsf{lin}})) \cdot B_{\mathsf{lin}} \cdot \hat{\sigma}(\mathbf{diag}(B_{\mathsf{lin}}))$$

Hence, since $\hat{\sigma}(\mathbf{diag}(B_{\mathsf{lin}}))$ is a diagonal matrix, it readily follows that

$$\lambda_{min}^2 ||B_{\mathsf{lin}}||_F \leq ||B||_F \leq \lambda_{max}^2 ||B_{\mathsf{lin}}||_F \tag{127}$$

where $\lambda_{max} = \max_i (\hat{\sigma}(\mathbf{diag}(B_{\mathsf{lin}})))_{ii}$ is the largest element on the diagonal of $\hat{\sigma}(\mathbf{diag}(B_{\mathsf{lin}}))$ and therefore, also its largest eigenvalue. Similarly, $\lambda_{min} = \min_i (\hat{\sigma}(\mathbf{diag}(B_{\mathsf{lin}})))_{ii}$ is the smallest element on the diagonal of $\hat{\sigma}(\mathbf{diag}(B_{\mathsf{lin}}))$ and its smallest eigenvalue. Note that all elements on the diagonal of $B_{\mathsf{lin}}$ are non-negative. From equation 101, we have

$$
\begin{aligned}
\mathbf{tr}\,(QB) &\geq \lambda_{min}^2 \mathbf{tr}\,(QB_{\mathsf{lin}}) \\
&\geq \xi \lambda_{min}^2 ||Q||_F ||B_{\mathsf{lin}}||_F \\
&\geq \xi \frac{\lambda_{min}^2}{\lambda_{max}^2} ||Q||_F ||B||_F \\
\Rightarrow \mathbf{tr}\,(QB) &= \left| \mathbf{tr}\,(Q \cdot B)_{+} \right| - \left| \mathbf{tr}\,(Q \cdot B)_{-} \right| \\
&\geq \xi \frac{\lambda_{min}^2}{\lambda_{max}^2} ||Q||_F ||B||_F
\end{aligned}
\tag{128}
$$

Since changing the signs of individual elements of a matrix does not change the Frobenius norm of said matrix, we can write

$$\left| \mathbf{tr}\,(Q \cdot B)_{+} \right| + \left| \mathbf{tr}\,(Q \cdot B)_{-} \right| \leq ||Q||_F ||B||_F \tag{129}$$

From equation 128 and equation 129, we have the following

$$\left| \mathbf{tr}\,(Q \cdot B)_{-} \right| \leq 1/2 (1 - \xi \frac{\lambda_{min}^2}{\lambda_{max}^2}) ||Q||_F ||B||_F \tag{130}$$

Next, recalling equation 126 and using equation 130, we have

$$
\begin{aligned}
\mathbf{tr}\left(Q \cdot (\Delta B)\right) &\geq -\beta/2(1 - \xi\frac{\lambda_{min}^2}{\lambda_{max}^2})||Q||_F||B||_F \\
&\geq -\beta/2(\frac{\lambda_{max}^2}{\xi\lambda_{min}^2} - 1)\mathbf{tr}\left(QB\right)
\end{aligned}
\tag{131}
$$

Using equation 131, we can lower bound $\mathcal{A}$ as follows

$$
\begin{aligned}
\mathcal{A} &= \mathbf{tr}\left(QB\right) + \mathbf{tr}\left(Q \cdot (\Delta B)\right) \\
&\geq \left(1 - \beta/2(\frac{\lambda_{max}^2}{\xi\lambda_{min}^2} - 1)\right)\mathbf{tr}\left(QB\right) \\
&\geq \lambda_{min}^2\left(1 - \beta/2(\frac{\lambda_{max}^2}{\xi\lambda_{min}^2} - 1)\right)\mathcal{A}_{\mathsf{lin}}
\end{aligned}
\tag{132}
$$

Now recall from the proof of Lemma 4 (Equations 104 to 110):

$$
\lambda_{min}^2 = \left(\hat{\sigma}(\max_a \sum_{k=0}^{K-1} (\tilde{S}^k\tilde{\mathbf{x}})_a^2)\right)^2 \geq \left(\hat{\sigma}\left(\sum_{k=0}^{K-1} \nu^{2k}\right)\right)^2 = \rho
\tag{133}
$$

Similarly for $\lambda_{max}$, using the fact that $\hat{\sigma}(.)$ is a non-increasing function:

$$
\lambda_{max} = \max_a \hat{\sigma}(||\mathbf{z}^{(a)}||_2^2) \leq \hat{\sigma}(0) \leq 2.51
\tag{134}
$$

Using equation 133 and equation 134, from equation 132 we can continue to write:

$$
\mathcal{A} \geq \left(\rho\left(1 + \frac{\beta}{2}\right) - \frac{\beta}{2} \cdot \left(\frac{2.51}{\rho}\right)^2 \cdot \frac{1}{\xi}\right)\mathcal{A}_{\mathsf{lin}}
\tag{135}
$$

Further, using the following definitions

$$
c \triangleq \rho\left(1 + \frac{\beta}{2}\right), \quad \text{and} \quad d \triangleq \frac{\beta}{2} \cdot \left(\frac{2.51}{\rho}\right)^2,
\tag{136}
$$

we can rewrite equation 132 as

$$
\mathcal{A} \geq (c - \frac{d}{\xi})\mathcal{A}_{\mathsf{lin}}
\tag{137}
$$

Thus, the proof is concluded. $\qquad\square$

### D.9 PROOF OF LEMMA 9

*Proof.* Our approach is to upper bound the Rademacher complexity term in equation 190 for the class of hypotheses $\tilde{\mathcal{H}}_{filt}(B)$ where the vector of filter coefficients are close to some initialization. Recall the definition of $\tilde{\mathcal{H}}_{filt}(B)$ from equation 192:

$$
\tilde{\mathcal{H}}_{filt}(B) \triangleq \left\{ \boldsymbol{f}_{\mathbf{x}}(\boldsymbol{h}) = \sum_{k=0}^{K-1} h_k S^k\mathbf{x} \,\middle|\, \boldsymbol{h} \in \mathbb{R}^K, \ ||\boldsymbol{h} - \boldsymbol{h}^{(0)}||_2 \leq B \right\}
$$

We can write the complexity term from equation 190 as:

$$
\begin{aligned}
R(\ell \circ \tilde{\mathcal{H}}_{filt} \circ S) &= \frac{1}{M} \mathop{\mathbb{E}}_{\boldsymbol{\sigma}\sim\{\pm1\}^M}\left[\sup_{h\in\tilde{\mathcal{H}}_{filt}} \sum_{i=1}^{M} \sigma_i \cdot \ell(f(\mathbf{x}_i), \mathbf{y}_i)\right] \\
&= \frac{1}{M} \mathop{\mathbb{E}}_{\boldsymbol{\sigma}\sim\{\pm1\}^M}\left[\sup_{||\boldsymbol{h}-\boldsymbol{h}^{(0)}||_2\leq B} \sum_{i=1}^{M} \sigma_i \cdot ||\sum_{k=0}^{K-1} h_k S^k\mathbf{x}_i - \mathbf{y}_i||_2^2\right]
\end{aligned}
\tag{138, 139}
$$

Next, we provide the following lemma (an application of Lemma 26.9 from (Shalev-Shwartz & Ben-David (2014))).

**Lemma 6** (Contraction lemma)**.** *Let $\phi : \mathbb{R} \rightarrow \mathbb{R}$ be a $\rho$-Lipschitz function, i.e., we have $|\phi(\alpha) - \phi(\beta)| \leq \rho|\alpha - \beta|, \forall \alpha, \beta \in \mathbb{R}$. For $\mathbf{a} \in \mathbb{R}^m$, let $\phi(\mathbf{a})$ denote the vector $[\phi_1(a_1), \ldots, \phi_m(y_m)]$ and let $\phi \circ A = \{\phi(\mathbf{a}) : a \in A\}$. Then, we have*

$$R(\phi \circ \ell \circ \mathcal{H} \circ S) \leq \rho R(\ell \circ \mathcal{H} \circ S).$$

We recall the assumption that $||\sum_{k=0}^{K-1} h_k S^k \mathbf{x}_i - \mathbf{y}_i||_2 \leq \rho$. Also note that the function $\phi(z) = z^2$ is $\rho$-Lipschitz continuous over the domain $|z| \leq \rho$, therefore using Lemma 6 and equation 139 we can write:

$$R(\ell \circ \tilde{\mathcal{H}}_{filt} \circ S) \leq \rho \cdot \frac{1}{M} \mathop{\mathbb{E}}_{\boldsymbol{\sigma} \sim \{\pm 1\}^M} \left[ \sup_{||\boldsymbol{h} - \boldsymbol{h}^{(0)}||_2 \leq B} \sum_{i=1}^{M} \sigma_i \cdot || \sum_{k=0}^{K-1} h_k S^k \mathbf{x}_i - \mathbf{y}_i ||_2 \right] \quad (140)$$

Next, we re-state Corollary 4 from Maurer (2016) that enables us to bound the Rademacher complexity for our setting when the hypothesis functions are vector valued.

**Lemma 7.** *For a given set $\mathcal{X} = (x_1, \ldots, x_n) \in \mathcal{X}^n$, a class of fucntions $\mathcal{F}$, such that, $f : \mathcal{X} \rightarrow \ell_2$ and an $L$-Lispschitz function $g_i : \ell_2 \rightarrow \mathbb{R}$, we have*

$$\mathbb{E} \left[ \sup_{f \in \mathcal{F}} \sum_i \sigma_i g_i (f(x_i)) \right] \leq \sqrt{2} L \mathbb{E} \left[ \sup_{f \in \mathcal{F}} \sum_{i,j} \sigma_{ij} f_j(x_i) \right],$$

*where $\sigma_{ij}$ is an independent doubly indexed Rademacher sequence and $f_j(x_i)$ is the $j$-th component of $f(x_i)$.*

The above lemma still holds if instead of $f : \mathcal{X} \rightarrow \ell_2$ we have $f : \mathcal{X} \rightarrow \mathcal{B}^n(\rho) \subset \mathbb{R}^n$ where $\mathcal{B}^n(\rho)$ is an $\ell_2$ norm ball in $\mathbb{R}^n$ of radius $\rho$ centered at the origin.

In our case, we can take $\mathcal{X} = \mathbb{R} \times \mathbb{R}$. Note that $\{(\mathbf{x}_i, \mathbf{y}_i)\}_{i=1}^M \in \mathcal{X}^M = (\mathbb{R} \times \mathbb{R})^M$ and also $g(\boldsymbol{z}) = ||\boldsymbol{z}||_2$ which is $L$-Lipschitz with $L = 1$. Now we can apply Lemma 7 to upper bound equation 140:

$$M/\rho \cdot R(\ell \circ \tilde{\mathcal{H}}_{filt} \circ S) \leq \sqrt{2} \mathbb{E} \left[ \sup_{||\boldsymbol{h} - \boldsymbol{h}^{(0)}||_2 \leq B} \sum_{i=1}^{M} \sum_{j=1}^{n} \sigma_{ij} \cdot (\sum_{k=0}^{K-1} h_k S^k \mathbf{x}_i - \mathbf{y}_i) \right] \quad (141)$$

$$= \sqrt{2} \mathop{\mathbb{E}}_{\boldsymbol{\sigma}_i \sim \{\pm 1\}^n} \left[ \sup_{||\boldsymbol{h} - \boldsymbol{h}^{(0)}||_2 \leq B} \sum_{i=1}^{M} \langle \boldsymbol{\sigma}_i, \sum_{k=0}^{K-1} h_k S^k \mathbf{x}_i - \mathbf{y}_i \rangle \right] \quad (142)$$

Writing $\boldsymbol{h} = \boldsymbol{h}^{(0)} + \Delta \boldsymbol{h}$, we have

$$M/\rho \cdot R(\ell \circ \tilde{\mathcal{H}}_{filt} \circ S) \leq \sqrt{2} \mathop{\mathbb{E}}_{\boldsymbol{\sigma}_i \sim \{\pm 1\}^n} \left[ \sup_{||\Delta \boldsymbol{h}||_2 \leq B} \sum_{i=1}^{M} \langle \boldsymbol{\sigma}_i, \sum_{k=0}^{K-1} (h_k^{(0)} + \Delta h_k) S^k \mathbf{x}_i - \mathbf{y}_i \rangle \right] \quad (143)$$

$$\leq \sqrt{2} \mathop{\mathbb{E}}_{\boldsymbol{\sigma}_i \sim \{\pm 1\}^n} \left[ \sup_{||\Delta \boldsymbol{h}||_2 \leq B} \sum_{i=1}^{M} \langle \boldsymbol{\sigma}_i, \sum_{k=0}^{K-1} h_k^{(0)} S^k \mathbf{x}_i - \mathbf{y}_i \rangle \right]$$

$$+ \sqrt{2} \mathop{\mathbb{E}}_{\boldsymbol{\sigma}_i \sim \{\pm 1\}^n} \left[ \sup_{||\Delta \boldsymbol{h}||_2 \leq B} \sum_{i=1}^{M} \langle \boldsymbol{\sigma}_i, \sum_{k=0}^{K-1} \Delta h_k S^k \mathbf{x}_i \rangle \right] \quad (144)$$

In equation 144, the first term does not depend on $\Delta \boldsymbol{h}$ and hence, we can remove the supremum, i.e.,

$$\mathop{\mathbb{E}}_{\boldsymbol{\sigma}_i \sim \{\pm 1\}^n} \left[ \sup_{||\Delta \boldsymbol{h}||_2 \leq B} \sum_{i=1}^{M} \langle \boldsymbol{\sigma}_i, \sum_{k=0}^{K-1} h_k^{(0)} S^k \mathbf{x}_i - \mathbf{y}_i \rangle \right] = \mathop{\mathbb{E}}_{\boldsymbol{\sigma}_i \sim \{\pm 1\}^n} \left[ \sum_{i=1}^{M} \langle \boldsymbol{\sigma}_i, \sum_{k=0}^{K-1} h_k^{(0)} S^k \mathbf{x}_i - \mathbf{y}_i \rangle \right] \quad (145)$$

Further, using the linearity of expectations and inner products, we have

$$\sum_{i=1}^{M} \sum_{k=0}^{K-1} \mathop{\mathbb{E}}_{\boldsymbol{\sigma}_i \sim \{\pm 1\}^n} \left[ \langle \boldsymbol{\sigma}_i, h_k^{(0)} S^k \mathbf{x}_i - \mathbf{y}_i \rangle \right] = \sum_{i=1}^{M} \sum_{k=0}^{K-1} \sum_{j=1}^{n} (h_k^{(0)} S^k \mathbf{x}_i - \mathbf{y}_i)_j \mathop{\mathbb{E}}_{\boldsymbol{\sigma}_i \sim \{\pm 1\}^n} [(\boldsymbol{\sigma}_i)_j] = 0 \quad (146)$$

Thus, we can restate equation 144 as

$$M/\rho \cdot R(\ell \circ \tilde{\mathcal{H}}_{filt} \circ S) \leq \sqrt{2} \mathop{\mathbb{E}}_{\boldsymbol{\sigma}_i \sim \{\pm 1\}^n} \left[ \sup_{||\Delta \boldsymbol{h}||_2 \leq B} \sum_{i=1}^{M} \langle \boldsymbol{\sigma}_i, \sum_{k=0}^{K-1} \Delta h_k S^k \mathbf{x}_i \rangle \right] \quad (147)$$

Focusing on the supremum in equation 147, we have

$$\sup_{||\Delta \boldsymbol{h}||_2 \leq B} \sum_{i=1}^{M} \langle \boldsymbol{\sigma}_i, \sum_{k=0}^{K-1} \Delta h_k S^k \mathbf{x}_i \rangle = \sup_{||\Delta \boldsymbol{h}||_2 \leq B} \sum_{i=1}^{M} \sum_{k=0}^{K-1} \langle \boldsymbol{\sigma}_i, \Delta h_k S^k \mathbf{x}_i \rangle \quad (148)$$

$$= \sup_{||\Delta \boldsymbol{h}||_2 \leq B} \sum_{k=0}^{K-1} \Delta h_k \left( \sum_{i=1}^{M} \langle \boldsymbol{\sigma}_i, S^k \mathbf{x}_i \rangle \right) \quad (149)$$

We introduce the notation $a_k \triangleq \sum_{i=1}^{M} \langle \boldsymbol{\sigma}_i, S^k \mathbf{x}_i \rangle$. Note that $a_k$ doesn't depend on $\Delta \boldsymbol{h}$. Hence,

$$\sup_{||\Delta \boldsymbol{h}||_2 \leq B} \sum_{k=0}^{K-1} \Delta h_k \left( \sum_{i=1}^{M} \langle \boldsymbol{\sigma}_i, S^k \mathbf{x}_i \rangle \right) = \sup_{||\Delta \boldsymbol{h}||_2 \leq B} \sum_{k=0}^{K-1} \Delta h_k a_k = \sup_{||\Delta \boldsymbol{h}||_2 \leq B} \langle \Delta \boldsymbol{h}, \boldsymbol{a} \rangle = B \cdot ||\boldsymbol{a}||_2 \quad (150)$$

Inserting equation 150 into equation 147, we have

$$M/\rho \cdot R(\ell \circ \tilde{\mathcal{H}}_{filt} \circ S) \leq \sqrt{2} \mathop{\mathbb{E}}_{\boldsymbol{\sigma}_i \sim \{\pm 1\}^n} [B \cdot ||\boldsymbol{a}||_2] = \sqrt{2} B \mathop{\mathbb{E}}_{\boldsymbol{\sigma}_i \sim \{\pm 1\}^n} \left[ \sqrt{\sum_{k=0}^{K-1} \left( \sum_{i=1}^{M} \langle \boldsymbol{\sigma}_i, S^k \mathbf{x}_i \rangle \right)^2} \right] \quad (151)$$

Further, using Jensen's inequality in equation 151, we get

$$M/\rho \cdot R(\ell \circ \tilde{\mathcal{H}}_{filt} \circ S) \leq \sqrt{2} B \sqrt{\sum_{k=0}^{K-1} \mathop{\mathbb{E}}_{\boldsymbol{\sigma}_i \sim \{\pm 1\}^n} \left[ \left( \sum_{i=1}^{M} \langle \boldsymbol{\sigma}_i, S^k \mathbf{x}_i \rangle \right)^2 \right]} \quad (152)$$

$$= \sqrt{2} B \sqrt{\sum_{k=0}^{K-1} \mathop{\mathbb{E}}_{\boldsymbol{\sigma}_i \sim \{\pm 1\}^n} \left[ \left( \sum_{i=1}^{M} \sum_{j=1}^{n} (\boldsymbol{\sigma}_i)_j (S^k \mathbf{x}_i)_j \right)^2 \right]} \quad (153)$$

$$= \sqrt{2} B \sqrt{\sum_{k=0}^{K-1} \mathop{\mathbb{E}}_{\boldsymbol{\sigma}_i \sim \{\pm 1\}^n} \left[ \sum_{i=1}^{M} \sum_{j=1}^{n} ((\boldsymbol{\sigma}_i)_j)^2 ((S^k \mathbf{x}_i)_j)^2 \right]} \quad (154)$$

$$= \sqrt{2} B \sqrt{\sum_{k=0}^{K-1} \sum_{i=1}^{M} \sum_{j=1}^{n} ((S^k \mathbf{x}_i)_j)^2} \quad (155)$$

$$= \sqrt{2} B \sqrt{\sum_{i=1}^{M} \sum_{k=0}^{K-1} ||S^k \mathbf{x}_i||_2^2} \quad (156)$$

In the above set of equations from equation 154 to equation 155, we have used the fact that $(\boldsymbol{\sigma}_i)_j$ are independent and therefore:

$$\mathbb{E}\left[(\boldsymbol{\sigma}_i)_j (\boldsymbol{\sigma}_k)_\ell\right] = \begin{cases} 1 & \text{if } i = k, \ j = l \\ 0 & o.w. \end{cases} \quad (157)$$

Finally, we provide an upper bound on the Rademacher complexity that is proportional to $1/\sqrt{M}$. From equation 156:

$$M/\rho \cdot R(\ell \circ \tilde{\mathcal{H}}_{filt} \circ S) \leq \sqrt{2} B \sqrt{\sum_{i=1}^{M} \sum_{k=0}^{K-1} ||S^k \mathbf{x}_i||_2^2} \leq \sqrt{2} B \sqrt{MK \max_{k,i} ||S^k \mathbf{x}_i||_2^2} \quad (158)$$

$$\Rightarrow R(\ell \circ \tilde{\mathcal{H}}_{filt} \circ S) \leq B\rho \sqrt{\frac{2K \max_{k,i} ||S^k \mathbf{x}_i||_2^2}{M}} \quad (159)$$

We thus conclude the proof. $\qquad \square$

### D.10   PROOF OF LEMMA 10

*Proof.* Our focus is on the setting where the NTK does not change during training. In such a case, the "movement" of the parameters (i.e., how much the vector of parameters changes) during gradient descent is directly related to the NTK.

Recall that the NTK matrix $\tilde{\boldsymbol{\Theta}}(\boldsymbol{h}) \in \mathbb{R}^{nM \times nM}$ consists of $M^2$ blocks, each of which is an $n \times n$ matrix $\Theta(\mathbf{x}_i, \mathbf{x}_j) = \mathrm{J}_{\boldsymbol{f}_{\mathbf{x}_i}}(\boldsymbol{h}^{(t)})\big(\mathrm{J}_{\boldsymbol{f}_{\mathbf{x}_j}}(\boldsymbol{h}^{(t)})\big)^{\mathsf{T}}$ (see equation 7). Hence, assuming that the NTK does not change during training is equivalent to assuming that the Jacobian matrices $\mathrm{J}_{\boldsymbol{f}_{\mathbf{x}_i}}(\boldsymbol{h}^{(t)})$ are constant. Also, for ease of notation, we define the big Jacobian matrix $\tilde{\mathbf{J}} \in \mathbb{R}^{nM \times p}$ that consists of all the Jacobian matrices $\mathrm{J}_{\boldsymbol{f}_{\mathbf{x}_i}}$ stacked on top of one another. Therefore, we have

$$\tilde{\boldsymbol{\Theta}}(\boldsymbol{h}) = (\tilde{\mathbf{J}}(\boldsymbol{h}))(\tilde{\mathbf{J}}(\boldsymbol{h}))^{\mathsf{T}} \tag{160}$$

Again, we note that in general both $\tilde{\mathbf{J}}$ (and consequently, $\tilde{\boldsymbol{\Theta}}$) can depend on the parameters $\boldsymbol{h}$ but we are considering cases where they do not.[5]

Similar to section 2, we denote the vector of all parameters of the predictor as $\boldsymbol{h} \in \mathbb{R}^p$. For the $t$-th step of gradient descent we can write:

$$\boldsymbol{h}^{(t+1)} = \boldsymbol{h}^{(t)} - \eta \cdot \nabla\Phi(\boldsymbol{h}^{(t)}) \tag{161}$$

Also, we can write the gradient in terms of the Jacobian evaluated at different input points as follows

$$\nabla\Phi(\boldsymbol{h}^{(t)}) = \sum_{j=1}^{M} \big(\mathrm{J}_{\boldsymbol{f}_{\mathbf{x}_j}}\big)^{\mathsf{T}} \cdot (\boldsymbol{f}_{\mathbf{x}_j}(\boldsymbol{h}^{(t)}) - \mathbf{y}_j) \tag{162}$$

Further, using the linearization of $\boldsymbol{f}_{\mathbf{x}_j}$ around the initialization $\boldsymbol{h}^{(0)}$, we have

$$\boldsymbol{f}_{\mathbf{x}_j}(\boldsymbol{h}^{(t)}) - \mathbf{y}_j = \boldsymbol{f}_{\mathbf{x}_j}(\boldsymbol{h}^{(0)}) + \mathrm{J}_{\boldsymbol{f}_{\mathbf{x}_j}} \cdot (\boldsymbol{h}^{(t)} - \boldsymbol{h}^{(0)}) - \mathbf{y}_j \tag{163}$$

Since $\mathrm{J}_{\boldsymbol{f}_{\mathbf{x}_j}}$ is assumed to be constant, we can further replace $\boldsymbol{f}_{\mathbf{x}_j}(\boldsymbol{h}^{(0)})$ with a linearization around $\boldsymbol{h} = 0$, such that, we have

$$\boldsymbol{f}_{\mathbf{x}_j}(\boldsymbol{h}^{(0)}) = \boldsymbol{f}_{\mathbf{x}_j}(0) + \mathrm{J}_{\boldsymbol{f}_{\mathbf{x}_j}}\boldsymbol{h}^{(0)} = \mathrm{J}_{\boldsymbol{f}_{\mathbf{x}_j}}\boldsymbol{h}^{(0)} \tag{164}$$

In equation 164, we used the fact that the output of the predictor is zero when all its parameters are set to zero. By inserting equation 164 into equation 163, we get

$$\boldsymbol{f}_{\mathbf{x}_j}(\boldsymbol{h}^{(t)}) - \mathbf{y}_j = \mathrm{J}_{\boldsymbol{f}_{\mathbf{x}_j}}\boldsymbol{h}^{(t)} - \mathbf{y}_j \tag{165}$$

And inserting equation 165 into equation 162, we have

$$\nabla\Phi(\boldsymbol{h}^{(t)}) = \sum_{j=1}^{M} \big(\mathrm{J}_{\boldsymbol{f}_{\mathbf{x}_j}}\big)^{\mathsf{T}} \cdot (\mathrm{J}_{\boldsymbol{f}_{\mathbf{x}_j}}\boldsymbol{h}^{(t)} - \mathbf{y}_j) \tag{166}$$

Replacing the gradient of the loss $\nabla\Phi(\boldsymbol{h}^{(t)})$ in equation 161 by equation 166 leads to

$$\boldsymbol{h}^{(t+1)} = \boldsymbol{h}^{(t)} - \eta \cdot \sum_{j=1}^{M} \big(\mathrm{J}_{\boldsymbol{f}_{\mathbf{x}_j}}\big)^{\mathsf{T}} \cdot (\mathrm{J}_{\boldsymbol{f}_{\mathbf{x}_j}}\boldsymbol{h}^{(t)} - \mathbf{y}_j) \,, \tag{167}$$

which implies

$$\mathrm{J}_{\boldsymbol{f}_{\mathbf{x}_i}}\boldsymbol{h}^{(t+1)} = \mathrm{J}_{\boldsymbol{f}_{\mathbf{x}_i}}\boldsymbol{h}^{(t)} - \eta \cdot \sum_{j=1}^{M} \mathrm{J}_{\boldsymbol{f}_{\mathbf{x}_i}}\big(\mathrm{J}_{\boldsymbol{f}_{\mathbf{x}_j}}\big)^{\mathsf{T}} \cdot (\mathrm{J}_{\boldsymbol{f}_{\mathbf{x}_j}}\boldsymbol{h}^{(t)} - \mathbf{y}_j) \tag{168}$$

---

[5]This is the case for any linear predictor like the graph filter as we saw in section equation 3 and it is also the case for some neural networks with infinite width in every layer. See Appendix F and Liu et al. (2020) for details.

Introducing the notation $\boldsymbol{r}_i^{(t)}$ to denote $\mathrm{J}_{\boldsymbol{f}_{\mathbf{x}_i}} \boldsymbol{h}^{(t)}$, we can rewrite equation 168 as

$$\boldsymbol{r}_i^{(t+1)} = \boldsymbol{r}_i^{(t)} - \eta \cdot \sum_{j=1}^{M} \Theta(\mathbf{x}_i, \mathbf{x}_j) \cdot (\boldsymbol{r}_j^{(t)} - \mathbf{y}_j) \tag{169}$$

Similar to the procedure in equation 5, we can stack all the vectors $\boldsymbol{r}_i^{(t)}$ together in one tall vector $\tilde{\boldsymbol{r}}(t) \in \mathbb{R}^{nM \times 1}$ and rewrite equation 169 in the vectorized form as follows

$$\tilde{\boldsymbol{r}}(t+1) = \tilde{\boldsymbol{r}}(t) - \eta \cdot \tilde{\boldsymbol{\Theta}}(\tilde{\boldsymbol{r}}(t) - \tilde{\mathbf{y}}) \tag{170}$$

$$\Rightarrow \tilde{\boldsymbol{r}}(t+1) - \tilde{\mathbf{y}} = \tilde{\boldsymbol{r}}(t) - \tilde{\mathbf{y}} - \eta \cdot \tilde{\boldsymbol{\Theta}}(\tilde{\boldsymbol{r}}(t) - \tilde{\mathbf{y}}) \tag{171}$$

$$\Rightarrow \tilde{\boldsymbol{r}}(t) - \tilde{\mathbf{y}} = (I - \eta \cdot \tilde{\boldsymbol{\Theta}})^t (\tilde{\boldsymbol{r}}(0) - \tilde{\mathbf{y}}) \tag{172}$$

Now, writing equation 167 in terms of the big Jacobian matrix $\tilde{\mathbf{J}}$

$$\boldsymbol{h}^{(t+1)} = \boldsymbol{h}^{(t)} - \eta \cdot \tilde{\mathbf{J}}^{\mathsf{T}}(\tilde{\boldsymbol{r}}(t) - \tilde{\mathbf{y}}) \tag{173}$$

and replacing $(\tilde{\boldsymbol{r}}(t) - \tilde{\mathbf{y}})$ by the quantity from equation 172 we get

$$\Rightarrow \boldsymbol{h}^{(t+1)} = \boldsymbol{h}^{(t)} + \eta \cdot \tilde{\mathbf{J}}^{\mathsf{T}}(I - \eta \cdot \tilde{\boldsymbol{\Theta}})^t \tilde{\mathbf{y}} - \eta \cdot \tilde{\mathbf{J}}^{\mathsf{T}}(I - \eta \cdot \tilde{\boldsymbol{\Theta}})^t \tilde{\boldsymbol{r}}(0) \tag{174}$$

$$\Rightarrow \boldsymbol{h}^{(\infty)} - \boldsymbol{h}^{(0)} = \eta \sum_{t=0}^{\infty} \tilde{\mathbf{J}}^{\mathsf{T}}(I - \eta \cdot \tilde{\boldsymbol{\Theta}})^t \tilde{\mathbf{y}} - \eta \sum_{t=0}^{\infty} \tilde{\mathbf{J}}^{\mathsf{T}}(I - \eta \cdot \tilde{\boldsymbol{\Theta}})^t \tilde{\boldsymbol{r}}(0) \tag{175}$$

We write the eigen-decomposition of $\tilde{\boldsymbol{\Theta}}$ (assuming it has rank $r$) as

$$\tilde{\boldsymbol{\Theta}} = \sum_{\ell=1}^{r} \lambda_\ell \boldsymbol{v}_\ell \boldsymbol{v}_\ell^{\mathsf{T}}$$

which implies the following eigen-decomposition for $(I - \eta \cdot \tilde{\boldsymbol{\Theta}})$.

$$I - \eta \cdot \tilde{\boldsymbol{\Theta}} \;=\; \sum_{\ell=1}^{r}(1 - \eta\lambda_\ell)\boldsymbol{v}_\ell \boldsymbol{v}_\ell^{\mathsf{T}} + \sum_{\ell=r+1}^{nM} 1 \cdot \boldsymbol{v}_\ell \boldsymbol{v}_\ell^{\mathsf{T}} \tag{176}$$

Keeping in mind that $\lambda_\ell = 0$ for $\ell > r$ we use the eigen-decomposition in equation 176 to write

$$\Rightarrow \boldsymbol{h}^{(\infty)} - \boldsymbol{h}^{(0)} = \eta \sum_{t=0}^{\infty} \sum_{\ell=1}^{nM} \tilde{\mathbf{J}}^{\mathsf{T}}(1 - \eta\lambda_\ell)^t \boldsymbol{v}_\ell \boldsymbol{v}_\ell^{\mathsf{T}} \tilde{\mathbf{y}} - \eta \sum_{t=0}^{\infty} \sum_{\ell=1}^{nM} \tilde{\mathbf{J}}^{\mathsf{T}}(1 - \eta\lambda_\ell)^t \boldsymbol{v}_\ell \boldsymbol{v}_\ell^{\mathsf{T}} \tilde{\boldsymbol{r}}(0) \tag{177}$$

Note that

$$\text{for } j > r, \; \tilde{\boldsymbol{\Theta}}\boldsymbol{v}_j = \sum_{\ell=1}^{r} \lambda_\ell \boldsymbol{v}_\ell \boldsymbol{v}_\ell^{\mathsf{T}} \boldsymbol{v}_j = 0 \;. \tag{178}$$

Hence, from equation 160 and equation 178, we have for $j > r$:

$$\tilde{\mathbf{J}} \cdot \tilde{\mathbf{J}}^{\mathsf{T}} \boldsymbol{v}_j = 0 \tag{179}$$

Assuming that $\tilde{\mathbf{J}} \in \mathbb{R}^{nM \times p}$ is full-column rank, equation 179 implies that

$$\text{for } j > r, \; \tilde{\mathbf{J}}^{\mathsf{T}} \boldsymbol{v}_j = 0 \tag{180}$$

Hence, we can write equation 177 as

$$\boldsymbol{h}^{(\infty)} - \boldsymbol{h}^{(0)} = \eta \sum_{t=0}^{\infty} \sum_{\ell=1}^{r} \tilde{\mathbf{J}}^{\mathsf{T}}(1 - \eta\lambda_\ell)^t \boldsymbol{v}_\ell \boldsymbol{v}_\ell^{\mathsf{T}} \tilde{\mathbf{y}} - \eta \sum_{t=0}^{\infty} \sum_{\ell=1}^{r} \tilde{\mathbf{J}}^{\mathsf{T}}(1 - \eta\lambda_\ell)^t \boldsymbol{v}_\ell \boldsymbol{v}_\ell^{\mathsf{T}} \tilde{\boldsymbol{r}}(0) \tag{181}$$

$$= \eta \sum_{\ell=1}^{r} \tilde{\mathbf{J}}^{\mathsf{T}} \frac{1}{\eta\lambda_\ell} \boldsymbol{v}_\ell \boldsymbol{v}_\ell^{\mathsf{T}} \tilde{\mathbf{y}} - \eta \sum_{\ell=1}^{r} \tilde{\mathbf{J}}^{\mathsf{T}} \frac{1}{\eta\lambda_\ell} \boldsymbol{v}_\ell \boldsymbol{v}_\ell^{\mathsf{T}} \tilde{\boldsymbol{r}}(0) \tag{182}$$

$$= \tilde{\mathbf{J}}^{\mathsf{T}} \tilde{\boldsymbol{\Theta}}^{\dagger} \tilde{\mathbf{y}} - \tilde{\mathbf{J}}^{\mathsf{T}} \tilde{\boldsymbol{\Theta}}^{\dagger} \tilde{\boldsymbol{r}}(0) \tag{183}$$

Now, similar to what we saw in the proof of Theorem 1 (equation 47 - equation 50), if we choose the parameter controlling the magnitude of initialization to be $\kappa = \mathcal{O}(\varepsilon\sqrt{\frac{\delta}{nM}})$, then with probability at least $1 - \delta$ we have $\tilde{\boldsymbol{r}}(0) = \mathcal{O}\left(\varepsilon\sqrt{\frac{\delta}{nM}}\right)$ and $||\tilde{\boldsymbol{r}}(0)||_2 = \mathcal{O}(\varepsilon)$.

$$||\boldsymbol{h}^{(\infty)} - \boldsymbol{h}^{(0)}||_2 = \sqrt{\tilde{\mathbf{y}}^\mathsf{T}\tilde{\boldsymbol{\Theta}}^\dagger \tilde{\mathbf{J}} \cdot \tilde{\mathbf{J}}^\mathsf{T}\tilde{\boldsymbol{\Theta}}^\dagger \tilde{\mathbf{y}}} \pm \mathcal{O}(\varepsilon) \tag{184}$$

$$= \sqrt{\tilde{\mathbf{y}}^\mathsf{T}\tilde{\boldsymbol{\Theta}}^\dagger \tilde{\boldsymbol{\Theta}}\tilde{\boldsymbol{\Theta}}^\dagger \tilde{\mathbf{y}}} \pm \mathcal{O}(\varepsilon) \tag{185}$$

Finally, since $\tilde{\boldsymbol{\Theta}}^\dagger$ acts as a weak inverse, we get

$$||\boldsymbol{h}^{(\infty)} - \boldsymbol{h}^{(0)}||_2 = \sqrt{\tilde{\mathbf{y}}^\mathsf{T}\tilde{\boldsymbol{\Theta}}^\dagger \tilde{\mathbf{y}}} \pm \mathcal{O}(\varepsilon) \tag{186}$$

Thus, we've concluded the proof by showing that the "movement" of the parameters, $||\boldsymbol{h}^{(\infty)} - \boldsymbol{h}^{(0)}||_2$, is directly related to the NTK. (More precisely, the pseudo-inverse of the NTK $\tilde{\boldsymbol{\Theta}}^\dagger$) $\qquad\square$

## E  ALIGNMENT, THE NTK AND GENERALIZATION

In this section, we will briefly go over how alignment relates to generalization. We start by providing a few definitions from Shalev-Shwartz & Ben-David (2014). Consider a function class $\mathcal{F}$ and a function $f \in \mathcal{F}$ and a distribution $\mathcal{D}$ from which we sample data points $z$. Also consider a set $\mathcal{S} = \{z_1, z_2, \cdots, z_M\}$ of M samples, sampled i.i.d from $\mathcal{D}$. The population average $L_\mathcal{D}(f)$ and sample average $L_\mathcal{S}(f)$ of the function $f$ are then defined as follows:

$$L_\mathcal{D}(f) \triangleq \mathbb{E}_{z\sim\mathcal{D}}[f(z)], \quad L_\mathcal{S}(f) \triangleq \frac{1}{M}\sum_{i=1}^{M} f(z_i) \tag{187}$$

Also, the Rademacher complexity of class $\mathcal{F}$ with respect to $\mathcal{S}$ is defined as:

$$R(\mathcal{F} \circ \mathcal{S}) \triangleq \frac{1}{M}\mathop{\mathbb{E}}_{\boldsymbol{\sigma}\sim\{\pm1\}^M}\left[\sup_{f\in\mathcal{F}}\sum_{i=1}^{M}\sigma_i f\left(z_i\right)\right] \tag{188}$$

where each element of the random vector $\boldsymbol{\sigma}$ is either 1 or -1 with equal probability.

Now consider class of functions (called hypotheses here) $\mathcal{H}$ and some loss function $\ell(h, z)$. In an empirical risk minimization problem, our goal is to find the function $h^* \in \mathcal{H}$ that minimizes the sample loss $L_\mathcal{S}(\ell(h))$:

$$h^* = \mathrm{ERM}_\mathcal{H}(\mathcal{S}) = \arg\min_{h\in\mathcal{H}} L_\mathcal{S}(\ell(h)) = \arg\min_{h\in\mathcal{H}} \frac{1}{M}\sum_{i=1}^{M} \ell(h, z_i) \tag{189}$$

But minimizing the sample loss does not necessarily lead to a small population loss $L_\mathcal{D}(\ell(h)) = \mathbb{E}_{z\sim\mathcal{D}}[\ell(z, h)]$. The gap between these two losses is called the generalization error. We present the following lemma which upper bounds the generalization error for a class of functions $\mathcal{H}$ based on the Rademacher complexity of the class:

**Lemma 8** (Theorem 26.5 from Shalev-Shwartz & Ben-David (2014)). *Assume that for all $z$ and $h \in \mathcal{H}$ we have that $|\ell(h, z)| \leq c$. Then with probability of at least $1 - \delta$, for all $h \in \mathcal{H}$,*

$$L_\mathcal{D}(\ell(h)) - L_\mathcal{S}(\ell(h)) \leq 2R(\ell \circ \mathcal{H} \circ \mathcal{S}) + 4c\sqrt{\frac{2\ln(4/\delta)}{M}} \tag{190}$$

*where $\ell(h, z)$ is some loss function. In particular, this holds for $h = \mathrm{ERM}_\mathcal{H}(\mathcal{S})$ which is the solution of the empirical risk minimization problem.*

In the case of the multivariate regression problem that we consider in this paper, each point $z$ is an input output pair of vectors $(\mathbf{x}_i, \mathbf{y}_i)$ and the loss function is quadratic. Also the set $\mathcal{H}$ in our case

corresponds to the set of all possible predictors or the set of possible parameter vectors $\boldsymbol{h} \in \mathbb{R}^p$ and the set $\mathcal{S}$ is our training set.

From Lemma equation 8, we can surmise that if the complexity of the function class $\ell \circ \mathcal{H}$ is sufficiently small, our trained model generalizes to unseen input output pairs sampled from the same distribution as the training set. Henceforth in this section we will only be considering the hypothesis class of graph filters with $K$ filter taps:

$$\mathcal{H}_{filt} \triangleq \left\{ \boldsymbol{f}_{\mathbf{x}}(\boldsymbol{h}) = \sum_{k=0}^{K-1} h_k S^k \mathbf{x} \,\middle|\, \boldsymbol{h} \in \mathbb{R}^K \right\} \tag{191}$$

First, in the next lemma, we will be considering the subset $\tilde{\mathcal{H}}_{filt} \subset \mathcal{H}_{filt}$, for which the vector of filter coefficients is close to some initialization $\boldsymbol{h}^{(0)}$. Formally:

$$\tilde{\mathcal{H}}_{filt}(B) \triangleq \left\{ \boldsymbol{f}_{\mathbf{x}}(\boldsymbol{h}) = \sum_{k=0}^{K-1} h_k S^k \mathbf{x} \,\middle|\, \boldsymbol{h} \in \mathbb{R}^K, \, ||\boldsymbol{h} - \boldsymbol{h}^{(0)}||_2 \leq B \right\} \tag{192}$$

The following lemma gives us an upper bound for the complexity of $\tilde{\mathcal{H}}_{filt}$:

**Lemma 9.** *Consider the class of hypotheses* $\tilde{\mathcal{H}}_{filt}(B)$ *defined in equation 192. Assuming that* $||\sum_{k=0}^{K-1} h_k S^k \mathbf{x}_i - \mathbf{y}_i||_2 \leq \rho, \, \forall (\mathbf{x}_i, \mathbf{y}_i) \in S, \boldsymbol{h} \in \tilde{\mathcal{H}}_{filt}$, *the Rademacher complexity of* $\tilde{\mathcal{H}}_{filt}$ *can be upper bounded as follows:*

$$R(\ell \circ \tilde{\mathcal{H}}_{filt} \circ \mathcal{S}) \leq B\rho \sqrt{\frac{2K \max_{k,i} ||S^k \mathbf{x}_i||_2^2}{M}} \tag{193}$$

*where the loss function is quadratic* $\ell(\boldsymbol{h}, z = (\mathbf{x}_i, \mathbf{y}_i)) = ||\sum_{k=0}^{K-1} h_k S^k \mathbf{x}_i - \mathbf{y}_i||_2^2$.

For proof of Lemma 9 see subsection D.9. In order to meaningfully utilize Lemma 9, we next bound the movement of the vector of parameters (in this case the filter coefficients) from initialization using a straightforward extension of a result from Arora et al. (2019).

**Lemma 10.** *Consider the prediction task defined in section 2. If the NTK* $\tilde{\Theta}(\boldsymbol{h}^{(t)})$ *is constant during training and* $\kappa = \mathcal{O}(\varepsilon \sqrt{\frac{\delta_1}{nM}})$, *then with probability at least* $1 - \delta_1$, *The change in the vector of parameters,* $\boldsymbol{h}$, *after infinitely many steps of gradient descent is related to the NTK matrix,* $\tilde{\Theta}$, *as follows:*

$$||\boldsymbol{h}^{(\infty)} - \boldsymbol{h}^{(0)}||_2^2 = \sqrt{\tilde{\mathbf{y}}^\mathsf{T} \tilde{\Theta}^\dagger \tilde{\mathbf{y}}} \pm \mathcal{O}(\varepsilon) \, ^{[6]} \tag{194}$$

For proof of Lemma 10 see subsection D.10. Motivated by the above lemma we define the class of functions $\mathcal{H}_{trained}$ which includes the family of graph filters where the coefficients have been initialized randomly to some vector $\boldsymbol{h}^{(0)}$ and subsequently updated using an arbitrary number of gradient descent steps:

$$\mathcal{H}_{trained} \triangleq \left\{ \boldsymbol{f}_{\mathbf{x}}(\boldsymbol{h}) = \sum_{k=0}^{K-1} h_k S^k \mathbf{x} \,\middle|\, \begin{array}{l} \boldsymbol{h} \in \mathbb{R}^K \text{ initialized to } \boldsymbol{h}^{(0)} \text{ with } \kappa = \mathcal{O}\left(\varepsilon \sqrt{\frac{\delta_1}{nM}}\right), \\ \text{then updated using Gradient Descent} \end{array} \right\} \tag{195}$$

Finally, putting the previous Lemmas 8, 9 and 10 together we get the result that relates generalization to the NTK:

**Theorem 4.** *Consider the prediction task defined in section 2 and the hypothesis class* $\mathcal{H}_{trained}$ *of graph filters trained using Gradient Descent. Under the assumption that* $|\ell(\boldsymbol{h}, z)| = ||\sum_{k=0}^{K-1} h_k S^k \mathbf{x}_i - \mathbf{y}_i||_2^2 \leq \rho^2, \, \forall (\mathbf{x}_i, \mathbf{y}_i) \in S, \boldsymbol{h} \in \mathcal{H}_{trained}$, *with probability of at least* $1 - \delta$, *for all* $\boldsymbol{h} \in \mathcal{H}_{trained}$, *we have*

$$L_{\mathcal{D}}(\ell(h)) - L_S(\ell(h)) \leq 2\rho \sqrt{\frac{2K \cdot (\max_{k,i} ||S^k \mathbf{x}_i||_2^2) \cdot (\tilde{\mathbf{y}}^\mathsf{T} \tilde{\Theta}^\dagger \tilde{\mathbf{y}})}{M}} + 4\rho^2 \sqrt{\frac{2 \ln(4/\delta_2)}{M}} \tag{196}$$

---

[6]If $\tilde{\Theta}$ is full rank then this becomes $\sqrt{\tilde{\mathbf{y}}^\mathsf{T} \tilde{\Theta}^{-1} \tilde{\mathbf{y}}}$

where $\delta = \delta_1 + \delta_2$. *In particular, equation 196 holds for $h = \mathrm{ERM}_{\mathcal{H}}(S)$ which is the solution of the empirical risk minimization problem.*

Theorem 4 reveals that the upper bound on generalization error is proportional to the term $\sqrt{\tilde{\mathbf{y}}^{\mathsf{T}} \tilde{\mathbf{\Theta}}^{\dagger} \tilde{\mathbf{y}}}$. While maximizing the alignment i.e., $\mathcal{A} = \tilde{\mathbf{y}}^{\mathsf{T}} \tilde{\mathbf{\Theta}} \tilde{\mathbf{y}}$ and minimizing the term $\tilde{\mathbf{y}}^{\mathsf{T}} \tilde{\mathbf{\Theta}}^{\dagger} \tilde{\mathbf{y}}$ are clearly not identical objectives, there is a close relationship between the two as formalized in the following result (Theorem 2 from Wang et al. (2022a)):

$$\frac{\tilde{\mathbf{y}}^{\mathsf{T}} \tilde{\mathbf{y}}}{\mathcal{A}(X, Y, S)} \leq \tilde{\mathbf{y}}^{\mathsf{T}} \tilde{\mathbf{\Theta}}^{\dagger} \tilde{\mathbf{y}} \leq \frac{\lambda_{max}(\tilde{\mathbf{\Theta}})}{\lambda_{min}(\tilde{\mathbf{\Theta}})} \frac{\tilde{\mathbf{y}}^{\mathsf{T}} \tilde{\mathbf{y}}}{\mathcal{A}(X, Y, S)} \tag{197}$$

[7] equation 197 tells us that for a class of predictors where the ratio of the largest and smallest eigenvalues of the NTK i.e. $\frac{\lambda_{max}(\tilde{\mathbf{\Theta}})}{\lambda_{min}(\tilde{\mathbf{\Theta}})}$ is more or less constant between different predictors, the generalization error bound given in Theorem 4 is governed by $1/\mathcal{A}(X, Y, S)$. This means within such a class, predictors with larger alignment are likely to have a smaller generalization error. The question remains that for the classes of predictors discussed within the paper, namely Graph filters and two-layer GNNs, how much does $\frac{\lambda_{max}(\tilde{\mathbf{\Theta}})}{\lambda_{min}(\tilde{\mathbf{\Theta}})}$ vary between predictors. This is an avenue for potential future work.

# F WHY THE NTK IS CONSTANT FOR A TWO-LAYER GNN WITH INFINITE WIDTH

We start this section with the following proposition from Liu et al. (2020):

**Proposition 3** ((Small Hessian norm implies small change in tangent kernel)). *Given a point $\mathbf{w}_0 \in \mathbb{R}^p$ and a ball $B(\mathbf{w}_0, R) := \{\mathbf{w} \in \mathbb{R}^p : \|\mathbf{w} - \mathbf{w}_0\| \leq R\}$ with fixed radius $R > 0$, if the Hessian matrix satisfies $\|H(\mathbf{w})\| < \epsilon$, where $\epsilon > 0$, for all $\mathbf{w} \in B(\mathbf{w}_0, R)$, then the tangent kernel $K(\mathbf{w})$ of the model, as a function of $\mathbf{w}$, satisfies*

$$\left| K_{(\mathbf{x},\mathbf{z})}(\mathbf{w}) - K_{(\mathbf{x},\mathbf{z})}(\mathbf{w}_0) \right| = \mathcal{O}(\epsilon R), \quad \forall \mathbf{w} \in B(\mathbf{w}_0, R), \forall \mathbf{x}, \mathbf{z} \in \mathbb{R}^d$$

**Remark 4.** *It will now suffice to show that the norm of the Hessian matrix $\|H\|$ is sufficiently small if we choose our network width $F$ large enough. This means if $F$ is chosen to be large enough, the NTK will be almost constant within a ball of arbitrary fixed radius $R$ around initialization.*

Note that the above proposition is given for the case where the network output $f$ is a scalar and an element of the NTK is defined as $K_{(\mathbf{x}_i,\mathbf{x}_j)}(\mathbf{w}) := \nabla f_{\mathbf{x}_i}(\mathbf{w})^{\mathsf{T}} \nabla f_{\mathbf{x}_j}(\mathbf{w})$. Since for a GNN the output is a vector $\boldsymbol{f}_{\mathbf{x}}(\boldsymbol{h})$, we will consider the Hessian of a network with a scalar output equal to the first element of this vector which we'll call $f_1(\boldsymbol{h})$. Also for illustration purposes, we will consider a GNN where each filter only has a single coefficient (e.g. the $i$-th filter in the first layer: $G_i(S) = g_i S$ and the $j$-th filter in the second layer: $H_j(S) = h_j S$). Similar end results hold for GNNs with filters that have $K > 1$ coefficients. We now derive the different elements of the Hessian.

$$\frac{\partial^2 f_1}{\partial g_i \partial g_j} = \begin{cases} \frac{1}{\sqrt{F}} \left( h_i S \mathbf{diag} \left( \mathbf{diag} \left( \sigma'' \left( g_i S \mathbf{x} \right) \right) S \mathbf{x} \right) S \mathbf{x} \right)_1, & \text{if } i = j \\ 0, & \text{otherwise} \end{cases} \tag{198}$$

$$\frac{\partial^2 f_1}{\partial h_i \partial h_j} = 0 \tag{199}$$

$$\frac{\partial^2 f_1}{\partial g_i \partial h_j} = \begin{cases} \frac{1}{\sqrt{F}} \left( S \mathbf{diag} \left( \sigma' \left( g_i S \mathbf{x} \right) S \mathbf{x} \right) \right)_1, & \text{if } i = j \\ 0, & \text{otherwise} \end{cases} \tag{200}$$

The Hessian can be written as follows:

---

[7] Since in our case $\tilde{\mathbf{\Theta}}$ isn't full-rank, by $\lambda_{min}(\tilde{\mathbf{\Theta}})$ we mean the smallest eigenvalue of $\tilde{\mathbf{\Theta}}$ that is greater than 0.

$$H = \begin{bmatrix} H_{gg} & H_{gh} \\ (H_{gh})^{\mathsf{T}} & H_{hh} \end{bmatrix} \tag{201}$$

where each of the sub-matrices $H_{gg}, H_{gh}$ are diagonal $F \times F$ matrices and $H_{hh} = 0$.

$$||H|| = ||\begin{bmatrix} H_{gg} & 0 \\ 0 & 0 \end{bmatrix} + \begin{bmatrix} 0 & H_{gh} \\ (H_{gh})^{\mathsf{T}} & 0 \end{bmatrix}|| \leq ||\begin{bmatrix} H_{gg} & 0 \\ 0 & 0 \end{bmatrix}|| + ||\begin{bmatrix} 0 & H_{gh} \\ (H_{gh})^{\mathsf{T}} & 0 \end{bmatrix}|| \tag{202}$$

$$= \max_i |(H_{gg})_{ii}| + \max_i |(H_{gh})_{ii}| = \max_i |\left(\frac{\partial^2 f_1}{\partial^2 g_i}\right)| + \max_i |\left(\frac{\partial^2 f_1}{\partial g_i \partial h_i}\right)| \tag{203}$$

$$\Rightarrow ||H|| \leq \max_i |\left(\frac{\partial^2 f_1}{\partial^2 g_i}\right)| + \max_i |\left(\frac{\partial^2 f_1}{\partial g_i \partial h_i}\right)| \tag{204}$$

We will assume the following. The function $\sigma(.)$ is twice differentiable and the magnitude of its second derivative is at most $B_\sigma$ (e.g. tanh or Sigmoid). We also recall that our input data has been normalized such that $||\mathbf{x}||_2 \leq 1$. We will also assume $||S||_{\mathsf{op}} \leq \nu$.

$$\max_i |\left(\frac{\partial^2 f_1}{\partial^2 g_i}\right)| = \max_i |\frac{1}{\sqrt{F}} \left(h_i S \mathbf{diag}\left(\mathbf{diag}\left(\sigma''\left(g_i S \mathbf{x}\right)\right) S \mathbf{x}\right) S \mathbf{x}\right)_1| \tag{205}$$

$$\leq \max_i ||\frac{1}{\sqrt{F}} h_i S \mathbf{diag}\left(\mathbf{diag}\left(\sigma''\left(g_i S \mathbf{x}\right)\right) S \mathbf{x}\right) S \mathbf{x}||_\infty \tag{206}$$

$$\leq \max_i \frac{1}{\sqrt{F}} h_i ||S \mathbf{diag}\left(\mathbf{diag}\left(\sigma''\left(g_i S \mathbf{x}\right)\right) S \mathbf{x}\right) S||_{\mathsf{op}} ||\mathbf{x}||_2 \tag{207}$$

$$\leq \max_i \frac{1}{\sqrt{F}} h_i ||S||_{\mathsf{op}}^2 ||\mathbf{diag}\left(\mathbf{diag}\left(\sigma''\left(g_i S \mathbf{x}\right)\right) S \mathbf{x}\right)||_{\mathsf{op}} \tag{208}$$

For the rightmost term in equation 208 we can write:

$$||\mathbf{diag}\left(\mathbf{diag}\left(\sigma''\left(g_i S \mathbf{x}\right)\right) S \mathbf{x}\right)||_{\mathsf{op}} \leq ||\mathbf{diag}\left(\sigma''\left(g_i S \mathbf{x}\right)\right) S \mathbf{x}||_\infty \leq B_\sigma ||S||_{\mathsf{op}} ||\mathbf{x}||_2 \tag{209}$$

using equation 208 and equation 209 we get:

$$\max_i |\left(\frac{\partial^2 f_1}{\partial^2 g_i}\right)| \leq (\max_i h_i) \frac{1}{\sqrt{F}} B_\sigma ||S||_{\mathsf{op}}^3 = \mathcal{O}(\frac{1}{\sqrt{F}}) \tag{210}$$

We can similarly show that $\max_i |\left(\frac{\partial^2 f_1}{\partial g_i \partial h_i}\right)| = \mathcal{O}(\frac{1}{\sqrt{F}})$. using these and equation 204 we conclude:

$$||H|| = \mathcal{O}(\frac{1}{\sqrt{F}}) \tag{211}$$

The important take away is the order of $||H||$ in terms of $F$. note that even if we set aside the simplifying assumption that each filter has only a single coefficient, it still won't be too difficult to show that $||H|| = \mathcal{O}(\frac{1}{\sqrt{F}})$ since $H$ will be similarly sparse with $\mathcal{O}(K)$ number of non zero diagonals, and aside from the factor of $\frac{1}{\sqrt{F}}$, none of its elements depend on $F$. Going back to proposition 3 we conclude that for the two-layer GNN discussed in this paper, as $F \to \infty$ the NTK converges to a constant matrix.

## G  TRAINING THE FIRST LAYER

The NTK, and subsequently the alignment, for the case where we train the filter coefficients of both layers is equal to the sum of the cases where we train only the first layer and only the second layer respectively. We analyzed alignment when training the second layer in Section 3. For completeness, here we analyze the alignment when only training the first layer to show that similar results hold.

The NTK for the two-layer GNN where we randomly initialize all filter coefficients by sampling i.i.d from a Gaussian distribution and then only train the filter coefficients in the first layer, is the first term of the NTK in Proposition 2. It is re-stated here as follows.

$$\tilde{\Theta}_{GNN}^{(1)}(\boldsymbol{h}) = \frac{1}{F} \sum_{f=1}^{F} \sum_{k=0}^{K-1} \left(\mathbf{c}_{f,k}^{(1)}\right) \left(\mathbf{c}_{f,k}^{(1)}\right)^{\mathsf{T}} \tag{212}$$

where we recall from equation 23 that:

$$\mathbf{c}_{f,k}^{(1)} = H_f(\tilde{S}) \cdot \mathbf{diag}(\sigma'(G_f(\tilde{S})\tilde{\mathbf{x}})\tilde{S}^k \tilde{\mathbf{x}}) \tag{213}$$

In the asymptote of the width of the hidden layer, i.e., $F \to \infty$, we have

$$\tilde{\Theta}_{GNN}^{(1)}(\boldsymbol{h}) = \lim_{F \to \infty} \frac{1}{F} \sum_{f=1}^{F} \sum_{k=0}^{K-1} \left(H_f(\tilde{S}) \cdot \mathbf{diag}(\sigma'(G_f(\tilde{S})\tilde{\mathbf{x}})\tilde{S}^k \tilde{\mathbf{x}})\right) \left(H_f(\tilde{S}) \cdot \mathbf{diag}(\sigma'(G_f(\tilde{S})\tilde{\mathbf{x}})\tilde{S}^k \tilde{\mathbf{x}})\right)^{\mathsf{T}} \tag{214}$$

$$= \mathop{\mathbb{E}}_{\boldsymbol{g}\sim\mathcal{N}(0,I),\, \boldsymbol{h}\sim\mathcal{N}(0,I)} \left[ \sum_{k=0}^{K-1} \left(H(\tilde{S}) \cdot \mathbf{diag}(\sigma'(G(\tilde{S})\tilde{\mathbf{x}})\tilde{S}^k \tilde{\mathbf{x}})\right) \left(H(\tilde{S}) \cdot \mathbf{diag}(\sigma'(G(\tilde{S})\tilde{\mathbf{x}})\tilde{S}^k \tilde{\mathbf{x}})\right)^{\mathsf{T}} \right] \tag{215}$$

We begin by focusing on the expectation over $\boldsymbol{h}$, such that,

$$\tilde{\Theta}_{GNN}^{(1)}(\boldsymbol{h}) = \sum_{k=0}^{K-1} \mathop{\mathbb{E}}_{\boldsymbol{g}\sim\mathcal{N}(0,I)} \left[ \mathop{\mathbb{E}}_{\boldsymbol{h}\sim\mathcal{N}(0,I)} \left[ \left(\sum_{k'=0}^{K-1} h_{k'} \tilde{S}^{k'} \mathbf{diag}(\sigma'(G(\tilde{S})\tilde{\mathbf{x}})\tilde{S}^k \tilde{\mathbf{x}})\right) \right.\right.$$
$$\left.\left. \times \left(\sum_{k''=0}^{K-1} h_{k''} \tilde{S}^{k''} \mathbf{diag}(\sigma'(G(\tilde{S})\tilde{\mathbf{x}})\tilde{S}^k \tilde{\mathbf{x}})\right)^{\mathsf{T}} \right]\right] \tag{216}$$

We first evaluate the inner expected value from equation 216 with respect to $\boldsymbol{h}$:

$$\mathop{\mathbb{E}}_{\boldsymbol{h}\sim\mathcal{N}(0,I)} \left[ \left(\sum_{k'=0}^{K-1} h_{k'} \tilde{S}^{k'} \mathbf{diag}(\sigma'(G(\tilde{S})\tilde{\mathbf{x}})\tilde{S}^k \tilde{\mathbf{x}})\right) \left(\sum_{k''=0}^{K-1} h_{k''} \tilde{S}^{k''} \mathbf{diag}(\sigma'(G(\tilde{S})\tilde{\mathbf{x}})\tilde{S}^k \tilde{\mathbf{x}})\right)^{\mathsf{T}} \right] \tag{217}$$

$$= \sum_{k'=0}^{K-1} \sum_{k''=0}^{K-1} \mathop{\mathbb{E}}_{\boldsymbol{h}\sim\mathcal{N}(0,I)} \left[ h_{k'} h_{k''} \left(\tilde{S}^{k'} \mathbf{diag}(\sigma'(G(\tilde{S})\tilde{\mathbf{x}})\tilde{S}^k \tilde{\mathbf{x}})\right) \left(\tilde{S}^{k''} \mathbf{diag}(\sigma'(G(\tilde{S})\tilde{\mathbf{x}})\tilde{S}^k \tilde{\mathbf{x}})\right)^{\mathsf{T}} \right] \tag{218}$$

$$= \sum_{k'=0}^{K-1} \left(\tilde{S}^{k'} \mathbf{diag}(\sigma'(G(\tilde{S})\tilde{\mathbf{x}})\tilde{S}^k \tilde{\mathbf{x}})\right) \left(\tilde{S}^{k'} \mathbf{diag}(\sigma'(G(\tilde{S})\tilde{\mathbf{x}})\tilde{S}^k \tilde{\mathbf{x}})\right)^{\mathsf{T}} \tag{219}$$

Above, in equation 219 we used the fact that $\mathop{\mathbb{E}}_{\boldsymbol{h}\sim\mathcal{N}(0,I)}[h_k h_{k'}] = \delta_{kk'}$. Now we can replace the inner expectation with respect to $\boldsymbol{h}$ in equation 216 with the quantity from equation 219 to get

$$\tilde{\Theta}_{GNN}^{(1)}(\boldsymbol{h}) = \sum_{k=0}^{K-1} \mathop{\mathbb{E}}_{\boldsymbol{g}\sim\mathcal{N}(0,I)} \left[ \sum_{k'=0}^{K-1} \left(\tilde{S}^{k'} \mathbf{diag}(\sigma'(G(\tilde{S})\tilde{\mathbf{x}})\tilde{S}^k \tilde{\mathbf{x}})\right) \left(\tilde{S}^{k'} \mathbf{diag}(\sigma'(G(\tilde{S})\tilde{\mathbf{x}})\tilde{S}^k \tilde{\mathbf{x}})\right)^{\mathsf{T}} \right] \tag{220}$$

$$= \sum_{k'=0}^{K-1} \tilde{S}^{k'} \mathop{\mathbb{E}}_{\boldsymbol{g}\sim\mathcal{N}(0,I)} \left[ \sum_{k=0}^{K-1} \left(\mathbf{diag}(\sigma'(G(\tilde{S})\tilde{\mathbf{x}})\tilde{S}^k \tilde{\mathbf{x}})\right) \left(\mathbf{diag}(\sigma'(G(\tilde{S})\tilde{\mathbf{x}})\tilde{S}^k \tilde{\mathbf{x}})\right)^{\mathsf{T}} \right] \tilde{S}^{k'} \tag{221}$$

Now we turn our attention to the expectation in equation 221 which we shall call $E^{(1)} \in \mathbb{R}^{nM \times nM}$[8].

$$E^{(1)} = \underset{\boldsymbol{g} \sim \mathcal{N}(0,I)}{\mathbb{E}} \left[ \sum_{k=0}^{K-1} \left( \mathbf{diag}(\sigma'(G(\tilde{S})\tilde{\mathbf{x}})\tilde{S}^k \tilde{\mathbf{x}}) \right) \left( \mathbf{diag}(\sigma'(G(\tilde{S})\tilde{\mathbf{x}})\tilde{S}^k \tilde{\mathbf{x}}) \right)^{\mathsf{T}} \right] \quad (222)$$

$$\Rightarrow (E^{(1)})_{ab} = \underset{\boldsymbol{g} \sim \mathcal{N}(0,I)}{\mathbb{E}} \left[ \sigma' \left( \langle \boldsymbol{g}, \boldsymbol{z}^{(a)} \rangle \right) \cdot \sigma' \left( \langle \boldsymbol{g}, \boldsymbol{z}^{(b)} \rangle \right) \right] \cdot \sum_{k=0}^{K-1} (\tilde{S}^k \tilde{x})_a (\tilde{S}^k \tilde{x})_b \quad (223)$$

$$= \underset{\boldsymbol{g} \sim \mathcal{N}(0,I)}{\mathbb{E}} \left[ \sigma' \left( \langle \boldsymbol{g}, \boldsymbol{z}^{(a)} \rangle \right) \cdot \sigma' \left( \langle \boldsymbol{g}, \boldsymbol{z}^{(b)} \rangle \right) \right] \cdot \langle \boldsymbol{z}^{(a)}, \boldsymbol{z}^{(b)} \rangle \quad (224)$$

Similar to our analysis in subsection 3.2, we begin by considering the case where the non-linearity is an identity function, i.e., $\sigma(z) = z$ which implies that $\sigma'(z) = 1$:

$$\sigma' \left( \langle \boldsymbol{g}, \boldsymbol{z}^{(a)} \rangle \right) = \sigma' \left( \langle \boldsymbol{g}, \boldsymbol{z}^{(b)} \rangle \right) = 1 \quad (225)$$

For this linear case, we shall name the expectation from equation 222, $B_{\mathsf{lin}}^{(1)} \in \mathbb{R}^{nM \times nM}$. Using equation 224 and equation 225 we can write the elements of $B_{\mathsf{lin}}^{(1)}$ as

$$(B_{\mathsf{lin}}^{(1)})_{ab} = \langle \boldsymbol{z}^{(a)}, \boldsymbol{z}^{(b)} \rangle \quad (226)$$

Thus, our analysis in this context renders equation 226, which is the same conclusion as that for the alignment when we only train the second layer (see equation 29).

Next, we analyze the expectation matrix $E^{(1)}$ when $\sigma(\cdot)$ is non-linear. In this non-linear case, for each element of $E^{(1)}$ we have from equation 224:

$$(E^{(1)})_{ab} = \underset{u,u' \sim \mathcal{N}(0,\Lambda)}{\mathbb{E}} \left[ \sigma' \left( ||\boldsymbol{z}^{(a)}||_2 \cdot u \right) \cdot \sigma' \left( ||\boldsymbol{z}^{(b)}||_2 \cdot u' \right) \right] \cdot \langle \boldsymbol{z}^{(a)}, \boldsymbol{z}^{(b)} \rangle \quad (227)$$

The Hermite expansion of the two functions in equation 227 is given by

$$\sigma' \left( ||\boldsymbol{z}^{(a)}||_2 \cdot u \right) = \sum_{\ell=0}^{\infty} \alpha_\ell \cdot p_\ell(u), \quad \text{and} \quad \sigma' \left( ||\boldsymbol{z}^{(b)}||_2 \cdot u' \right) = \sum_{\ell'=0}^{\infty} \beta_{\ell'} \cdot p_{\ell'}(u') \quad (228)$$

Inserting equation 228 into equation 227, we get

$$(E^{(1)})_{ab} = \sum_{\ell=0}^{\infty} \sum_{\ell'=0}^{\infty} \alpha_\ell \beta_{\ell'} \underset{u,u' \sim \mathcal{N}(0,\Lambda)}{\mathbb{E}} [p_\ell(u) p_{\ell'}(u')] \cdot \langle \boldsymbol{z}^{(a)}, \boldsymbol{z}^{(b)} \rangle \quad (229)$$

$$= \sum_{\ell=0}^{\infty} \alpha_\ell \beta_\ell \cdot \left( \frac{\langle \boldsymbol{z}^{(a)}, \boldsymbol{z}^{(b)} \rangle}{||\boldsymbol{z}^{(a)}||_2 \cdot ||\boldsymbol{z}^{(b)}||_2} \right)^l \cdot \langle \boldsymbol{z}^{(a)}, \boldsymbol{z}^{(b)} \rangle \quad (230)$$

We recall that for simpler analysis, we are analyzing the case where the non-linearity is the $\tanh$ function therefore the derivative of the activation i.e., $\sigma'(z) = \frac{1}{\cosh^2(z)}$, is an even function. This further leads to the Hermite expansion coefficients $\alpha_\ell, \beta_\ell$ being zero whenever $\ell$ is odd. With this in mind we divide the expansion from equation 230 into the two following parts and name them $B^{(1)}$ and $\Delta B^{(1)}$ respectively:

$$(E^{(1)})_{ab} = \underbrace{\alpha_0 \beta_0 \cdot \langle \boldsymbol{z}^{(a)}, \boldsymbol{z}^{(b)} \rangle}_{(B^{(1)})_{ab}} + \underbrace{\sum_{\ell=2,4,\cdots}^{\infty} \alpha_\ell \beta_\ell \cdot \left( \frac{\langle \boldsymbol{z}^{(a)}, \boldsymbol{z}^{(b)} \rangle}{||\boldsymbol{z}^{(a)}||_2 \cdot ||\boldsymbol{z}^{(b)}||_2} \right)^\ell \cdot \langle \boldsymbol{z}^{(a)}, \boldsymbol{z}^{(b)} \rangle}_{(\Delta B^{(1)})_{ab}} \quad (231)$$

Now, recalling equation 26 the alignment, which we will call $\mathcal{A}^{(1)}$ here to emphasize that we are only training the first layer, becomes:

$$\mathcal{A}^{(1)} = \mathbf{tr}(QE^{(1)}) = \mathbf{tr}(QB^{(1)}) + \mathbf{tr}(Q\Delta B^{(1)}) \quad (232)$$

---

[8]The superscript $(1)$ denotes that these matrices are defined for the analysis of the NTK for the first layer in contrast to those defined in the main body of the paper

Focusing on the first term in equation 232, we define the family of hermite transforms of the function $\frac{1}{\cosh^2(||\boldsymbol{z}^{(a)}||_2 u)}$ as $\tau_l(\cdot)$ for $l = 0, 1, 2, \cdots$ :

$$\tau_\ell(||\boldsymbol{z}^{(a)}||_2^2) \triangleq \int_{-\infty}^{\infty} \frac{1}{\cosh^2(||\boldsymbol{z}^{(a)}||_2 u)} \cdot p_\ell(u) \cdot e^{-u^2/2} du = \alpha_\ell \tag{233}$$

Note that $\tau_0(\cdot)$ is similar in functionality to the function $\hat{\sigma}(\cdot)$ defined in equation 95. Now, using the fact that $(B_{\mathsf{lin}})_{aa} = \langle \boldsymbol{z}^{(a)}, \boldsymbol{z}^{(a)} \rangle = ||\boldsymbol{z}^{(a)}||_2^2$, we have

$$(B^{(1)})_{ab} = \tau_0((B_{\mathsf{lin}})_{aa}) \cdot \tau_0((B_{\mathsf{lin}})_{bb}) \cdot (B_{\mathsf{lin}})_{ab} \tag{234}$$

$$\Rightarrow B^{(1)} = \tau_0(\mathbf{diag}(B_{\mathsf{lin}})) \cdot B_{\mathsf{lin}} \cdot \tau_0(\mathbf{diag}(B_{\mathsf{lin}})) \tag{235}$$

Therefore, similar to equation 103 from the proof of Lemma 4, we have

$$\lambda_{min}^2 \mathcal{A}_{\mathsf{lin}} \leq \mathbf{tr}(QB^{(1)}) \tag{236}$$

where $\lambda_{min} \triangleq \min_a \tau_0(||\boldsymbol{z}^{(a)}||_2^2)$. From equations 104-108 and with the assumption from equation 109 i.e. $||S||_{\mathsf{op}} \leq \nu$ we have:

$$\lambda_{min}^2 \geq \rho^{(1)} \tag{237}$$

$$\text{where } \rho^{(1)} \triangleq \left( \tau_0 \left( \sum_{k=0}^{K-1} \nu^{2k} \right) \right)^2 \tag{238}$$

which together with equation 236 gives us the following similar to Lemma 4:

$$\mathbf{tr}(QB^{(1)}) \geq \rho^{(1)} \mathcal{A}_{\mathsf{lin}} \tag{239}$$

Next, we aim to show that a result similar to Lemma 5 holds for this scenario, as this allows us to conclude that Theorem 3 also holds for the case when alignment is optimized based on the first layer. We will now check to see whether the element-wise inequality from lemma 5 also holds here.

From equation 231 and using the definition from equation 233, we have

$$(\Delta B^{(1)})_{ab} = \langle \boldsymbol{z}^{(a)}, \boldsymbol{z}^{(b)} \rangle \sum_{i=1}^{\infty} \alpha_{2i} \beta_{2i} \cdot \left( \frac{\langle \boldsymbol{z}^{(a)}, \boldsymbol{z}^{(b)} \rangle}{||\boldsymbol{z}^{(a)}||_2 \cdot ||\boldsymbol{z}^{(b)}||_2} \right)^{2i} \tag{240}$$

$$= \langle \boldsymbol{z}^{(a)}, \boldsymbol{z}^{(b)} \rangle \sum_{i=1}^{\infty} \tau_{2i}(||\boldsymbol{z}^{(a)}||_2^2) \cdot \tau_{2i}(||\boldsymbol{z}^{(b)}||_2^2) \cdot \left( \frac{\langle \boldsymbol{z}^{(a)}, \boldsymbol{z}^{(b)} \rangle}{||\boldsymbol{z}^{(a)}||_2 \cdot ||\boldsymbol{z}^{(b)}||_2} \right)^{2i} \tag{241}$$

Similar to the family of functions $g_\ell$, it can readily be verified numerically that for a given $\ell$, we either have $\tau_\ell(y) \geq 0, \forall y \geq 0$ or $\tau_\ell(y) \leq 0, \forall y \geq 0$.[9] Hence, $\tau_{2i}(||\boldsymbol{z}^{(a)}||_2^2) \cdot \tau_{2i}(||\boldsymbol{z}^{(b)}||_2^2) \geq 0$ and we can conclude that $(\Delta B^{(1)})_{ab}$ and $(B^{(1)})_{ab}$ have the same sign, which is the sign of $\langle \boldsymbol{z}^{(a)}, \boldsymbol{z}^{(b)} \rangle$. Now, for the case where $\langle \boldsymbol{z}^{(a)}, \boldsymbol{z}^{(b)} \rangle \geq 0$, we have:

$$\sum_{i=1}^{\infty} \tau_{2i}(||\boldsymbol{z}^{(a)}||_2^2) \cdot \tau_{2i}(||\boldsymbol{z}^{(b)}||_2^2) \cdot \left( \frac{\langle \boldsymbol{z}^{(a)}, \boldsymbol{z}^{(b)} \rangle}{||\boldsymbol{z}^{(a)}||_2 \cdot ||\boldsymbol{z}^{(b)}||_2} \right)^{2i} \cdot \langle \boldsymbol{z}^{(a)}, \boldsymbol{z}^{(b)} \rangle$$

$$\leq \sum_{i=1}^{\infty} \tau_{2i}(||\boldsymbol{z}^{(a)}||_2^2) \cdot \tau_{2i}(||\boldsymbol{z}^{(b)}||_2^2) \cdot \langle \boldsymbol{z}^{(a)}, \boldsymbol{z}^{(b)} \rangle \tag{242}$$

$$= \frac{\left( \sum_{i=1}^{\infty} \tau_{2i}(||\boldsymbol{z}^{(a)}||_2^2) \cdot \tau_{2i}(||\boldsymbol{z}^{(b)}||_2^2) \right)}{\tau_0(||\boldsymbol{z}^{(a)}||_2^2) \cdot \tau_0(||\boldsymbol{z}^{(b)}||_2^2)} \cdot (B^{(1)})_{ab} \tag{243}$$

---

[9] A simple Python script can be found in https://github.com/shervinkh2000/Cross_Covariance_NTK that plots $\tau_\ell(y)$ against $y$ for any given index $\ell$. See the file "numerical_verification_3.py"

It is straightforward to check numerically that $\left|\frac{\tau_{2i}(||\boldsymbol{z}^{(a)}||_2^2)}{\tau_0(||\boldsymbol{z}^{(a)}||_2^2)}\right|$ is increasing in $||\boldsymbol{z}^{(a)}||_2$, $\forall i \geq 1$.[10] Hence, we can continue from equation 240 and equation 243 to have

$$(\Delta B^{(1)})_{ab} \leq \frac{\left(\sum_{i=1}^{\infty} \tau_{2i}(||\boldsymbol{z}^{(a)}||_2^2) \cdot \tau_{2i}(||\boldsymbol{z}^{(b)}||_2^2)\right)}{\tau_1(||\boldsymbol{z}^{(a)}||_2^2) \cdot \tau_1(||\boldsymbol{z}^{(b)}||_2^2)} \cdot B_{ab} \tag{244}$$

$$\leq \left(\frac{\sum_{i=1}^{\infty}(\tau_{2i}((||\boldsymbol{z}||_2^2)_{max}))^2}{(\tau_0((||\boldsymbol{z}||_2^2)_{max}))^2}\right)(B^{(1)})_{ab} \tag{245}$$

$$= \beta^{(1)} \cdot (B^{(1)})_{ab} \tag{246}$$

where

$$\beta^{(1)} \triangleq \left(\frac{\sum_{i=1}^{\infty}(\tau_{2i}((||\boldsymbol{z}||_2^2)_{max}))^2}{(\tau_0((||\boldsymbol{z}||_2^2)_{max}))^2}\right) \tag{247}$$

and $(||\boldsymbol{z}||_2^2)_{max} = \max_{a'}(||\boldsymbol{z}_{a'}||_2^2)$. In the proof of Lemma 5, we made no further assumptions and considered the worst case upper bound on $\Delta B_{ab}$ when $(||\boldsymbol{z}||_2^2)_{max}$ is infinitely large. But a similar approach cannot be adopted here since the limit of the sum $\lim_{y \to \infty} \sum_{\ell=1}^{\infty}(\tau_{2i}(y))^2$ is unbounded. However, given the assumptions on the data and the shift operator $S$ so far and with some additional mild assumptions, it is possible to upper bound $(||\boldsymbol{z}||_2^2)_{max}$ and thus, derive a meaningful expression for constant $\beta$. Recall the definition of $\boldsymbol{z}_\ell \in \mathbb{R}^K$:

$$\boldsymbol{z}_\ell \triangleq \left[\tilde{\mathbf{x}}_\ell, \ (\tilde{S}\tilde{\mathbf{x}})_\ell, \ \cdots, \ (\tilde{S}^{K-1}\tilde{\mathbf{x}})_\ell\right]^\mathsf{T} \tag{248}$$

For the $k$-th element of $\boldsymbol{z}_\ell$ we can write:

$$(\tilde{S}^k\tilde{\mathbf{x}})_\ell \leq \max_i ||S^k\mathbf{x}_i||_\infty \leq \max_i ||S^k\mathbf{x}_i||_2 \leq \max_i ||S^k||_{\mathsf{op}}||\mathbf{x}_i||_2 \leq ||S^k||_{\mathsf{op}} \tag{249}$$

$$\Rightarrow ||\boldsymbol{z}_\ell||_2^2 \leq \sum_{k=0}^{K-1} ||S^k||_{\mathsf{op}}^2 \tag{250}$$

In order to give an upper bound on the maximum possible value for $||\boldsymbol{z}_\ell||_2^2$, we need an upper bound on $||S||_{\mathsf{op}}$. To see why such an upper bound makes sense, recall the constraint from Lemma 2: $||\sum_{k=0}^{K-1} S^k||_F \leq \sqrt{\alpha/(\eta M)}$. One straightforward way to make sure that this constraint is satisfied, is to normalize the shift operator $S$, such that its Frobenius norm is bounded (which is indeed the method used in the experiments for this paper. See Appendix E). Assuming that $||S||_F \leq \frac{1}{K}\sqrt{\alpha/(\eta M)} \leq 1$, we have

$$||\sum_{k=0}^{K-1} S^k||_F \leq \sum_{k=0}^{K-1} ||S^k||_F \leq \sum_{k=0}^{K-1} ||S||_F^k \leq K||S||_F \leq \sqrt{\alpha/(\eta M)} \tag{251}$$

Since in practice we don't know precisely what the constant $\alpha$ should be, we opt to simply normalize $S$ so that $||S||_F = 1$. Therefore:

$$||S||_{\mathsf{op}} \leq ||S||_F = 1 \Rightarrow ||\boldsymbol{z}_\ell||_2^2 \leq \sum_{k=0}^{K-1} ||S^k||_{\mathsf{op}}^2 \leq K \Rightarrow (||\boldsymbol{z}||_2)_{max} \leq \sqrt{K} \tag{252}$$

Going back to equation 247, with the assumption in equation 252 we have

$$\beta^{(1)} = \left(\frac{\sum_{i=1}^{\infty}(\tau_{2i}(K))^2}{(\tau_0(K))^2}\right) \tag{253}$$

The above constant can be numerically evaluated for different values of $K$. For example, for $K = 3$, we have $\beta^{(1)} = 0.7320$ which is close to the constant we had when training only the second layer ($\beta = 0.57$, see equation 120). Now that we've shown similar results to Lemma 4 and Lemma 5 (see equation 239 and equation 247 respectively) for this case where we train only the first layer, we can conclude with a result similar to Theorem 3 which we shall present as the following corollary:

---

[10] A simple Python script can be found in https://github.com/shervinkh2000/Cross_Covariance_NTK that plots $|\frac{\tau_{2i}(y)}{\tau_0(y)}|$ against $y$ for any given index $i$. See the file "numerical_verification_4.py"

**Corollary 3.** *Under the assumption* $\mathcal{A}_{\mathsf{lin}} = \boldsymbol{tr}\,(QB_{\mathsf{lin}}) \geq \xi \cdot ||Q||_F ||B_{\mathsf{lin}}||_F$, $\mathcal{A}_{\mathsf{lin}}$ *lower bounds the alignment for the two-layer GNN with* $\tanh$ *non-linearity where we only train the first layer,* $\mathcal{A}^{(1)}$, *up to a constant as follows*

$$\mathcal{A}^{(1)} \geq \left(b - \frac{s}{\xi}\right)\mathcal{A}_{\mathsf{lin}}\,, \tag{254}$$

*for some constants positive constants $b$ and $s$ and $0 \leq \xi \leq 1$.*

## H  ADDITIONAL EXPERIMENTAL DETAILS AND RESULTS

### H.1  CONVERGENCE OF TRAINING ERROR.

We recall that there was a time series associated with each of the 100 features in the rfMRI time series data for an individual. For each individual, these 100 time series were utilized in the experiments. For a given individual, we denote the time series across 100 nodes at time step $t$ as the vector $\boldsymbol{z}^{(t)} \in \mathbb{R}^{100 \times 1}$.

For each individual, we created $N_{train} = 1000$ and $N_{test} = 100$ training and test samples respectively by randomly sampling (without replacement) pairs of vectors $\boldsymbol{z}^{(t)}, \boldsymbol{z}^{(t+1)}$ from the time series of length $N = 4500$. Next, the normalized sample covariance and sample cross covariance matrices were constructed using only the training data:

$$C_{XX}^{normalized} = \frac{X_{train}X_{train}^{\mathsf{T}}}{||X_{train}X_{train}^{\mathsf{T}}||}, \; C_{XY}^{normalized} = \frac{X_{train}Y_{train}^{\mathsf{T}}}{||X_{train}Y_{train}^{\mathsf{T}}||} \tag{255}$$

Afterwards, batch stochastic gradient descent with the Adam optimizer Kingma & Ba (2014) and the Pytorch library Paszke et al. (2019) were used to train the following four models:

- Two-layer GNN with $K = 2$, $F = 50$ and GSO $S = C_{XY}^{normalized}$
- Two-layer GNN with $K = 2$, $F = 50$ and GSO $S = C_{XX}^{normalized}$
- Single graph filter with $K = 2$ and GSO $S = C_{XY}^{normalized}$
- Single graph filter with $K = 2$ and GSO $S = C_{XX}^{normalized}$

Regarding the choices of the parameters, $K = 2$ was chosen as equation 19 directly motivates using $C_{XY}$ as the GSO for the $K = 2$ case. Note that using $C_{XY} - I$ and $C_{XY}$ is essentially the same since in the graph filter polynomial $\sum_{k=0}^{K-1} h_k S^k \mathbf{x}$ we always have the term $h_0 I$ regardless of the choice of GSO. Furthermore, in subsection H.3 and Fig. 5, we observed that changing $K$ does not have a noticeable impact on the model performance. Therefore, $K = 2$ is a reasonable choice for the experiments here. For choosing the number of features in the hidden layer, $F$, and the learning rate $\eta$ for training the GNN models, the Optuna hyperparameter optimization framework was leveraged Akiba et al. (2019). The learning rate $\eta_1 = 0.0125$ was chosen for training the GNN models and $\eta_2 = 50 \cdot \eta_1$ for training the graph filters. For each individual, the training process was run 10 times with different permutations of the training and test sets, and the average over these was computed for all of the individual training and test error plots.

We also acknowledge that the GNN and Graph filter architectures were implemented using the Alelab Graph Neural Network library for Python based on the work of Gama et al. (2019).

### H.2  GOING DEEPER THAN TWO LAYERS.

Here, we provide the experimental results for the setting when the GNNs may have more than two layers. Before that, we formalize a multi-layer GNN architecture with a graph filter as the building block and that has multiple-input-multiple-output (MIMO) information processing functionality.

**Multi-layer GNN.** We recall that the ability to learn non-linear mappings by GNNs are fundamentally based on addition of an element-wise non-linearity to a graph filter to form a graph perceptron, which is realized via a point-wise non-linearity $\sigma(\cdot)$ as $\sigma(H(S)\mathbf{x})$. We can further build upon the expressivity (and therefore, representational power) of a graph perceptron by concatenating multiple graph perceptrons to form a multi-layer GNN architecture. In this scenario, the relationship between the input $\mathbf{q}_{(l-1)}$ and the output $\mathbf{q}_{(\ell)}$ of the $\ell$-th layer of the GNN is given by

$\mathbf{q}_{(\ell)} = \sigma(H_{(\ell)}(S)\mathbf{q}_{(\ell-1)})$. Based on the definitions of a graph perceptron and a multi-layer GNN, we next formalize the GNN architecture with MIMO functionality.

**Definition 3** (Multiple-Input-Multiple-Output Graph Neural Network). *Gama et al. (2020) We can substantially increase the representation power of GNNs by incorporating multiple parallel features per layer. These features are the result of processing multiple input features with a parallel bank of graph filters. Let us consider $F_{\ell-1}$ $n$-dimensional inputs $\mathbf{q}^1_{(\ell-1)}, \ldots, \mathbf{q}^{F_{l-1}}_{(\ell-1)}$ at layer $\ell$. Each input, $\mathbf{q}^g_{(\ell-1)}$ for $g \in \{1, \ldots, F_{\ell-1}\}$ is processed in parallel by $F_\ell$ different graph filters to output the $F_\ell$ $n$-dimensional outputs denoted by $\mathbf{u}^{fg}_{(\ell)}$ with the following relationships*

$$\mathbf{u}^{fg}_{(\ell)} = H^{fg}_{(\ell)}(S)\mathbf{q}^g_{(\ell-1)} = \sum_{k=0}^{K} h^{fg}_{(\ell),k} S^k \mathbf{q}^g_{(\ell-1)}, \quad f \in \{1, \ldots, F_\ell\}.$$

*The outputs $\mathbf{u}^{fg}_{(\ell)}$ are subsequently summarized along the input index $g$ to yield the aggregated outputs*

$$\mathbf{u}^f_{(\ell)} = \sum_{g=1}^{F_{l-1}} H^{fg}_{(\ell)}(S)\mathbf{q}^g_{(\ell-1)}, \quad f \in \{1, \ldots, F_\ell\}.$$

*The aggregated outputs $\mathbf{u}^f_{(\ell)}$ are finally passed through a non-linearity $\sigma(\cdot)$ to compute the $\ell$-th layer output as follows*

$$\mathbf{q}^f_{(\ell)} = \sigma\left(\mathbf{u}^f_{(\ell)}\right), \quad f \in \{1, \ldots, F_\ell\}.$$

*A GNN in its complete form is a concatenation of $L$ such layers, in which each layer computes the above operations.*

In the main paper, we theoretically analyzed a two-layer GNN without non-linearity in the final layer, with $F_0 = F_2 = 1$ (since we only have one input and one output feature vector in each sample), and $F$ number of features in the hidden layer i.e., $F_1 = F$ (see Fig. 3).

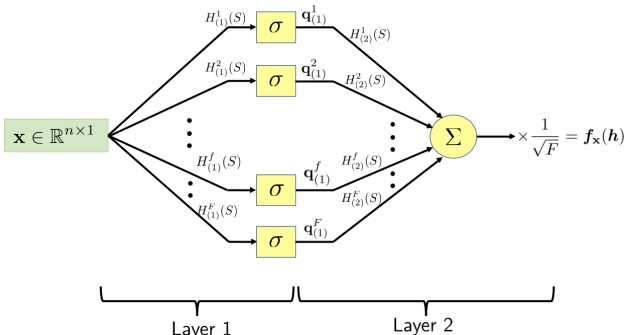

Figure 3: The two layer GNN architecture defined in section 3.2

**Experiment results.** To assess the effect of increasing the depth of the GNN in the experiments, we present results for two-layer, three-layer and four-layer GNNs that had been trained for one individual in the HCP-YA dataset. The training and test loss for the gradient descent for these models is illustrated in Fig. 4. It can be seen that regardless of the depth of the GNNs, the training loss and test loss for the GNN with $S = C_{XY}$ converged faster as compared to those with $S = C_{XX}$ and to a smaller final value.

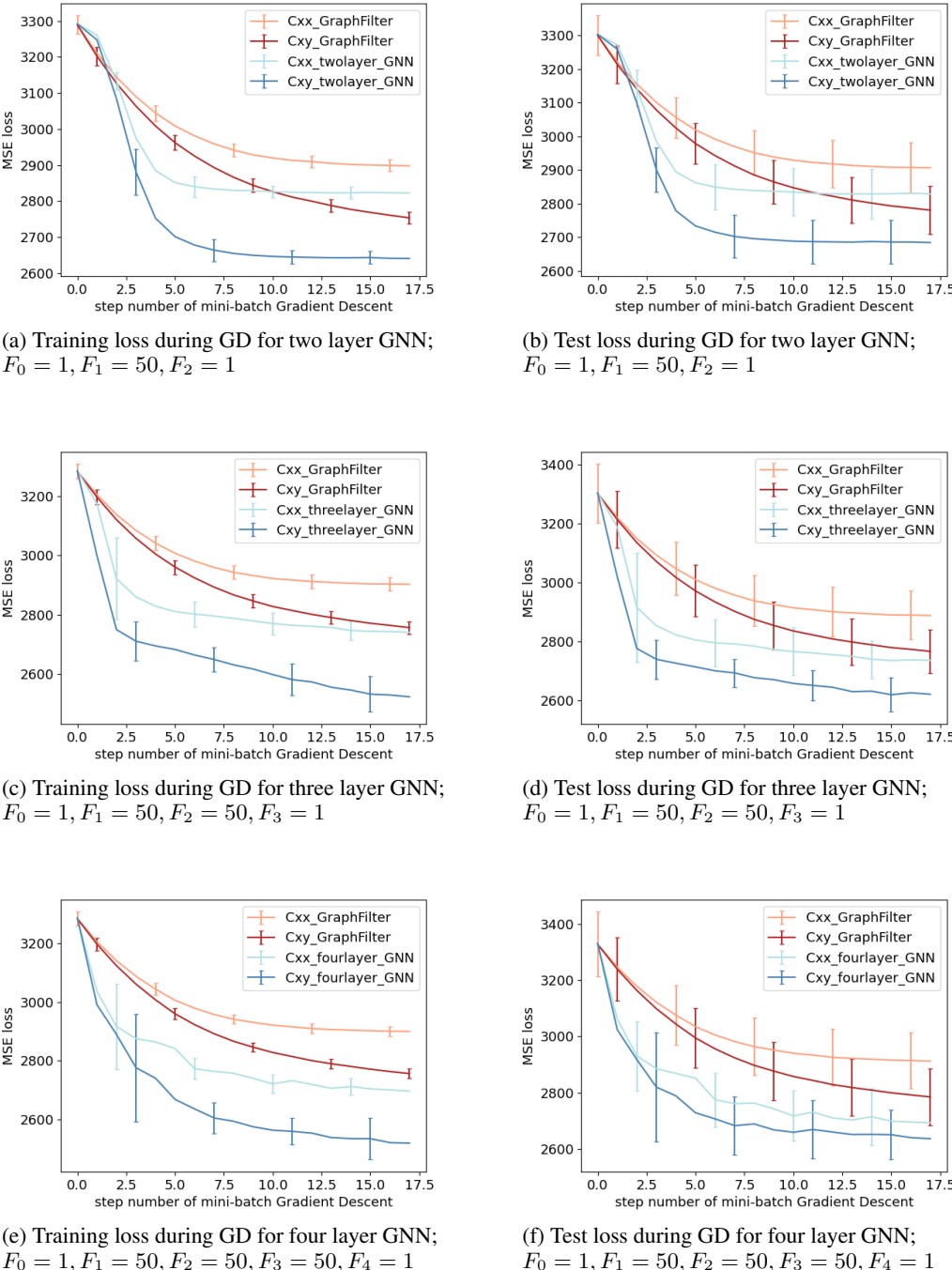

(a) Training loss during GD for two layer GNN; $F_0 = 1, F_1 = 50, F_2 = 1$

(b) Test loss during GD for two layer GNN; $F_0 = 1, F_1 = 50, F_2 = 1$

(c) Training loss during GD for three layer GNN; $F_0 = 1, F_1 = 50, F_2 = 50, F_3 = 1$

(d) Test loss during GD for three layer GNN; $F_0 = 1, F_1 = 50, F_2 = 50, F_3 = 1$

(e) Training loss during GD for four layer GNN; $F_0 = 1, F_1 = 50, F_2 = 50, F_3 = 50, F_4 = 1$

(f) Test loss during GD for four layer GNN; $F_0 = 1, F_1 = 50, F_2 = 50, F_3 = 50, F_4 = 1$

Figure 4: The effect of increasing depth of the GNN

## H.3 THE EFFECT OF CHANGING THE NUMBER OF GRAPH FILTER TAPS $K$

We recall our motivation for using the cross-covariance graph as the GSO from Theorem 2 where we concluded that the optimal GSO $S^*$ should satisfy $\sum_{k=0}^{K-1}(S^*)^k = \mu \cdot C_{XY}$. For the case $K = 2$, this leads to $S^*$ being proportional to $C_{XY}$ (see equation 19). However, for $K > 2$, while the optimal GSO is still clearly a function of the cross-covariance $C_{XY}$, solving equation 18 to find a closed form expression for $S^*$ is not trivial. In this context, we investigated empirically whether $S = C_{XY}$ was a better choice than $S = C_{XX}$ when $K > 2$. The plots in Fig. 5 illustrate the training error for GNN and graph filter models with different values of $K$ for one individual in the HCP-YA dataset. Clearly, GNNs with $S = C_{XY}$ outperformed those with $S = C_{XX}$ as GSO. These experiments indicate that the cross-covariance matrix is still a better choice as a GSO for GNNs when $K > 2$.

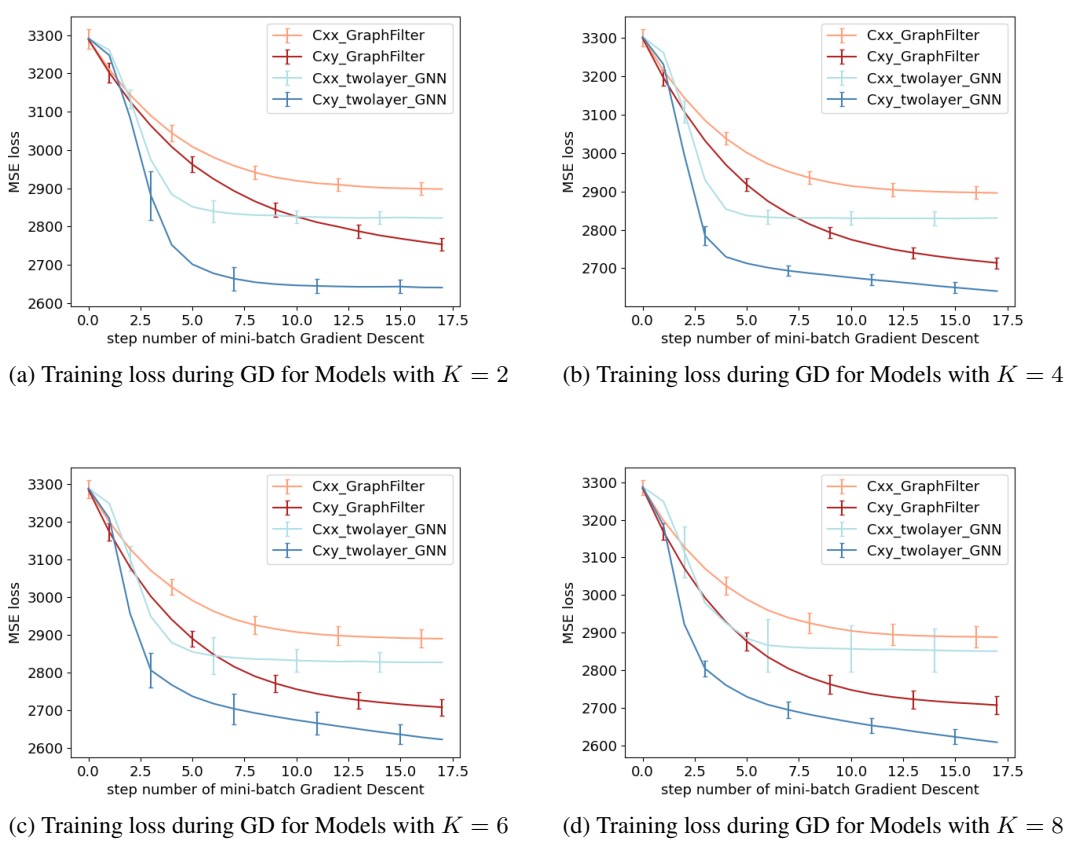

(a) Training loss during GD for Models with $K = 2$

(b) Training loss during GD for Models with $K = 4$

(c) Training loss during GD for Models with $K = 6$

(d) Training loss during GD for Models with $K = 8$

Figure 5: The effect of changing the number of filter taps $K$

## H.4 THE EFFECT OF CHANGING THE NON-LINEAR ACTIVATION FUNCTION $\sigma(\cdot)$.

Our theoretical results hold for $\tanh$ function as the non-linearity $\sigma(\cdot)$. In this section, we investigated empirically whether the cross-covariance matrix was a better choice as a GSO than the covariance matrix for different choices of $\sigma(\cdot)$. The plots in Fig. 6 demonstrate the results from training for GNN and graph filter models with different activation functions, for a single individual. Aside from the setting where the activation function was the ReLU function (for which the convergence depends highly on the initialization thus the variance in the training process between different runs is too high to conclude anything meaningful), the experiments for other activation functions showed that the GNNs with $S = C_{XY}$ converged faster and to a smaller final value as compared to GNNs with $S = C_{XX}$. This observation suggests that the theoretical insights drawn from the scenario of $\sigma = \tanh$ extends empirically to settings with the choice of other activation functions.

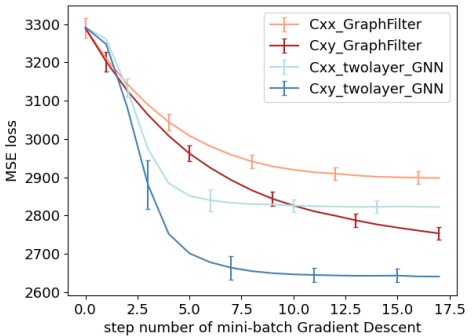
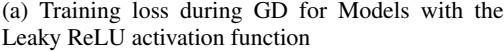

(a) Training loss during GD for Models with the Leaky ReLU activation function

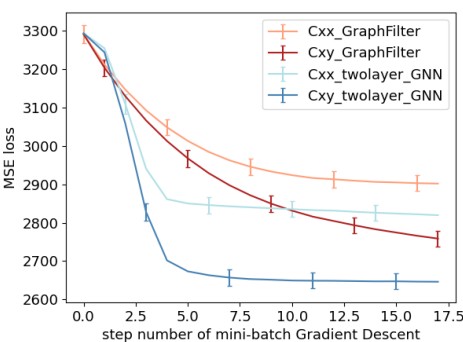

(b) Training loss during GD for Models with the $\tanh$ activation function

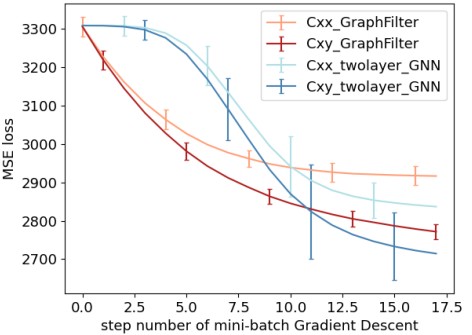

(c) Training loss during GD for Models with the Sigmoid activation function

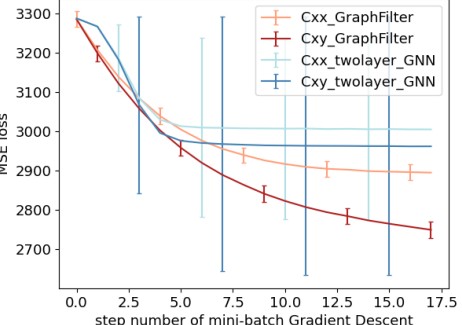

(d) Training loss during GD for Models with the ReLU activation function

Figure 6: The effect of changing the non-linear activation function $\sigma(\cdot)$ on GNN performance. (Graph filter curves are only plotted for comparison)

## H.5 Time-series prediction using more than just the previous time step

In the experiments in the main paper, we considered a prediction task where we use the value of the signal at just the previous time step to predict the current signal. However, in practice, the time series forecasting algorithms typically use information from a longer history, i.e., more than just the previous time step. Here, we investigate a time-series prediction model that uses the model in Section 4 as the building block to form predictions from an arbitrary set of time steps in the past.

In this design, we used $D$ separate graph filters to process the signal values at the previous $D$ time steps and aggregated the outputs of the filters to form the final prediction:

$$\boldsymbol{f}(\boldsymbol{z}^{(t-1)}, \boldsymbol{z}^{(t-2)}, \cdots, \boldsymbol{z}^{(t-D)}) = \sum_{k=0}^{K-1} h_{k,1} S_1^k \boldsymbol{z}^{(t-1)} + \cdots + \sum_{k=0}^{K-1} h_{k,D} S_D^k \boldsymbol{z}^{(t-D)} \qquad (256)$$

This methodology can be extended to the two-layer GNN architecture by replacing each graph filter in the GNN with multiple graph filters as per Equation 256. Note that we do not restrict are not restricted to use the same graph shift operator in these filters. For example, for processing the signal value from $d$ time steps ago, we could utilize the cross-covariance between signal values with a distance of $d$ time steps i.e.

$$S_d = \mathbb{E}\left[z^{(t)}(z^{(t-d)})^T\right] \qquad (257)$$

To gauge the usefulness of this setting, we consider the multiple filter model in equation 256 for $D = 1, 2, 3, 4, 5$. The result can be seen in Figures 7a, 7b. It can be observed that there is a significant gain in increasing from $D = 1$ to $D = 2$. However increasing $D$ further does not seem to yield better test performance, at least for this dataset. Note that the cross-covariance matrix was used for all the models in Figure 7.

Next we set $D = 2$ to compare the cross-covariance and covariance graph constructions in this setting. The results can be seen in Figures 7c, 7d. It can be observed that in this setting as well, models with cross-covariance graph shift operator outperform those with covariance graphs. This holds for both the graph filter models and the two-layer GNNs. Also note that increasing $D$ to 2, results in better performance for the two-layer GNN in addition to the Graph filter models However similar to the results observed for the graph filter, increasing $D$ further for the GNNs did not result in any observable gains.

## H.6 Other Graphs constructed based on the input data

In our experiments in the main paper, we have considered the cross-covariance matrix as the graph shift operator to be representative of the class of graphs constructed from only the input data. In order to further emphasize on the advantages offered by cross-covariance graphs, we also consider the two following additional methods of constructing a graph from the input data (See Qiao et al. (2018) for a detailed review of different methods of graph construction used in the literature). The first method is based on Euclidean distance between values of the signals on each node of the graph and using the nonlinear Gaussian kernel, the weight between node $i$ and $j$ of the graph is quantified as follows:

$$S_{ij} = e^{-\frac{||x_i - x_j||_2^2}{2\varrho^2}}, \qquad (258)$$

where $\varrho = 1$ in our experiments.

The second method for graph construction set the weights of the adjacency matrix proportional to the Pearson's correlation coefficient between two nodes. In this case, the weight between node $i$ and $j$ of the graph is

$$S_{ij} = \frac{(x_i - \bar{x}_i)^T (x_j - \bar{x}_j)}{||x_i - \bar{x}_i||_2 ||x_j - \bar{x}_j||_2} \qquad (259)$$

where in both Equation 258 and Equation 259, $x_i, x_j \in \mathbb{R}^{N_{\text{train}} \times 1}$ are the $i$-th and $j$-th columns of the input data matrix $X_{\text{train}} \in \mathbb{R}^{n \times N_{\text{train}}}$ respectively.

The results corresponding to these two choices of graph construction for one individual have been illustrated in Fig. 8. It is observed that while the performance of the models with different input-based graph constructions vary, they are all consistently outperformed by the models with the cross-covariance based graph. This trend is consistent across different individuals in the dataset.

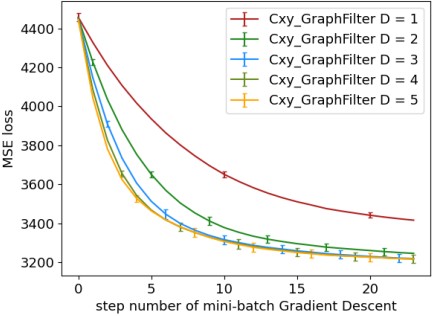

(a) Training loss during GD for Time-series prediction using $D$ previous time steps.

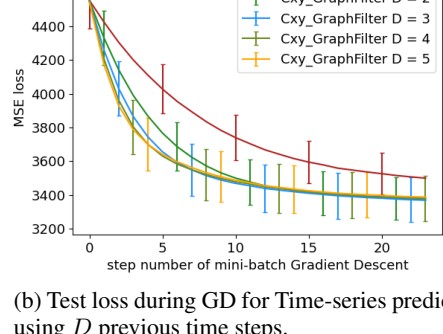

(b) Test loss during GD for Time-series prediction using $D$ previous time steps.

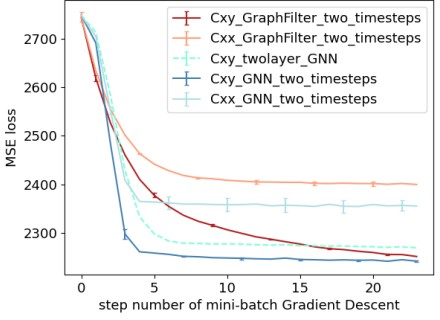

(c) Training loss during GD for Time-series prediction using $D = 2$ previous time steps.

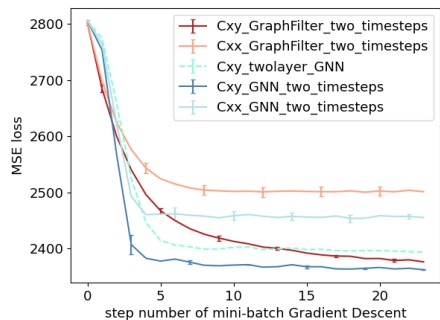

(d) Test loss during GD for Time-series prediction using $D = 2$ previous time steps.

Figure 7: Time-series prediction using $D$ previous time steps

As an additional baseline, we have also included the result for a Fully connected two layer neural network (FCNN) for comparison. The FCNN and the two layer GNN with cross-covariance graph exhibit comparable performance in terms of test final test error, while the GNN converges faster than FCNN. In general, for the complete HCP-YA dataset, the FCNN often has slightly smaller final test error, which is achieved at the expense of complexity as it has almost 100 times larger number of trainable parameters as compared to GNN models.

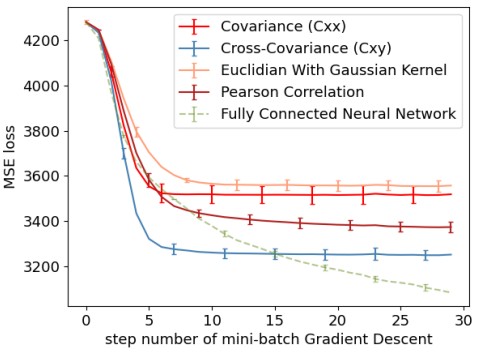

(a) Training loss during GD when using different constructions for the graph shift operator

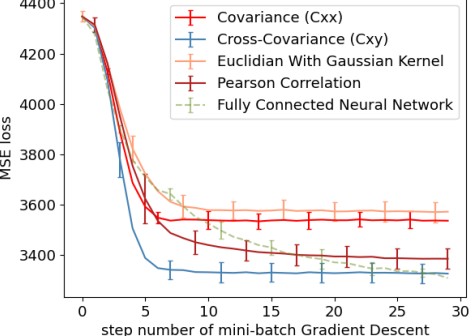

(b) Test loss during GD when using different constructions for the graph shift operator

Figure 8: Comparison between GNN models with cross-covariance graphs and GNN models with graphs constructed according to different construction methods. The models compared here are two-layer GNNs.

## H.7  OTHER DATASETS

We use a similar setup as described in Appendix H.1 to conduct some preliminary experiments on two other public datasets. These two datasets have been previously investigated in Cao et al. (2021a), although for a different variation of forecasting task.

**Traffic Flow Dataset.** We utilize the PEMS07 traffic flow dataset Chen et al. (2001) for traffic flow prediction. The data collected is from the California Department of Transportation network. It is an $n = 228$-dimensional time-series with $N = 12671$ time steps. The time interval between each consecutive point in the time series is 5 minutes.

**ECG dataset** We utilize the ECG5000 dataset from UCR time-series classification archive Chen et al. (2015). The data is an $n = 140$ dimensional ECG time series with $N = 5000$ time steps.

For each dataset, we created $N_{train} = 1000$ and $N_{test} = 100$ training and test samples respectively by randomly sampling pairs of vectors $z^{(t)}, z^{(t+\Delta t)}$ from the respective time series. Next, the normalized sample covariance and sample cross covariance matrices were constructed using only the training data. Note that for small values of $\Delta t$, since the signals $z^{(t)}$ and $z^{(t+\Delta t)}$ tended to be very similar, which led to very similar covariance and cross-covariance matrices. Therefore, although the cross-covariance based models achieved better performance for all tested values of $\Delta t$, we showcase our results for a relatively large value of $\Delta t$ ($\Delta t = 1000$ for the ECG dataset and $\Delta t = 20$ for the PEMS07 dataset) such that, the performance improvements are observable. As seen in Figure 9, for both datasets, the cross-covariance based models converge faster and to a smaller final test error, which is a consistent observation with HCP-YA dataset and has been predicted by our theoretical results.

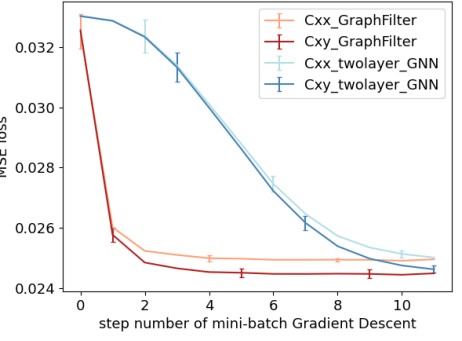

(a) Training loss for time series prediction task on the PEMS07 dataset.

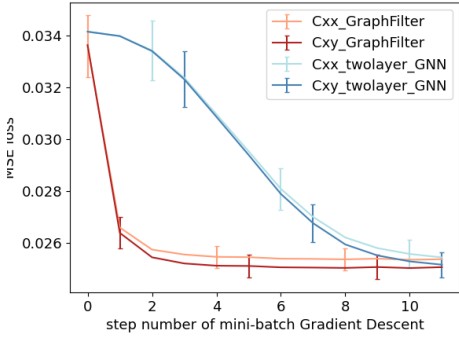

(b) Test loss for time series prediction task on the PEMS07 dataset.

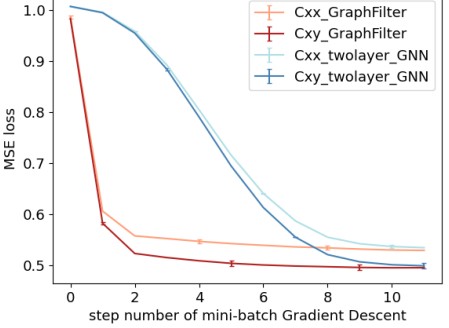

(c) Training loss for time series prediction task on the ECG5000 dataset.

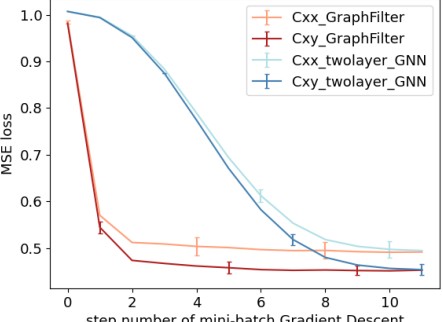

(d) Test loss for time series prediction task on the ECG5000 dataset.

Figure 9: Experimental results for different datasets

