# OpenReview forum: "Neural Tangent Kernels Motivate Graph Neural Networks with Cross-Covariance Graphs"
_ICLR.cc/2024/Conference — Submitted to ICLR 2024_

### Official Review · Reviewer_bUqG · 2023-10-30

**Soundness:** 3 good
**Presentation:** 2 fair
**Contribution:** 3 good
**Rating:** 5
**Confidence:** 3

**Summary:**

This study offers a theoretical analysis of Graph Neural Networks (GNNs), emphasizing their infinite-width limit behavior through the lens of the Neural Tangent Kernel (NTK). The research illuminates the constancy of the NTK in such scenarios, advancing the understanding of neural network learning dynamics.

Furthermore, the paper explores the implications of varying training intensities across GNN layers, enhancing interpretability in a field often perceived as a 'black box.' The authors validate their theoretical assertions with detailed experiments, utilizing public datasets to ensure reproducibility and credibility.

This paper bridges complex theoretical insights with practical machine-learning applications, contributing substantially to the discourse on neural network training, optimization, and generalization. Its rigorous approach and novel findings mark a significant stride in understanding and leveraging the full potential of GNNs.

**Strengths:**

1. By exploring the constancy of the NTK in the infinite-width limit, the study sheds light on complex learning dynamics, enhancing the scientific understanding of how GNNs train and generalize. This rigorous theoretical foundation pushes the boundaries of existing knowledge in neural network behavior.
2. By dissecting layer-specific training implications, the paper contributes to the interpretability of neural networks, helping researchers to better understand and optimize their GNN models, which is particularly valuable in the 'black box' context of deep learning.

**Weaknesses:**

1. While the paper is strong in theoretical analysis, it may not provide extensive insight into the practical applications of the findings. The implications for real-world scenarios, particularly how these theoretical insights into GNNs and NTK behavior could be harnessed for tangible improvements in specific use cases, might not be thoroughly discussed.
2. If the experiments were conducted within a narrow set of conditions or datasets, they might not fully represent the complexities of real-world data and scenarios. This limitation could raise questions about the generalizability of the findings and their robustness when applied to diverse, practical challenges in the field of machine learning.

**Questions:**

1. Given the depth of the research, what do the authors see as the next steps or future directions in this domain? Additionally, are there any inherent limitations or challenges in the proposed methods or findings that might need to be addressed in subsequent research?
2. Can the analysis be extended to deeper (more than 2 layers) GNNs where the aggregation operation is used in internal layers?

---

> ### Author Response · Authors · 2023-11-17
> **Response to Reviewer bUqG**
>
> We thank the reviewer for their appreciation of the key theoretical contributions in our paper. Specific concerns have been addressed below.
>
>
> **Practical applications for specific use-cases in real-world scenarios.** Our experiments originally considered the task of predicting brain functional activity in a future time step based on that in the current time step. This task forms the foundation of the time series forecasting problem in neuroimaging data. Thus, our findings suggest that the choice of cross-covariance graphs in GNNs will result in better prediction performance as compared to the GNNs that rely on the covariance graph. We expanded the scope of our experiments to test this intuition. Specifically, in revised **Appendix H.5**, we have provided a design of a time series prediction model that leverages information from an arbitrary length of data in the past to predict the future. For this specific application, the time series prediction improved by incorporating historical data and more importantly, GNNs with cross-covariance graphs indeed outperformed GNNs with input-based graphs.
>
>
> **Additional datasets.** We have updated the **Appendix H.7** with preliminary results on two other datasets (the PEMS07 traffic flow dataset and an ECG dataset.). For these settings, the cross-covariance based GNN models still retain a performance advantage over the alternatives.
>
> The limits of our theoretical framework (primarily the fact that the input and output are vectors with the same dimension) certainly impose limits regarding the scope of practical tasks on which we could investigate our theoretical results. In future work, generalizing our theory beyond the setting considered in the paper will let us investigate the generalizability of our findings on a more holistic set of inference problems.
>
>
> **GNNs deeper than 2 layers.** Our current experimental results suggest that our theoretical findings extend to GNNs deeper than two layers (see Fig. 4. that shows the effect of increasing the depth of the model). Even as the depth increases beyond two layers, the cross-covariance based GNNs consistently converge faster and achieve a smaller loss). Additionally, the fact that NTK-based results in the literature are usually generalizable to deeper models makes us optimistic that our results can also be generalized to deeper models. While this was not a focus of the current paper, it is an avenue we aim to explore theoretically in future work.
>
>
> **Future directions, limitations, and challenges.** As discussed previously, extending the theoretical analyses to GNNs deeper than 2 layers is an immediate direction of interest. We expect this analysis to have novel theoretical challenges pertaining to intermediate GNN layers. Furthermore, we will aim to expand the scope of our results for inference tasks beyond the considered setting of the multi-variate regression problem  (for instance, to node-level inference tasks and classification). Another limitation of our theoretical contributions is the lack of thorough analysis of the tightness of the lower bounds on alignment, i.e., the gap between ${\cal A}$ and ${\cal A}_L$.
>
> We hope that we addressed the reviewer's concerns sufficiently, in which case, we would be grateful if your rating of our paper could be re-evaluated. We would be happy to clarify any additional concerns.

---

### Official Review · Reviewer_TKzH · 2023-10-31

**Soundness:** 2 fair
**Presentation:** 3 good
**Contribution:** 2 fair
**Rating:** 5
**Confidence:** 4

**Summary:**

The authors delve into the theoretical aspects of Neural Tangent Kernels (NTKs) and their influence on the learning and generalization behaviors of over-parameterized neural networks in supervised learning tasks. They introduce the concept of "alignment" between the eigenvectors of the NTK kernel and the given data, which appears to play a significant role in governing the rate of convergence of gradient descent and the generalization to unseen data. The paper specifically explores NTKs and alignment in the context of Graph Neural Networks (GNNs). The authors' analysis reveals that optimizing alignment corresponds to optimizing the graph representation or the graph shift operator within a GNN. This investigation leads to the establishment of theoretical guarantees concerning the optimality of certain design choices in GNNs.

**Strengths:**

* The paper is well-organized, with a clear delineation of concepts such as NTKs, alignment, and their relevance in the context of GNNs.

* The paper's findings have the potential to advance the understanding and analysis of Graph Neural Networks, providing theoretical insights that could be valuable for the community.

**Weaknesses:**

* The paper seems to miss a crucial related work: Huang, W., Li, Y., Du, W., Yin, J., Da Xu, R. Y., Chen, L., & Zhang, M. (2021). "Towards deepening graph neural networks: A GNTK-based optimization perspective," ICLR 2022. Including and discussing this work could provide a more comprehensive perspective and strengthen the literature review section.

* The theoretical framework primarily relies on the existing NTK theory regarding optimization and generalization. While the authors have cited relevant works, a more distinct and innovative theoretical contribution that extends beyond the current NTK theories would enhance the paper's novelty and impact.

* The paper's discussion on alignment seems closely related to node classification and graph classification tasks. However, there appears to be a lack of relevant examinations or experiments to empirically validate the proposed concepts and theories in these tasks, making it difficult to assess their practical relevance and effectiveness.

* The paper could significantly benefit from a more robust and comprehensive experimental section. Ensuring that the experiments thoroughly validate the theoretical findings, involve extensive comparisons with baseline methods, and are evaluated across various datasets and tasks is essential for demonstrating the approach's practical significance and effectiveness.

* A more detailed and thorough comparison with existing NTK and GNN methods is necessary. The paper should highlight the proposed approach's novelty and advantages, supported by theoretical or empirical evidence, to clearly showcase the contributions and distinguish the work from existing literature.

**Questions:**

* Could the authors elaborate on how the alignment concept is related to node and graph classification tasks? Are there any practical insights or guidelines on how to effectively apply the proposed theories to these tasks?

* Can the authors highlight the novel aspects of their theoretical framework that go beyond the existing Neural Tangent Kernel (NTK) theories? What are the unique contributions that differentiate this work from existing NTK-based studies?

* Are there plans to include more comprehensive experiments to validate the proposed theories and concepts? What datasets, tasks, and baselines are considered for these experiments?

*Will there be experiments conducted specifically to verify the proposed theorems, such as theorem 2 and theorem 3? If so, could you provide insights into how these verifications will be carried out empirically?

* Why the size of $\mathbf{y}_i$ is $\mathbb{R}^{n \times 1}$, given the size of $\mathbf{x}_i$ is $\mathbb{R}^{n \times 1}$? Is there a specific rationale behind this choice of dimensionality?

* Could you elucidate the rationale behind choosing the HCP-YA dataset for your experiments? How does this dataset align with the objectives and hypotheses of your study, especially considering that common node classification or graph classification tasks are typically used in related works?

**Details Of Ethics Concerns:**

The ACKNOWLEDGEMENTS on page 13 might violate the anonymity.

---

> ### Author Response · Authors · 2023-11-17
> **Response to Reviewer TKzH (1/2)**
>
> We thank the reviewer for their insightful comments. Specific concerns have been addressed below.
>
>
> **Alignment, NTKs, and GNNs.** We clarify that the notion of alignment (defined as $y^{\sf T} \Theta y$ for NTK $\Theta$ in equation (2) and Definition 1, where $y$ is the output) is a generic property of learning with gradient descent and *not specific to only the GNN architecture or related inference tasks like node classification*. Alignment dictates the convergence of gradient descent (as established in equation (9)). Our objective was to leverage (i) these facts about alignment and gradient descent, and (ii)  specific analytical form of the NTKs for a GNN architecture; to *demonstrate that optimizing alignment for a GNN is equivalent to optimizing the choice of graph shift operator* for the supervised learning task. **To the best of our knowledge, no existing work has motivated the practical choice of graph shift operator in GNNs using either these arguments or the theoretical analyses presented in our paper.**
>
>
> **Same input and output dimensionality.** The input and the output dimensionality were assumed to be the same for analytical tractability. Such settings are of practical relevance in spatio-temporal datasets, where the multiple correlated features evolve over time (for instance, in datasets that describe brain activity, traffic flow, stock prices, etc.). In these settings, the use of past data to predict the future data is a learning task of interest.
>
>
> **Experiments.** Our theoretical results (Theorem 2 and 3) explicitly establish the relationship between the graph shift operator and cross-covariance graphs in the context of optimizing the alignment function. In this respect, our experiments were focused on validating this insight on a real dataset for a suitable practical learning task.
>
>
> **Choice of HCP-YA dataset.** HCP-YA dataset is one of the larger datasets available for brain functional activity in terms of sample size and has been used extensively by the research community. Thus, this dataset was chosen for both its suitability to the considered setting in our theoretical analyses and in the spirit of reproducibility of our findings.
>
> **Validation of theoretical results.** To validate the insights drawn from theoretical results, we considered the inference task of predicting the brain activity at the future using the current time step. This inference task forms the foundation for an algorithm that addresses a typical time series forecasting problem. Here, we compared the performance of GNNs modeled by cross-covariance graphs with GNNs that leveraged only the covariance graph estimated from the input data. As predicted by our theoretical results, the GNNs with cross-covariance graphs indeed outperformed those with covariance graphs.
>
> **Rigor of the Experiments.** We note that we validated these findings across more than 1000 individual subjects in the dataset and a noticeable performance gain can be seen on average and has been reported in our results. The effects of changing different architectural parameters such as the number of layers, number of filter taps and the nonlinearity used in the GNNs were considered in the experiments reported in the appendices, where we observed that the benefit of using cross-covariance graphs remained consistent throughout.
>
> **Additional Experiments.**  We have further expanded our empirical evaluations (see Appendix H.6) by adding two more datasets and comparing GNNs with cross-covariance graphs with additional baselines.
>
>
> - **Additional baselines.** These additional baselines include a graph based on the Pearson Correlation between the features at different nodes (which is commonly used in tasks pertaining to brain data), and another graph with edges based on the Euclidean distance between node features passed through a Gaussian kernel. (See ‘Data-driven graph construction and graph learning: A review’ by Qiao et al. for a comprehensive review of many such input-based graph construction methods). We observed that in general, the models with graphs constructed using only the input features with both linear and non-linear affinities exhibit similar performance and the models using the cross-covariance, consistently outperform the input-based graphs.
>
>
> - **More datasets.** To ensure that these results don’t just pertain to one particular dataset, we have updated the Appendix (H7) with preliminary results on two other datasets (the PEMS07 traffic flow dataset and an ECG dataset.). While the performance gap between the different models isn’t as pronounced in these other datasets compared to HCP, the cross-covariance based models still retain an advantage.

---

> ### Author Response · Authors · 2023-11-17
> **Response to Reviewer TKzH (2/2)**
>
> **Missing literature.** We thank the reviewer for suggesting the work in Huang et. al, 2021 as a relevant study to be discussed in our paper. We note that this study leverages the analyses of NTKs to investigate the impact of number of layers in GNNs on their inference. Broadly, this work leverages NTK of a GNN to motivate various architectural choices and in this respect, is similar in spirit to our work. We have discussed such works in our literature review and will add this paper to the discussion as well. However, we firmly believe that the analytical approaches and motivations relevant to our work and those in Huang et. al, 2021 are very distinct and do not overlap in scope.
>
>
> **Node classification task.**  While the extension of the definition of Alignment to tasks like node classification is straightforward, generalizing our results to such cases is not trivial and attempting to do so would vastly expand the scope of the paper. We have therefore left attempting such an extension for future work.
>
>
> **Ethical concern.** We have provided the acknowledgement for HCP-YA dataset as per the data use agreement. The authors are not among the PI names listed there and have no collaboration with them for this work.
>
>
> We hope that we addressed the reviewer's concerns sufficiently, in which case, we would be grateful if your rating of our paper could be re-evaluated. We would be happy to clarify any additional concerns.

---

### Official Review · Reviewer_GUVA · 2023-11-01

**Soundness:** 3 good
**Presentation:** 3 good
**Contribution:** 3 good
**Rating:** 8
**Confidence:** 3

**Summary:**

This paper draws on the idea that NTK based generalization is based on the associations of eigenvectors of the NTK with the data. Drawing on this alignment in the case of a GNN is used to derive the graph shift operator (i.e. equivalent of a graph laplacian) that is different from the input graph. To do this they solve an optimization to derive the graph as being the cross-covariance matrix of the input with the output.

**Strengths:**

Generalizing the NTK to GNNs expands the theory associated with this area, and the suggestion of using a cross covariance matrix involves a graph that uses both input and output variables as nodes, which is not commonly done in current GNNs. The theory could be useful in cases where the graph is not given but constructed as an affinity matrix from data as well.

**Weaknesses:**

The key weakness is that empirically VNNs and graphs that are based on covariance matrices rather than non-linear affinities have fared worse in practice. This may be an instance of the NTK not explaining the entire behavior of neural networks. Moreover the experimental validations seem fairy limited without comparison to GCNs and Graphormers and other modern graph neural network architecture.

**Questions:**

Is this a case of the theory not explaining the entire empirical phenomenon? Can further experiments be performed on a variety of kernels to see what works better in practice on common datasts/

---

> ### Author Response · Authors · 2023-11-17
> **Response to Reviewer GUVA**
>
> We thank the reviewer for their insightful comments. Specific concerns are addressed below.
>
>
> **Theoretical analysis.** Here, we emphasize that *our theory has a bottom-up characteristic*, where we have begun with the analysis of a simple graph convolutional filter$^1$ as the predictor. Due to inherent linearities in the graph filter, the NTK for a graph filter has, by default, a constant behavior with respect to learnable parameters. The analysis of alignment using NTK in this setting leads naturally to the conclusion of cross-covariance graphs being the optimal choice. Our subsequent analyses adds various different complexities, such as, non-linearities and extension to two layers for a GNN in practice. For the analysis of the GNN model, we operate under the standard assumptions of infinite width associated with NTKs and obtain conclusions similar to that for the simple graph filter setting. Therefore, due to the bottom up characteristic of our analysis and consistency of the conclusions for graph filters and GNNs, we do not believe that our theory overlooked any missing phenomenon that could have led to a drastically different choice of kernel for the considered regression task.
>
> $^1$ *This graph convolutional filter is the fundamental information processing block in the general GNN (or specifically, GCN) architecture considered in our paper.*
>
> **Cross-covariance graphs versus graphs with non-linear affinities.** Regarding the concern that graphs with non-linear affinities may outperform covariance-based graphs, we have performed additional experiments. We refer the reviewer to Appendix H.6 in the revised paper, where we have added the performance comparison between models with different graphs. These include a graph based on the Pearson Correlation between the features at different nodes (which is commonly used in tasks pertaining to brain data), and another graph with edges based on the Euclidean distance between node features passed through a Gaussian kernel. (See ‘Data-driven graph construction and graph learning: A review’ by Qiao et al. for a comprehensive review of many such input-based graph construction methods). We observed that in general, the models with graphs constructed using only the input features with both linear and non-linear affinities exhibit similar performance and the models using the cross-covariance consistently outperform the input-based graphs. As another baseline, we have also added comparison of  the performance of our models to a Fully-connected Neural Network (FCNN; which has roughly 100 times more number of learnable parameters as compared with GNN models) and we observe that the two-layer GNN with cross-covariance graph exhibits comparable performance to the FCNN.
>
>
> **Choice of GNN architecture.** We clarify that our analysis focuses on graph convolutional networks (GCNs) among the different variants of GNNs that exist in the literature. Using this setting, we have demonstrated how to leverage the theory-inspired insights into the construction for a graph from the data.  Hence, our experiments have focused on the setting consistent with that studied in the theoretical analysis to validate our results. We conjecture that a similar performance advantage will appear when using the cross-covariance matrix as a graph in more complex graph-based architectures. Expanding our theory and experiments to show the advantage of using cross-covariance in more complex models is certainly a direction that we will pursue in future work.

---

### Author Response · Authors · 2023-11-17
**Global response**

We are grateful to all reviewers for their insightful feedback and appreciation of our work. We have provided individual responses to all reviewers. In this global response, we focus on (a) the major novelties and implications of our theoretical analysis; (b) a summary of our response to the common concerns regarding experiments; and (c) future directions of our work.




**Novelty of our theoretical analysis.** The motivation of our study is based on a generic property of gradient descent, which dictates that the **alignment** between the vector of output data and NTK leads to faster convergence and better generalization. In our paper, we leverage the fact that NTK for a GNN is an explicit function of its graph shift operator and hence, optimizing alignment can inform the choice of graph shift operator in GNNs. *To the best of our knowledge, this optimization viewpoint on alignment and leveraging it for selecting a graph shift operator that leads to provable performance gains is novel when compared with other NTK-based studies in GNNs.*


Our theoretical analysis naturally provides a novel insight into constructing a graph in supervised learning tasks, based on the interdependencies between the input and output data, namely, the cross-covariance. This is in contrast to choices prevalent in existing literature that leverage only the relations between the input features for graph construction in GNN models.

**Experiments.** The reviewers raised concerns regarding the relevance and sufficiency of the experimental results with respect to the theory presented in the paper. The primary goal of our experiments was to validate the implications of our theoretical analysis on a real-world dataset. Since the proposed choice of cross-covariance-based models consistently outperformed the models with input-based graphs, our initial set of experiments succeeded in validating the theoretical results.

In addition, we have **expanded our experiments** in the revised paper as follows:


- *Additional datasets.* (see Appendix H.7)  To demonstrate that our empirical observations were not constrained to one particular setting, we have also added results for a similar multivariate time-series prediction task on two more datasets that pertain to *traffic forecasting* and *brain activity prediction for ECG modality*.


- *Graphs with non-linear affinities.* (see Appendix H.6) We have added results derived by additional choices of input-based graph constructions (including nonlinear affinities). Our results demonstrate that GNNs with cross-covariance graphs outperform these different choices of graphs constructed only from the input data.


- *Time-series prediction as specific application.* (see Appendix H.5) The setting in our initial experiment provided the building block for a formal task of time series prediction using past data. We have expanded our experiments in this direction to build a more sophisticated model for time-series prediction on the HCP dataset which utilizes more than just a single previous time step. Within this setting as well, the models with cross-covariance based graph structure outperform those with graphs based solely on the input data.

**Future Directions.** We believe that our paper lays the foundation for various research directions and concur with Reviewer bUqG’s assessment that it *“bridges complex theoretical insights with practical machine-learning applications”*. Extending the analyses to GNNs with depth more than two layers or with alternative architectures is expected to add additional analytical challenges that must be explored further. Analyzing the more practical setting where the NTK evolves during training is of particular interest since feature learning does not happen in the infinite-width constant NTK regime. Further, expanding our theoretical framework beyond the limits of our current analysis is another important direction. We have discussed these future directions in Appendix C.

---

> ### Author Response · Authors · 2023-11-22
>
> Dear Reviewers,
>
> Thanks again for your invaluable feedback.
>
> Since the author-reviewer discussion period is near its conclusion, we hope to ensure that our responses have adequately addressed your concerns. If you have any further questions or require any additional information from us, please let us know.
>
> Best regards,
>
> The Authors

---

### Meta-Review · Area_Chair_BLxT · 2023-12-07

**Metareview:**

The authors delve into the theoretical aspects of Neural Tangent Kernels (NTKs) and their influence on the learning and generalization behaviors of over-parameterized neural networks in supervised learning tasks. They introduce the concept of "alignment" between the eigenvectors of the NTK kernel and the given data, which appears to play a significant role in governing the rate of convergence of gradient descent and the generalization to unseen data. The paper specifically explores NTKs and alignment in the context of Graph Neural Networks (GNNs). The authors' analysis reveals that optimizing alignment corresponds to optimizing the graph representation or the graph shift operator within a GNN. This investigation leads to the establishment of theoretical guarantees concerning the optimality of certain design choices in GNNs.

However, the theoretical framework primarily relies on the existing NTK theory regarding optimization and generalization, which has been quite extensively studied. The experiments were conducted within a narrow set of conditions or datasets, and they might not fully represent the complexities of real-world data and scenarios. Therefore, the AC recommends rejection.

**Justification For Why Not Higher Score:**

The theoretical framework primarily relies on the existing NTK theory regarding optimization and generalization, which has been quite extensively studied. The experiments were conducted within a narrow set of conditions or datasets, and they might not fully represent the complexities of real-world data and scenarios.

**Justification For Why Not Lower Score:**

N/A

---

### Decision · Program_Chairs · 2024-01-16

Reject